# STRUCTURE GUIDED EQUATION DISCOVERY WITH INFLUENCE-BASED FEEDBACK FOR LARGE LANGUAGE MODELS

## ABSTRACT

Large Language Models (LLMs) hold significant promise for scientific discovery, particularly in identifying interpretable, closed-form equations from complex data. However, existing LLM-driven approaches often rely on coarse, scalar feedback (e.g., overall Mean Squared Error), limiting the LLM's ability to discern the individual contributions of components within a proposed equation. This forces the LLM to rely heavily on its priors or engage in inefficient trial-and-error exploration. We introduce *Structure Guided Equation Discovery* (SGED), a novel framework where LLMs act as dual agents in an iterative symbolic modeling pipeline. An LLM agent first proposes candidate basis functions $\psi_j(x)$ for a linear symbolic model $f(x) = \sum_j w_j \psi_j(x)$. A second LLM agent then refines this set of terms, critically guided by detailed, per-term *influence scores* $\Delta_j$ and fitted weights $w_j$. These scores quantify each basis function's contribution to predictive accuracy, providing the crucial granular feedback needed for effective model refinement. SGED can operate as a direct iterative refinement loop or be integrated into Monte Carlo Tree Search (MCTS) for a more comprehensive exploration of the equation space. We demonstrate that providing LLMs with this structured, influence-based feedback improves the accuracy of discovered equations and the efficiency of the discovery process on diverse biological and synthetic datasets. SGED highlights the broader principle that equipping LLMs with detailed, interpretable feedback about sub-components of their generative output can unlock more sophisticated reasoning and self-improvement capabilities.

## 1 INTRODUCTION

The quest for interpretable and generalizable mathematical models from data is a cornerstone of scientific advancement. In fields like biology, pharmacology, and physics, concise equations that accurately predict phenomena and offer mechanistic insights are invaluable for guiding research and fostering new discoveries. This paper focuses on discovering *closed-form, nonlinear symbolic models* of the form $f(\mathbf{x}) = \sum_j w_j \psi_j(\mathbf{x})$, where $\psi_j(\mathbf{x})$ are basis functions (potentially complex transformations of input features $\mathbf{x}$) and $w_j$ are their corresponding weights. The goal is to find models that are not only accurate but also interpretable.

Large Language Models (LLMs) have emerged as powerful tools for scientific tasks, including equation discovery (Ma et al., 2023b; Holt et al., 2024b). Their vast knowledge priors and reasoning capabilities allow them to navigate the complex search space of potential equations. However, current methods often guide LLMs with only a single scalar metric, such as the overall validation loss (e.g., Mean Squared Error). This coarse feedback, while indicating *if* a proposed equation is good, fails to provide information on *why* it is good or bad, or *which specific parts* of the equation contribute most to its performance or its deficiencies. Without this granular credit assignment, the LLM must resort to less efficient exploration strategies, relying heavily on its pre-trained biases or engaging in near-random perturbations. This limitation becomes particularly acute when dealing with high-dimensional data, where the number of potential input features and their interactions is vast.

To overcome this, we propose **Structure Guided Equation Discovery (SGED)**. SGED transforms LLMs into sophisticated agents within an iterative discovery pipeline by providing them with *granular,*

Table 1: **Problem Settings and Method Applicability (SGED Context).** Comparison of typical problem characteristics, system focus, and applications for different modeling paradigms. SGED targets interpretable symbolic equations for static or dynamic systems, where LLMs propose and refine basis functions for a linear model using detailed influence-based feedback and optional tree search. For a comparison with Automated Feature Engineering approaches, see Appendix A.6.

| Method Paradigm | Input Dim ($d$) | Typical Data Regime | Target Model Form | Handles Feat. Eng. ($\phi/\psi_j$)? | System Type Focus | Primary Objective | Example Applications |
|---|---|---|---|---|---|---|---|
| Classical Symbolic Reg. (e.g., GP-SR) [1] | Low ($< 20$) / Struggles with High | Small-Large | Symbolic Eq ($\mathcal{E}$) | Manual / Limited | Static (mostly) | Interpretation | Basic physics laws, simple regressions |
| Black-Box ML (NN, GBDT) [2] | Any | Large Pref. | Black-Box Prediction | Implicit (Learned Rep.) | Static / Dynamical | Prediction Accuracy | Image recognition, complex risk scoring |
| Neural ODEs [3] | Any (if deep) | Large Pref. | ODE / Dynamics | Implicit | Dynamical | Prediction (Dynamics) | Modeling physical processes |
| LLM for ODEs (e.g., D3) [4] | Low Focus | Small-Large | ODE / Dynamics | $\chi$(Assumes features) | Dynamical | Interpretation (ODE) | Discovering physics ODEs |
| **SGED (Ours)** | **Any (LLM selects from $d$)** | **Small-Large** | **Symbolic Eq ($\sum w_j \psi_j(\mathbf{x})$)** | **✓(LLM proposes $\psi_j$)** | **Static / Dynamical** | **Accuracy, Interpretation, Generalization** | **Biomarker discovery, genomic risk models** |

**Notes:** ✓: Supported/Designed for, ✗: Not primary focus/supported, Implicit: Learned internally. $\psi_j$ are basis functions. "Low Dim": struggles beyond $d \approx 20$.
SGED's current focus is regression; exploring other model forms (e.g., ODEs) with influence feedback could be explored in future work.
**References:** [1] (Stephens, 2015; Cranmer et al., 2020b), [2] (Chen & Guestrin, 2016; Gorishniy et al., 2021), [3] (Chen et al., 2018), [4] (Holt et al., 2024b)

*per-term influence feedback.* In our framework, one LLM agent proposes candidate basis functions $\psi_j(\mathbf{x})$. After these terms are used to fit a linear model, we calculate influence scores $\Delta_j$ for each term, quantifying its impact on the model's predictive performance (e.g., the change in MSE if the term were removed).

This detailed feedback, along with the fitted weights $w_j$, is then provided to a second LLM agent, which decides which terms to keep, discard, or implicitly refine for the next iteration. This process provides the LLM with the critical missing information: a clear understanding of each component's utility. This targeted feedback enables more efficient and effective exploration of the equation space, leading to more accurate and interpretable models. Furthermore, SGED can employ this iterative propose-and-prune cycle within a Monte Carlo Tree Search (MCTS) framework, allowing for a more structured and robust exploration of complex hypothesis spaces, potentially avoiding local optima.

We argue that providing LLMs with such detailed, interpretable feedback about the structural components of their proposals is a key enabler for more advanced reasoning and self-improvement in scientific discovery tasks. By understanding not just the overall quality but the specific value of individual contributions, LLMs can make more informed decisions, leading to faster convergence and ultimately, more accurate and insightful scientific models (see Figure 2). Additionally, we situate SGED within the landscape of modeling paradigms, see Table 1.

**Contributions**:

① **Conceptual Innovation**: We reframe LLM-driven equation discovery by emphasizing the need for granular, interpretable feedback. We propose per-term influence scores as a powerful mechanism to provide LLMs with the component-level understanding necessary for effective model refinement.
② **Methodological Framework (SGED)**: We introduce a novel framework employing two LLM agents: one for proposing candidate basis functions and another for pruning them, guided by per-term influence scores and weights. We detail how this iterative cycle can be enhanced with Monte Carlo Tree Search for systematic exploration.
③ **Empirical Validation**: We demonstrate through experiments on diverse biological and synthetic datasets that SGED, by leveraging influence-based feedback, discovers more accurate symbolic equations and converges more efficiently than approaches relying on coarser feedback.

## 2 METHODOLOGY: STRUCTURE GUIDED EQUATION DISCOVERY (SGED)

Structure Guided Equation Discovery (SGED) is an iterative framework where a Large Language Model (LLM) acts as an intelligent agent to discover symbolic mathematical models. The target models are of the form $f(\mathbf{x}) = \sum_{j=1}^{M} w_j \psi_j(\mathbf{x})$, where $\psi_j(\mathbf{x})$ are basis functions proposed by the LLM, and $w_j$ are their corresponding weights determined by fitting to data.

The central idea of SGED is to guide the LLM's search process through detailed, per-term influence feedback, enabling it to make informed decisions about refining the set of basis functions. This process can either follow a direct iterative refinement path or be augmented with a Monte Carlo Tree Search (MCTS) to explore a broader space of potential equations more systematically. The overall workflow is depicted in Figure 1, and we provide a concrete illustration of the discovery process in Appendix E.2.

### 2.1 CORE ITERATIVE STEP: PROPOSE-AND-PRUNE CYCLE

The engine of SGED is a propose-and-prune cycle, which constitutes a single iteration of model generation and refinement. This cycle takes the current set of basis functions (which can be empty initially or be the result of a previous iteration) and a history of past interactions and attempts. It

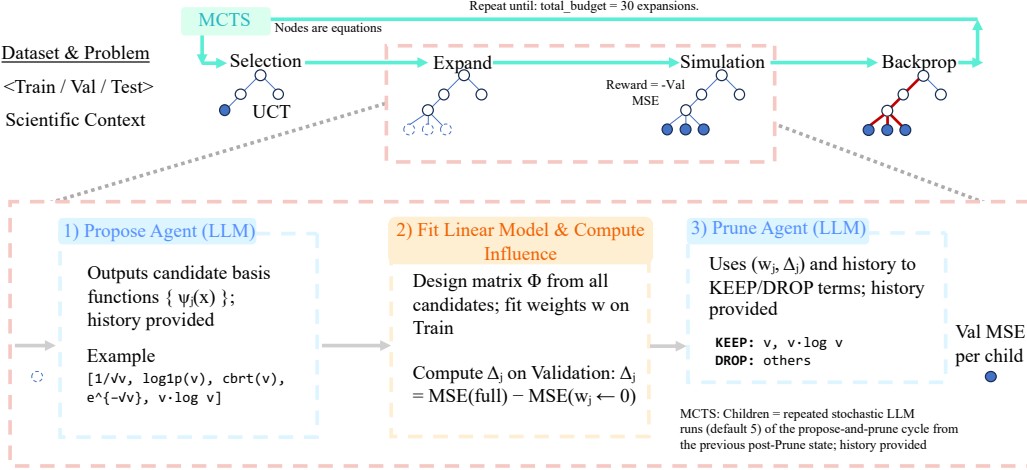

Figure 1: Conceptual block diagram of the Structure Guided Equation Discovery (SGED) method. The LLM agent iteratively proposes a set of basis functions $\{\psi_j\}$. A linear model $y = \sum w_j \psi_j(\mathbf{x})$ is fitted. The LLM receives detailed feedback, including weights $w_j$ and influence scores $\Delta_j$ for each term, and uses this to refine its proposal for the next iteration. This iterative loop can be enhanced with MCTS for structured exploration. The process aims for accurate and interpretable models by leveraging the LLM's reasoning with granular, influence-based guidance.

consists of two main phases involving LLM interaction: term generation and term pruning, with an intermediate evaluation step that calculates the crucial influence scores (see Algorithm 2 in Appendix B.4 for pseudocode).

First, in the **Term Generation Phase**, the LLM is prompted to suggest new candidate basis functions. This prompt, provides substantial context: a detailed description of the dataset and the scientific problem, a preview of the input features and target variables, the currently active set of basis functions, the best equation found so far (if applicable), and, importantly, a history of previous rounds. This history includes a summary of which terms were kept or dropped in earlier cycles and their corresponding Mean Squared Errors (MSEs), enabling the LLM to learn from its past decisions (full prompt details are in Appendix B.6). The LLM is tasked with generating a list of new terms, expressed as NumPy-compatible mathematical expressions. The number of terms to propose can be guided by configuration, for instance, suggesting more terms in the initial round versus subsequent rounds.

Next, the **Candidate Evaluation and Feedback Preparation** stage begins. The newly proposed terms are aggregated with the existing basis functions to form an expanded candidate set. Each of these candidate terms is then evaluated on the training data to construct a design matrix $\mathbf{\Phi}$. A linear model (e.g. Ordinary Least Squares (OLS), Ridge, or Lasso) is fitted to this design matrix to predict the target variable $y \approx \mathbf{\Phi w}$. This step yields an initial set of weights $\mathbf{w}$ for all candidate terms and the corresponding training MSE. A critical computation then occurs: per-term influence scores are determined. For every candidate term $\psi_j$, its influence score $\Delta_j$ represents the change in validation MSE if that specific term were removed from the model (i.e., its weight $w_j$ set to zero) while all other term weights remain fixed (see Appendix B.2 for calculation details). This vector of influence scores, $\mathbf{\Delta} = \{\Delta_j\}$, forms the core of the granular feedback provided to the LLM. The validation dataset is used for these calculations to assess generalization and mitigate overfitting. We use this computationally efficient 'no-refit' approach, which we empirically validated against more costly refit-aware alternatives and found to be as effective at guiding the discovery process in Appendix E.9.

Following this evaluation, the **Term Pruning Phase** commences. The LLM is again invoked, this time to prune the comprehensive set of candidate terms. It receives a detailed prompt, which includes: the dataset and problem description; the full list of candidate terms alongside their fitted weights $w_j$ and their calculated influence scores $\Delta_j$ (obtained from the validation set); the overall validation MSE of the model incorporating all candidate terms; the history of prior keep/drop decisions; and the human-readable current equation. The LLM is explicitly guided to utilize the influence scores as a key heuristic for its decisions, with instructions like "$\Delta_j \approx 0 \implies$ drop" and "large $\Delta_j \implies$ keep", while also encouraging the use of its own judgment. The LLM's pruning decisions are returned as a dictionary specifying which terms to "keep" and which to "drop". While the default framework

employs an LLM for pruning to incorporate semantic judgment, this step can alternatively be handled by a deterministic algorithm based on influence rankings (see Appendix J).

Finally, in the **Final Model Fitting and State Update** stage, the basis functions designated as "keep" by the LLM constitute the refined set of terms for the current iteration. A new linear model is trained exclusively with these surviving terms using the training data. Its performance metrics, namely MSE and optionally $R^2$ score, NRMSE, etc. are then evaluated on the validation set. An independent evaluation on the test set is also performed to record an unbiased measure of the model's generalization capability. All pertinent information from this entire propose-and-prune cycle — including the sets of terms before and after pruning, the LLM's keep/drop decisions, MSE values at different stages, other derived metrics, and the generated equations — is systematically collected. We also incorporate the updated history, which now includes a summary of the outcomes from the current iteration. This is the final output of one propose-and-prune cycle.

## 2.2 Search Strategies for Equation Discovery

SGED employs the propose-and-prune cycle as its core mechanism for generating and evaluating new candidate equations. Based on the configuration, SGED can adopt one of two main strategies to navigate the vast search space of possible equations.

The first strategy is a **Linear Iterative Refinement**. SGED operates by creating a linear sequence of model refinements. In this mode, the set of basis functions that survive the pruning phase directly becomes the input set of current terms for the immediately following propose-and-prune cycle. This iterative process is repeated for a predetermined number of iterations, or until a defined early stopping criterion is met. Such criteria might include observing no significant improvement in the validation MSE for a specified number of consecutive iterations, i.e., early-stopping. Each iteration in this chain is designed to incrementally enhance the quality of the equation by building upon the feedback and results from the preceding step. A comprehensive history of all generated is maintained throughout this process, enabling the system to monitor progress and, if needed, to revert to or reconsider previously high-performing solutions.

The second, more advanced strategy involves a **Tree-Based Search using Monte Carlo Tree Search (MCTS)**. The MCTS framework allows for a more structured and potentially more robust exploration of the complex hypothesis space of equations. In this configuration, the search for an optimal equation is formalized as an MCTS problem. Consequently, a node encapsulates a specific mathematical equation (defined by its constituent basis functions and their fitted weights) along with the historical sequence of decisions and refinements that led to its formulation.

The propose-and-prune procedure serves as the mechanism for generating successor states, which correspond to child nodes in the MCTS tree, from a given parent node. It is possible to generate multiple distinct successors from a single parent node by repeatedly executing the propose-and-prune, potentially introducing slight variations or leveraging inherent stochasticity in the LLM's responses (we use the latter). The evaluation of a node's quality, or its reward signal to be maximized, that we use for the MCTS algorithm, is its negative validation MSE, as a lower MSE indicates a better model. If the generation of successors is computationally expensive, the MCTS implementation can be set to use the immediate node reward for simulations or rollouts. A standard MCTS algorithm, governed by the Upper Confidence Bound for Trees (UCT) formula (Kocsis & Szepesvári, 2006) with the default exploration constant of $\sqrt{2}$ is used, balancing exploration and exploitation. The MCTS process continues until a predefined computational budget is exhausted, such as a maximum number of node expansions or a maximum search depth. The ultimate goal of the MCTS process is to identify the path through the search tree that culminates in exhibiting the best performance on the validation metric. Appendix B.3 provides full MCTS implementation details.

By enabling a parallel exploration of multiple refinement pathways, the MCTS approach helps SGED to potentially circumvent local optima that might ensnare a purely linear iterative refinement strategy.

## 2.3 Influence-Based Feedback and LLM Interaction

The effectiveness of SGED heavily relies on the quality of its interaction with the LLM. The prompts are designed to provide clear, unambiguous instructions and all necessary contextual information. The per-term influence scores, $\Delta_j$, are pivotal during the pruning phase. The prompt directly instructs the LLM:

> *Inspect every row. Decide "keep" or "drop" for each term using the rule: Use the heuristic: "$\Delta_j \approx 0 \implies drop$", "large $\Delta_j \implies keep$" and your own judgement.*

This directive empowers the LLM to synergize a quantitative, data-driven metric (the influence score) with its extensive general knowledge and reasoning capabilities. These capabilities might include assessing term complexity, predicting potential for generalization, or understanding the semantic relevance of a term within the context of the specific scientific problem description.

Furthermore, the provision of a historical record of past decisions – what terms were kept or dropped, and the resultant impact on MSE before and after pruning – facilitates a form of meta-learning or in-context learning, allowing the LLM to refine its strategies over the duration of the discovery process (Appendices B.5 and B.6 contain full details on LLM interaction and prompts). The LLM is also explicitly encouraged to consider how well the proposed terms might generalize beyond the immediate validation set. Robustness is enhanced by post-processing steps that validate the LLM's output (whether they are proposed terms or keep/drop decisions), ensuring correct formatting and evaluability, and by incorporating retry mechanisms to handle occasional LLM errors or malformed responses.

## 3 RELATED WORK

Table 2: Comparison of SGED with prior paradigms. SGED aims to overcome prior limitations via LLM guided proposal and refinement of basis functions using detailed influence-based feedback, optionally with MCTS.

| Method Paradigm | Handles High Dim ($d \approx 150+$) | Automated Basis Func. Eng. ($\psi_j$) | Interpretable Output ($\sum w_j \psi_j(\mathbf{x})$) | Feedback Granularity |
|---|---|---|---|---|
| Classical Symbolic Reg. | Often struggles | Limited / Manual | ✓ | Basic (Loss) |
| Black-Box ML (NNs, GBDT) | ✓ | Implicit (Learned Rep.) | ✗ | Basic (Loss) |
| Neural ODEs | ✓(if deep) | Implicit | ✗ | Basic (Loss) |
| LLM for Equations (e.g., D3) | Untested (low-d focus) | ✗(Assumes features given) | ✓/ Hybrid | Loss + Code Errors |
| **SGED (Ours)** | ✓ | ✓(**LLM proposes $\psi_j$**) | ✓(**Linear Comb. of $\psi_j$**) | **Detailed (Per-term Influence Scores)** |

Our work, SGED, intersects with and differentiates itself from several research areas, as summarized in Table 2, and we provide an extended related work in Appendix A.

**Symbolic Regression (SR).** Traditional SR techniques, such as genetic programming (GP) (Koza, 1994; Schmidt & Lipson, 2009) and sparse regression methods like SINDy for dynamical systems (Brunton et al., 2016), aim to find explicit mathematical equations. While effective for certain problems, these methods often operate on a predefined set of input features and basic mathematical operations. They can struggle with high-dimensional inputs or require significant manual feature engineering to define relevant transformations. Some modern SR approaches like PySR (Cranmer, 2023) incorporate more sophisticated search algorithms and a wider range of operators but typically do not leverage the generative and reasoning capabilities of LLMs for proposing complex basis functions or utilize semantic feedback like per-term influence scores. SGED differentiates itself by employing an LLM to actively generate and refine these basis functions $\psi_j(\mathbf{x})$, which can be arbitrarily complex, guided by specific, quantitative influence feedback for each proposed component.

**Black-Box Models.** Machine learning models like neural networks (NNs) (Chen et al., 2018; Gorishniy et al., 2021) and gradient-boosted decision trees (GBDTs) (Chen & Guestrin, 2016) are highly effective at fitting complex patterns in data and can handle high-dimensional inputs. However, their internal workings are often opaque, making them "black boxes" that lack the interpretability of symbolic equations. While techniques such as SHAP (Lundberg & Lee, 2017) or LIME (Ribeiro et al., 2016) can provide post-hoc explanations for feature importance, they do not directly yield a concise, closed-form mathematical model. SGED, in contrast, aims to produce inherently interpretable models.

**LLM-driven Equation Discovery.** The use of LLMs for scientific discovery, including equation generation, is a rapidly advancing field. Systems like AI Feynman (Udrescu & Tegmark, 2020) have shown success in rediscovering physics equations from data. Eureka (Ma et al., 2023b) employed LLMs as part of a reward modeling system for reinforcement learning in SR tasks, but not as the direct generator and refiner of basis functions based on granular feedback. The D3 framework (Holt et al., 2024b) utilizes LLMs for discovering ODEs, focusing on dynamical systems primarily from lower-dimensional data. While D3 involves iterative refinement, its feedback mechanisms are generally coarser (e.g., overall loss, code execution errors) compared to the per-term influence

scores used in SGED. Other recent methods also leverage LLMs in distinct ways. ICSR (Merler et al., 2024) uses in-context learning where an LLM is prompted with previous attempts and their scalar scores (combining MSE and complexity) to generate better candidates. LLM-SR (Shojaee et al., 2025a) treats equations as programs and uses data-driven feedback on the overall program fit to guide refinement. LaSR (Grayeli et al., 2024) evolves a high-level, abstract "concept library" (e.g., "exponential growth/decay") by observing patterns in successful equations.

SGED's novelty lies in its distinct feedback mechanism. Unlike the abstract concepts in LaSR or the single scalar scores in ICSR and LLM-SR, SGED provides a direct, quantitative vector of per-term influence scores ($\Delta_j$). This feedback measures each basis function's marginal contribution to validation accuracy, offering interpretable, component-level credit assignment. This granular guidance, coupled with a dual-agent architecture for proposing and pruning terms within a constrained linear model structure ($\sum w_j \psi_j(\mathbf{x})$), allows for a more analytical and targeted model refinement process.

**Influence Functions and Model Interpretability.** The concept of influence functions in statistics (Cook & Weisberg, 1980), traditionally measures the impact of individual data points on model parameters or predictions. Our use of "influence scores" for terms in a linear model is analogous: it assesses the importance of a structural component (a basis function $\psi_j$) to the overall model fit, similar to a leave-one-out analysis performed at the term level. This provides a principled way to assign credit to parts of the model, which the LLM then uses for refinement.

**Automated Feature Engineering (AFE).** SGED shares conceptual parallels with AFE methods that automate the construction of input transformations, such as AutoFeat (Horn et al., 2019) and the LLM-driven CAAFE (Hollmann et al., 2023). However, whereas standard AFE aims to augment the feature space to maximize the predictive performance of downstream black-box models (e.g., gradient-boosted trees), SGED optimizes for a single, globally interpretable symbolic equation. Our approach differentiates itself by enforcing a sparse linear structure and utilizing per-term influence scores to guide the LLM's refinement process, rather than relying solely on global validation metrics or tree-based feature importances (see Appendix A.6 for a detailed comparison).

## 4 EXPERIMENTS AND EVALUATION

We evaluate SGED on a variety of datasets to demonstrate its ability to discover accurate, interpretable white-box models. Our experiments focus on biomedical, bioinformatics, and pharmacokinetic domains, reflecting real-world challenges where such models are highly valuable. Full experimental details, including dataset descriptions, method configurations, and evaluation protocols, are provided in the Appendices (Appendices B, C and C.5).

**Benchmark Datasets**. Our evaluation uses six datasets. Three are derived from a sophisticated biomedical Pharmacokinetic-Pharmacodynamic (PKPD) model of lung cancer tumor growth, simulating effects of chemotherapy and radiotherapy (Geng et al., 2017, Appendix C.1), used in prior research (Bica et al., 2020; Seedat et al., 2022; Melnychuk et al., 2022). These are: **Lung Cancer** (no treatment), **Lung Cancer (with Chemo.)**, and **Lung Cancer (with Chemo. & Radio.)**. We also use a COVID-19 epidemic agent-based simulator (**COVID-19**, Kerr et al., 2021, Appendix C.2). For bioinformatics, we use an eNET-seq dataset for predicting RNA Polymerase II pausing (**RNA Polymerase**, Fong et al., 2022, Appendix C.4). Finally, a real-world Pharmacokinetic (PK) dataset of Warfarin patients (**Warfarin**, Janssen et al., 2022, Appendix C.3) is included. Additionally, we perform evaluation on a set of problems from **LLM-SRBench** (Shojaee et al., 2025b, Appendix F).

**Benchmark Methods**. We compare SGED against several relevant methods. For black-box comparisons, these include neural ODEs with action inputs (**DyNODE**, Chen et al., 2018; Alvarez et al., 2020), standard Recurrent Neural Networks (**RNN**), and a state-of-the-art Transformer model (**Transformer**, Vaswani et al., 2017). For white-box model discovery, we include Sparse Identification of Nonlinear Dynamics (**SINDy**, Brunton et al., 2016) and a Genetic Programming symbolic regression method (**GPLearn**, Stephens, 2015). We also compare against variants of LLM-based discovery: a zero-shot model generated by an LLM (**ZeroShot**), this model with optimized parameters (**ZeroOptim**), and an LLM-based iterative approach using only basic MSE feedback without influence scores (**ICL - Basic Feedback**). Furthermore, we benchmark against several recent state-of-the-art LLM-based frameworks: (**D3**, Holt et al., 2024b), which discovers ODEs through an iterative multi-agent approach; (**ICSR**, Merler et al., 2024), which uses in-context learning with previous

attempts and their scalar scores to generate new candidates; (**LLM-SR**, Shojaee et al., 2025a), which treats equations as programs and employs an evolutionary search; and (**LaSR**, Grayeli et al., 2024), which enhances genetic algorithms by using an LLM to evolve a library of abstract textual concepts.

**Evaluation Metrics**. To assess the performance of our benchmark methods, we use the mean squared error (MSE) on a held-out test dataset of state-action trajectories. This evaluation is conducted over multiple seeds, each initialized with different random seeds. We report the average MSE from these runs along with their 95% confidence intervals (see Appendix C.6 for the full protocol).

## 5 MAIN RESULTS

Comprehensive evaluations across the benchmark datasets are presented in Table 3. SGED (referred to as "Ours" in the table, representing the full method with influence feedback with iterative refinement and MCTS) consistently demonstrates strong performance, often achieving the lowest MSE among interpretable model classes and competitive results against black-box models. This indicates its ability to discover accurate and concise closed-form equations. The predictive performance and symbolic recovery metrics evaluated on LLM-SRBench are reported in Appendix F, including evaluations of the algorithmic pruning variant (Appendix J).

Table 3: **Evaluating Method Performance.** Test MSE (mean±95 % CI) on held-out data for six benchmarks. SGED demonstrates competitive or superior performance, particularly among interpretable models. Results are based on 25 seeds; GPT-4o `2024-11-20` is used as the LLM backbone. Three vertical subsections correspond to baseline classes: (1) *white-box non-LLM baselines*, (2) *white-box LLM baselines*, (3) *black-box baselines* (dark gray font). Dashes (—) indicate "unable to run", reason clarified in footnote. Average rank computed for white-box methods only.

| Method | Lung Cancer MSE ↓ | Lung Cancer (with Chemo.) MSE ↓ | Lung Cancer (with Chemo. & Radio.) MSE ↓ | COVID-19 MSE ↓ | RNA Polymerase MSE ↓ | Warfarin PK MSE ↓ | Avg. Rank ↓ |
|---|---|---|---|---|---|---|---|
| SINDy | 335±3.14 | 0.838±0.00635 | 0.664±0.00514 | 1.43e-07±4.48e-09 | —† | 0.914±0.169 | 4.80 |
| GPLearn | 7.56±1.11 | 46.8±15.5 | 46.8±4.91 | 0.000713±0.000506 | 0.0204±0.000555 | 2.53±0.169 | 7.33 |
| PySR | 669±7.86 | 0.12±0.0697 | 0.399 ± 0.123 | 4.23e-07±8.3e-08 | —† | 29.9±0.859 | 5.60 |
| ZeroShot | 2.13e+13±4.35e+13 | 4.97e+03±3.67e+03 | 2.54e+03±2.74e+03 | 1.34e+08±2.09e+08 | 1.35e+05±1.75e+05 | 5.3e+03±1.07e+04 | 12.17 |
| ZeroOptim | 0.142±0.119 | 86.2±27.3 | 122±6 | 1.41e-07±1.04e-07 | 0.0130±0.000287 | 0.861±0.177 | 5.50 |
| ICL (Basic Feedback) | 0.0557±0.0486 | 21.2±9.8 | 63.3±16.5 | 9.35e-08±1.77e-08 | 0.0119±0.000352 | 0.784±0.193 | 3.83 |
| D3-white-box | 1.01e+04±1.27e+04 | 45±28.9 | 253±273 | 7.81e-06±2.48e-07 | 0.043±0.0366 | 1.15±0.343 | 8.00 |
| LLM-SR | 33.4±0 | 42.2±35.6 | 32.1±48.4 | 0.000453±0.000912 | 1.44±1.91 | 1.24±0.564 | 7.00 |
| LaSR | 658±7.31 | 1.71±1.16 | 3.97±3.21 | 2.59e-06±8.66e-07 | 0.0172±0.000649 | 30.1±0.992 | 6.33 |
| ICSR | 0.407±0.244 | 0.688±0.39 | 6.1±1.05 | 1.03e-07±1.6e-08 | —‡ | **0.497**±0.0646 | 3.20 |
| DyNODE | 497±18.4 | 355±3.63 | 398±14.2 | 7.48e-05±5.58e-07 | 0.0136±6.7e-05 | 0.593±0.0769 | — |
| RNN | **0.00133**±0.000485 | 0.00685±0.0022 | **0.013**±0.00552 | 5.88e-08±1.5e-09 | 0.00728±4.79e-05 | 0.756±0.139 | — |
| Transformer | 0.0375±0.00653 | 0.179±0.0135 | 0.366±0.0269 | 4.05e-07±5.05e-08 | **0.00724**±3.3e-05 | 0.804±0.158 | — |
| XGBoost | 0.012±0.000964 | 0.155±0.00488 | 0.322±0.00884 | 8.43e-08±5.36e-09 | 0.00758±3.79e-05 | 1.67±0.473 | — |
| **SGED (Ours)** | **0.0033**±0.0035 | **0.0054**±**0.00107** | **0.0521**±0.0178 | **5.32e-08**±**1.35e-09** | **0.0115**±0.000312 | 0.646±0.105 | **1.17** |

† Method did not finish within maximum runtime of 3 hours per seed due to the large number of features in this dataset (263).

‡ ICSR implementation fails with this dataset due to the large number of features in this dataset (263).

### 5.1 CASE STUDY: RNA POLYMERASE II PAUSING

The discovery of quantitative rules governing biological processes is crucial for advancing our understanding of life. Automated machine learning approaches like SGED offer a path to generate interpretable, data-driven hypotheses from complex biological datasets, potentially accelerating discovery.

**Biological Background.** RNA polymerase II (Pol II) transcription is a fundamental process involving initiation, elongation, and termination (Cramer, 2019). Transcription speed is non-uniform, influenced by frequent Pol II pausing (Noe Gonzalez et al., 2021; Jonkers & Lis, 2015; Danko et al., 2013; Bentley, 2014; Zamft et al., 2012). Pause sites, particularly at G residues preceding T/C on the non-template DNA strand, are key determinants of elongation speed (Fong et al., 2022; Gajos et al., 2021). Nucleosomes and histone modifications like H3K36me3 are also implicated (Bondarenko et al., 2006; Churchman & Weissman, 2011; Lee et al., 2024; Wen et al., 2014), but their precise roles and the sequence determinants of pausing in human cells remain incompletely understood.

**Dataset.** We analyzed eNET-seq data mapping Pol II pause sites in human cells at single-base resolution (Fong et al., 2022). A "pause score" ($N_{\text{reads at pause site}}/N_{\text{reads in 200bp window}}$) quantified pausing. This score was aligned with 263 features, including local DNA sequence context around potential pause sites (e.g., one-hot encoded nucleotides A/T/G/C at positions -3 to +3 relative to site), MNase-seq signal (nucleosome occupancy), H3K4me3 and H3K36me3 ChIP-seq signals (histone modifications), and gene region annotations (TSS, gene body, termination). A balanced dataset of

48,000 pause sites and 48,000 control sites (pause score = 0) was split into training, validation, and test sets.

**Findings.** SGED, guided by influence-based feedback, discovered the following interpretable equation (coefficients rounded for brevity):

$$
\begin{aligned}
\text{pause\_score} = {}& 0.0178 \ln\big(1 + \text{signal}_{\text{MNase}}\big) \\
& - 0.000246 \, \text{signal}_{\text{H3K4me3}} + 0.00902 \ln\big(1 + \text{signal}_{\text{H3K4me3}}\big) \\
& + 0.0194 \ln\Big(1 + \sum_{\text{cond}} \text{signal}_{\text{H3K36me3,ds}}\Big) \\
& - 0.0291 \big(\mathbb{1}_{\{\text{seq}_{-1}=A\}} + \mathbb{1}_{\{\text{seq}_0=A\}} + \mathbb{1}_{\{\text{seq}_1=A\}} + \mathbb{1}_{\{\text{seq}_{-1}=T\}} + \mathbb{1}_{\{\text{seq}_0=T\}} + \mathbb{1}_{\{\text{seq}_1=T\}}\big) \\
& + 0.0223 \, \mathbb{1}_{\{\text{gene\_region=TSS}\}} + 0.0257 \, \mathbb{1}_{\{\text{gene\_region=body}\}} + 0.0636 \, \mathbb{1}_{\{\text{gene\_region=termination}\}} \\
& - 0.0146 \big(\mathbb{1}_{\{\text{seq}_{-3}=T\}} + \mathbb{1}_{\{\text{seq}_{-2}=T\}} + \mathbb{1}_{\{\text{seq}_{-1}=T\}}\big) \\
& - 0.0402 \big(\mathbb{1}_{\{\text{seq}_{-1}=G\}} + \mathbb{1}_{\{\text{seq}_{-1}=C\}}\big) + 0.0333 \, \mathbb{1}_{\{\text{seq}_0=G\}} - 0.0397 \, \mathbb{1}_{\{\text{seq}_0=C\}} \\
& + 0.0735 \, \mathbb{1}_{\{\text{seq}_1=T\}} + 0.0243 \, \mathbb{1}_{\{\text{seq}_{-1}=G\}},
\end{aligned}
$$

where "ds" sums over the H3K36me3 signals from several alternative data sources; $\text{seq}_i$ is the nucleotide at relative position $i$; $\mathbb{1}_{\{\cdot\}}$ is the indicator function.

Expert review by an Anonymous Biologist confirmed that this SGED-discovered equation aligns with known biology and offers novel insights:

> **Expert review (Anonymous Biologist)**
>
> ▶**Confirms existing knowledge:** Positive association of nucleosome occupancy ($\text{signal}_{\text{MNase}}$) and H3K36me3 with pausing, and higher pausing in termination regions, consistent with prior studies (Bondarenko et al., 2006; Churchman & Weissman, 2011; Lee et al., 2024; Wen et al., 2014; Gromak et al., 2006).
> ▶**Discovers novel sequence elements:** Beyond the known GT element at positions 0/+1 (Fong et al., 2022; Gajos et al., 2021), the model identifies C at positions 0 and -1, and T at -1, -2, -3 as significant negative predictors of pausing. These represent new, testable hypotheses about sequence-dependent pausing mechanisms.

This case study demonstrates SGED's capacity to learn complex relationships from high-dimensional biological data, producing interpretable models that both validate existing knowledge and generate novel scientific hypotheses. Further discussion is in Appendix D.

We furthermore conducted a targeted wet-lab experiment to empirically verify a novel hypothesis regarding DNA methylation suggested by SGED-discovered equation for a different variant of this experiment (detailed in Appendix D.3); the experimental results *confirmed the SGED prediction*.

## 5.2 INSIGHT EXPERIMENTS: IMPACT OF FEEDBACK AND SEARCH STRATEGY

To understand the contributions of SGED's core components, we performed ablation studies on the Lung Cancer benchmark dataset.

**Ablation Study: Influence Feedback and MCTS.** We compared the full SGED model against variants where either the MCTS tree search component was disabled (falling back to iterative refinement) or the detailed influence based feedback was removed (LLM receives only basic MSE for pruning), or both. Results are shown in Table 4. The reported MSE values are for the Lung Cancer (no treatment) dataset, averaged over 25 seeds.

Table 4: **Ablation study on the Lung Cancer (no treatment) dataset.** We quantify the impact of removing the MCTS component and/or the influence-feedback module. Results are test MSE (lower is better) averaged over 25 seeds with 95% confidence intervals.

| Variant | MCTS | Influence Feedback | MSE $\downarrow$ |
|---|:---:|:---:|:---:|
| **Full SGED (Ours)** | ✓ | ✓ | **0.0033**±0.0035 |
| w/o MCTS (Iterative + Influence Feedback) | ✗ | ✓ | 0.350±0.505 |
| w/o Influence Feedback (MCTS + Basic Feedback) | ✓ | ✗ | 4.56±6.71 |
| w/o MCTS or Influence Feedback (Iterative + Basic Feedback) | ✗ | ✗ | 61.98±20.52 |

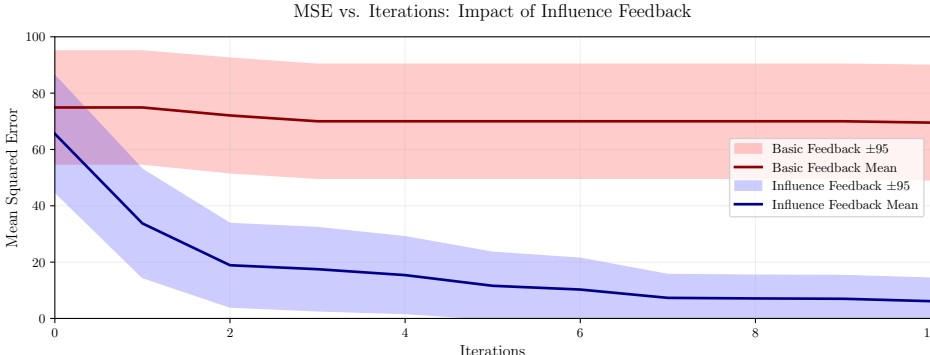

Figure 2: **MSE vs. Iterations: Impact of Influence Feedback.** Plot showing validation MSE convergence over iterations for SGED (linear iterative refinement variant) with full influence-based feedback versus a variant with only basic MSE feedback on the *Lung Cancer (with Chemo. & Radio.)* dataset. Detailed influence feedback leads to faster convergence and lowers final MSE. Analogous plot for the MCTS setting is shown in Figure 5.

Shaded regions denote 95% confidence intervals, solid lines show means. For equivalent comparison: both variants start with a candidate pool of 6 terms and 6 terms are kept after each propose-and-prune iteration; at iteration 0 performance before pruning is shown to ensure equivalent starting point.

The results clearly indicate that both the influence-based feedback and the MCTS tree search contribute significantly to SGED's performance. Removing influence feedback (even with tree search) leads to a substantial increase in MSE (from 0.0033 to 4.56), demonstrating that granular, per-term guidance is crucial for the LLM to effectively prune and refine equations.

Similarly, removing the tree search (even with influence feedback, MSE increases to 0.350) also results in higher MSE, suggesting that systematic exploration helps in finding better solutions compared to a purely linear iterative approach for this benchmark. The variant without both components performs the worst (MSE of 61.98), underscoring the synergistic benefits of these design choices.

**Iterative Improvement with Influence Feedback.** Figure 2 illustrates the convergence behavior, comparing SGED with influence feedback to a variant relying on basic MSE feedback. SGED with influence feedback converges faster and to lower MSE values, as the detailed per-term information allows for more targeted and efficient exploration of the equation space. Each iteration leverages the insights from influence scores to prune unhelpful terms and focus on promising ones, leading to a more rapid discovery of accurate and parsimonious models. This is shown for the linear iterative refinement variant of SGED here, and analogously for the MCTS variant in Appendix E.1.

These insight experiments underscore the importance of SGED's core principles: providing LLMs with *rich, structured feedback at the component level* and employing *systematic search strategies* to navigate the complex landscape of possible equations. A suite of additional experiments in the appendix further substantiates these findings. We demonstrate that SGED's performance advantage is *consistent across nine different underlying LLMs*, including open-weight models (Appendix E.5); that it is *robust to a large number of irrelevant features* (Appendix E.6); that it can successfully discover equations for diverse synthetic models (Appendix E.7); and, importantly, *generalizes robustly to unseen biological replicates*, indicating the discovered models capture reproducible scientific principles (Appendix E.8). Furthermore, the method *scales sub-linearly with an increasing number of input features*, making it suitable for high-dimensional problems (Appendix H). Critically, SGED also proves more computationally efficient, *achieving superior accuracy when constrained by a fixed LLM token budget* (Appendix H.3) and *converging significantly faster* than traditional methods like Genetic Programming (Appendix E.4). Methodological variants, including an extension for term-local constant optimization (Appendix E.10) and an ablation on alternative influence score calculations (Appendix E.9), are also explored. Finally, we assess the system's sensitivity to prompt variations (Appendix E.13) and hyperparameter choices regarding term proposal width (Appendix E.12), and quantify the diversity of the generated candidate terms (Appendix E.14).

## 6 DISCUSSION

This paper introduced Structure Guided Equation Discovery (SGED), a framework that leverages Large Language Models for symbolic modeling by providing them with fine-grained, per-term influence scores as feedback. This structured guidance allows the LLM to iteratively propose, evaluate, and refine basis functions for constructing interpretable models of the form $f(\mathbf{x}) = \sum_j w_j \psi_j(\mathbf{x})$. Integration of Monte Carlo Tree Search further enhances the systematic exploration of the equation space. Our empirical results across diverse datasets, including a detailed case study on RNA Polymerase II pausing, demonstrate that SGED can discover accurate and scientifically plausible equations. The ablation studies confirm that both the influence-based feedback mechanism and the MCTS contribute significantly to the framework's effectiveness, leading to lower MSE compared to variants lacking these components. The ability to identify important terms via influence scores and to prune irrelevant ones based on this quantitative metric, combined with the LLM's reasoning capabilities, proves to be a powerful approach for navigating the vast search space of symbolic models and achieving superior accuracy.

**Limitations & Future Work.** Though SGED's architectural advantages provide a robust performance uplift across a range of LLMs, its performance is still ultimately tied to the capabilities of the underlying language model. While current state-of-the-art models show strong reasoning abilities, their capacity to generate highly novel or counter-intuitive scientific insights based on the provided feedback is an area for ongoing research. The complexity of basis functions that can be reliably proposed and evaluated also presents a frontier. While SGED is computationally efficient, running many LLM calls at scale can still be intensive. Future work could explore more efficient LLM prompting strategies, methods for distilling learned heuristics from the LLM's successful refinements, and extending the framework to other types of mathematical models (e.g., differential equations with more complex structures, causal graphs). Further research into the types of feedback that are most effective for LLM-guided discovery remains a rich area of investigation.

**Ethics Statement.** Our work promotes transparency in AI-driven science by developing a method, SGED, that generates interpretable symbolic models. The primary ethical consideration is the potential for misinterpretation; a discovered equation is a data-driven hypothesis, not a validated scientific law, and may reflect spurious correlations in the data. We stress that any model produced by SGED must be rigorously scrutinized and validated by domain experts before any consideration for real-world application to prevent potential harm from decisions based on flawed but plausible-looking equations. The datasets used in this research were either simulated or publicly available and anonymized, and were handled in accordance with their usage terms. We position SGED as a tool to augment, not replace, human scientific inquiry, acknowledging that the underlying Large Language Models may introduce biases and lack genuine scientific understanding.

**Reproducibility Statement.** To ensure full reproducibility of our findings, we provide a comprehensive account of our methodology, data, and experimental setup. The core Structure Guided Equation Discovery (SGED) framework is described in Section 2, with a detailed breakdown of the propose-and-prune cycle (Section 2.1), search strategies (Section 2.2), and the influence-based feedback mechanism (Section 2.3). For implementation, high-level pseudocode is available in Appendix B.4, and an in-depth description of all methodological components, including the calculation of influence scores and MCTS configuration, is provided in Appendix B. The specifics of our interaction with Large Language Models, including the models used and the exact prompt templates for the "Propose" and "Prune" agents, are documented in Appendix B.5 and Appendix B.6, respectively. All benchmark datasets are thoroughly described in Appendix C, which details the simulation parameters for the Cancer PKPD and COVID-19 environments and provides sources for the public Warfarin PK and RNA Polymerase datasets. The experimental evaluation protocol, including benchmark methods and metrics is specified in Section 4 and further detailed in Appendix C.5. Hyperparameters and settings are collated in Appendix I. The computational resources used for our experiments are outlined in Appendix G. Upon publication, all code required to generate the datasets and reproduce the results will be made publicly available.

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

# Appendix

## Table of Contents

**Code.** All code will be made available upon publication.

**LLM Usage.** Large Language Models (LLMs) were employed for refining the language, grammar, and clarity of the manuscript. Additionally, they were used to assist with code implementation and debugging. All intellectual content, research ideas, and scientific arguments were developed solely by the authors.

## A   ADDITIONAL RELATED WORK

Structure Guided Equation Discovery (SGED) synergizes concepts from symbolic regression, the rapidly evolving capabilities of Large Language Models (LLMs) in scientific reasoning, machine learning interpretability, and advanced search techniques. This extended related work section aims to provide a more comprehensive contextualization of SGED, elaborating on its distinctions and contributions by delving deeper into these intersecting domains.

### A.1   SYMBOLIC REGRESSION: FOUNDATIONS AND EVOLUTION

Symbolic Regression (SR) is the problem of identifying a mathematical expression that best fits a given dataset, without assuming a pre-specified model structure. This inherently seeks interpretable models.

**Traditional Approaches:** Genetic Programming (GP) has historically been a dominant paradigm for SR (Koza, 1994; Schmidt & Lipson, 2009; Stephens, 2015). GP-based SR typically evolves a population of candidate expressions (often represented as trees) using evolutionary operators like crossover and mutation. While powerful, traditional GP can face challenges such as premature convergence, code bloat (expressions becoming overly complex), and difficulties in efficiently exploring vast search spaces, especially with high-dimensional data or when a diverse set of mathematical operators is required (Cranmer, 2023; Lu et al., 2021; Monda et al., 2021; Li et al., 2023a). The search can be computationally intensive, and the quality of discovered equations can be sensitive to the choice of initial function sets and hyperparameters. Other early approaches, like Symbolic Regression via Fast Function Class Discovery (SRFC) (Udrescu & Lipson, 2009), focused on identifying general classes of functions as a preliminary step, which could then guide more detailed equation discovery.

**Sparse Symbolic Regression:** Methods like SINDy (Sparse Identification of Nonlinear Dynamics) (Brunton et al., 2016) leverage sparse regression. SINDy constructs a library of candidate (often nonlinear) functions of the state variables and uses techniques like LASSO or sequentially thresholded least-squares to find a sparse combination of these functions that best describes the system's dynamics, primarily for ODEs. While effective for systems where the basis functions are well-chosen, the library of candidate functions is often pre-defined by the user, which might limit discovery of truly novel functional forms not anticipated by the domain expert.

**Modern Advancements in SR:** More recent SR methods have introduced innovations to tackle these challenges. PySR (Cranmer, 2023; Cranmer et al., 2020b) incorporates techniques from simulated annealing, genetic algorithms, and a highly optimized search process with a broad library of operators

to find Pareto-optimal equations (balancing accuracy and complexity). AI Feynman (Udrescu & Tegmark, 2020; Udrescu et al., 2020) introduced a recursive divide-and-conquer strategy inspired by physics problem-solving techniques. It attempts to discover symmetries, separability, and other properties of the target function to break it down into simpler components, often using neural networks to guide these decompositions. Deep Symbolic Regression (DSR) approaches (Petersen et al., 2020; Biggio et al., 2021; Kamienny et al., 2022) often use recurrent neural networks (RNNs) to generate expressions token by token, framing SR as a sequence generation problem, sometimes guided by reinforcement learning. Other neural approaches like SymbolNet (Cheng et al., 2024) focus on scalability to high-dimensional inputs and model compression by dynamically pruning operators and features.

**SGED's Differentiation in SR:** SGED distinguishes itself from these SR paradigms in several key aspects. Unlike traditional GP or sparse SR methods that rely on pre-defined or combinatorially generated basis functions, SGED tasks an LLM with proposing candidate basis functions $\psi_j(\mathbf{x})$. These can be arbitrarily complex and draw upon the LLM's vast pre-trained knowledge of mathematical and scientific relationships. Crucially, SGED provides the LLM with highly granular, *per-term influence scores* $\Delta_j$ as feedback, guiding a second LLM agent in the pruning and refinement process. This explicit, quantitative credit assignment for each component of the proposed equation is a core novelty that enables more targeted and efficient exploration than typical fitness-based evolution in GP or the global loss signals used in many DSR approaches. While AI Feynman uses NNs for specific decomposition tasks, SGED employs LLMs more broadly for generative proposal and structured refinement based on component-wise utility.

A.2 LARGE LANGUAGE MODELS IN SCIENTIFIC DISCOVERY AND EQUATION FORMULATION

LLMs are increasingly being explored as powerful tools for accelerating scientific discovery (OpenAI, 2023; Brown et al., 2020; Agashe et al., 2024). Their ability to process and generate human language, understand complex instructions, and synthesize information from vast training corpora makes them suitable for tasks ranging from hypothesis generation to experimental design and data analysis (Boiko et al., 2023; Dunn & Arrow, 2022; Ma et al., 2023a; Tonchev et al., 2024).

**LLMs for Equation Discovery and System Modeling:** Several works have specifically investigated LLMs for discovering mathematical models. Eureka (Ma et al., 2023b;c) leverages LLMs, particularly their code-writing capabilities, to design reward functions for reinforcement learning agents that then perform symbolic regression or other optimization tasks. While Eureka uses LLMs effectively for reward generation, SGED employs LLMs as the direct architects of the symbolic equations themselves, iteratively proposing and refining basis functions. The D3 framework (Holt et al., 2024b;a) uses LLMs to discover Ordinary Differential Equations (ODEs) for pharmacological systems. D3 employs multiple LLM agents for modeling, feature acquisition, and evaluation in an iterative loop. While sharing the iterative, LLM-driven discovery spirit with SGED, the feedback mechanism in D3 is generally coarser (e.g., overall model fit, code execution errors, qualitative evaluation). SGED's unique contribution is the fine-grained, per-term influence scores $\Delta_j$ provided to the LLM, enabling more precise credit assignment and refinement of the equation's structure, and its applicability to general symbolic models beyond just ODEs. D-CODE (Qian et al., 2022) also focuses on discovering closed-form ODEs, using a grammar-based approach and a coefficient optimizer, but does not involve LLMs in the same generative and iterative feedback loop as SGED or D3.

**Broader LLM Capabilities for Science:** LLMs have also shown promise in generating computer code for simulations (Li et al., 2022; Chen et al., 2023), assisting in mathematical reasoning (Imani et al., 2023; Lewkowycz et al., 2022), and forming hypotheses from literature (Bran et al., 2023; Zhang et al., 2023b). SGED specifically harnesses the LLM's pattern recognition and generative capabilities to propose scientifically plausible basis functions and its reasoning capabilities to interpret and act upon the structured influence feedback for model refinement. This structured interaction, providing quantitative evidence for the utility of each proposed component, pushes the LLM beyond simple generation towards a more analytical role.

A.3 INTERPRETABILITY IN MACHINE LEARNING: INHERENT VS. POST-HOC

The demand for interpretable machine learning models is growing, especially in high-stakes domains like science and medicine where understanding the "why" behind a prediction is as important as the prediction itself (Rudin, 2019; Doshi-Velez & Kim, 2017).

**Black-Box Models and Post-Hoc Explanations:** Many high-performing machine learning models, such as deep neural networks (Gorishniy et al., 2021; Chen et al., 2018) and gradient-boosted decision trees (Chen & Guestrin, 2016), are often considered "black boxes" due to their complex internal structures. To shed light on their predictions, post-hoc explanation methods have been developed. Prominent examples include LIME (Local Interpretable Model-agnostic Explanations) (Ribeiro et al., 2016) and SHAP (SHapley Additive exPlanations) (Lundberg & Lee, 2017). LIME approximates the black-box model locally with a simpler, interpretable model. SHAP uses concepts from cooperative game theory (Shapley values) to attribute the prediction to individual features. While these methods provide valuable insights, they offer explanations of an already-trained (and often opaque) model rather than producing a model that is inherently transparent. Limitations can include computational cost, potential instability of explanations, and the fact that the explanation itself is an approximation of the original model's behavior (Rudin, 2019; Kumar et al., 2020; Alvarez-Melis & Jaakkola, 2018).

**SGED's Pursuit of Inherent Interpretability:** SGED directly addresses the need for interpretability by aiming to discover models of the form $f(\mathbf{x}) = \sum_j w_j \psi_j(\mathbf{x})$. Each basis function $\psi_j(\mathbf{x})$ is a symbolic expression, and its contribution to the final output is explicitly weighted by $w_j$. This structure is inherently interpretable, allowing domain experts to examine, understand, and potentially validate or refute the discovered relationships based on their domain knowledge. The per-term influence scores $\Delta_j$ further enhance this by quantifying the contribution of each $\psi_j$ to the model's predictive power during the discovery process itself.

A.4 NEUROSYMBOLIC AI

Neurosymbolic AI seeks to combine the strengths of neural networks (e.g., learning from data, pattern recognition) with symbolic reasoning (e.g., logic, explicit knowledge representation, interpretability) (Garcez & Lamb, 2023; Chaudhuri et al., 2021; Besold et al., 2017). This integration aims to create AI systems that are more robust, generalizable, interpretable, and capable of incorporating existing domain knowledge.

SGED can be viewed as a neurosymbolic system. The LLM, a large neural network, acts as the "neuro" component, responsible for proposing candidate symbolic basis functions and for reasoning about their utility based on feedback. The symbolic regression task itself, the manipulation of mathematical expressions, and the resulting interpretable equation $f(\mathbf{x}) = \sum_j w_j \psi_j(\mathbf{x})$ represent the "symbolic" component. The per-term influence scores act as a critical bridge, translating numerical performance data into a structured format that the LLM can symbolically reason about to refine the symbolic model. This tight integration, where the neural component generates and refines symbolic structures based on quantitative feedback about those structures, aligns well with the goals of neurosymbolic AI, particularly in the context of scientific discovery (Cranmer et al., 2020a; Kubalík et al., 2023).

A.5 INFLUENCE ANALYSIS: FROM DATA POINTS TO MODEL COMPONENTS

The concept of "influence" in SGED, referring to the impact of individual terms $\psi_j(\mathbf{x})$ on the model's predictive performance, draws an analogy to classical influence functions in statistics and more recent feature importance techniques.

**Classical Influence Functions:** Influence functions, introduced in Hampel (1974) and further developed in Cook & Weisberg (1980), measure the effect of an individual data point on a model's parameters or predictions. They are valuable for outlier detection and understanding model sensitivity to specific observations.

**Feature Importance and Leave-One-Out Analysis:** In machine learning, various methods assess feature importance. Permutation feature importance (Breiman, 2001; Fisher et al., 2019) measures the decrease in model performance when a feature's values are randomly shuffled. Leave-One-Feature-Out (LOFO) importance involves retraining the model with one feature omitted and observing the

performance change (Lei et al., 2018; Aerdem, 2020). These methods help identify which input features are most critical for a model's predictions.

**SGED's Term Influence Scores:** SGED's per-term influence scores $\Delta_j$ adapt this concept to the components of the *discovered equation itself*. Instead of assessing the impact of raw input features or individual data points, $\Delta_j$ quantifies how much the removal of a specific basis function $\psi_j(\mathbf{x})$ (and its corresponding weight $w_j$) would affect the model's validation MSE. This provides a direct, interpretable measure of each term's contribution to the model's accuracy, akin to a leave-one-term-out analysis. This granular credit assignment is then fed back to the LLM, enabling it to make informed decisions about which terms to retain, discard, or refine, thereby steering the discovery process towards more accurate and parsimonious equations.

## A.6 RELATIONSHIP TO AUTOMATED FEATURE ENGINEERING

| Method | LLM / semantic signal | Generation mechanism | Feedback & selection | Typical predictor | Output representation |
|---|---|---|---|---|---|
| AutoFeat (Horn et al., 2019) | ✗ (no LLM; optional physical units) | Predefined analytic transforms and pairwise operators on numeric features | Correlation filtering and L1-regularised linear models with stability selection | Linear regression / classification | Small set of analytic feature expressions used in a linear model |
| OpenFE (Zhang et al., 2023a) | ✗ | Expand–and–reduce over unary/binary operators (incl. aggregations and group-by) | FeatureBoost residual fitting + two-stage pruning with GBDT feature importance | GBDT / LightGBM | Augmented tabular feature matrix consumed by tree ensembles |
| RuleFit (Friedman & Popescu, 2008) | ✗ (no LLM; tree-ensemble rules) | Decision rules extracted from ensembles of trees and converted to binary features | Sparse linear model (e.g., L1-regularised regression) over rule indicators | Sparse linear model on rule features (rules derived from trees) | Set of human-readable decision rules with weights capturing non-linear interactions |
| CAAFE (Hollmann et al., 2023) | ✓ (LLM + dataset description) | LLM-generated Python feature-engineering code and textual explanations | Keep only transformations that improve validation metrics of downstream classifiers | Logistic regression, random forests, TabPFN | Dataset with additional engineered columns plus natural-language rationales |
| FeatLLM (Han et al., 2024) | ✓ (LLM + few-shot examples) | LLM-generated per-class logical rules converted to binary features | Linear model trained on rule indicators; learned weights used to rank rules | Linear classifier on rule features | Weighted rule set defining class-specific decision heuristics |
| OCTree (Nam et al., 2024) | ✓ (LLM + decision-tree reasoning) | LLM-generated rule-based column generators iteratively refined | Validation score of prediction model + CART-based natural-language reasoning | XGBoost, MLP, HyperFast | Set of rule-defined columns that boost generic tabular predictors |
| **SGED (ours)** | ✓ (LLM + structure-guided feedback) | LLM-generated symbolic basis functions $\psi_j$ inside a linear equation | Global per-term influence scores $\Delta_j$ and pruning within a propose–and–prune loop | Sparse linear equation model | Closed-form symbolic equation $g(\mathbf{x}) = \sum_j w_j \psi_j(\mathbf{x})$ with ranked term contributions |

Table 5: Conceptual comparison between SGED and representative automated feature engineering methods for tabular data. All approaches can be seen as generating additional features, but they differ in how the search space is specified, what feedback is used for refinement, which downstream predictors they target, and whether the final output is a generic feature set or an explicit symbolic equation.

Automated feature engineering (AFE) aims to construct informative transformations of raw inputs, because the performance of many machine learning models depends heavily on the quality of the input representation. Classical approaches include polynomial expansions and other predefined transformations, while deep neural networks can be viewed as implicitly performing feature engineering by learning hierarchical representations of the data. However, these learned representations are typically not symbolic and are difficult to interpret analytically.

While SGED is framed as a symbolic equation discovery method, it can also be viewed as an automated feature discovery mechanism for linear models: the basis functions $\psi_j(\mathbf{x})$ play the role of engineered features and the symbolic equation $g(\mathbf{x}) = \sum_j w_j \psi_j(\mathbf{x})$ is a sparse linear model on top of them. This naturally connects SGED to the literature on automated feature engineering and to tree-based tabular learners.

**Classical automated feature engineering.** AutoFeat (Horn et al., 2019) provides a scikit-learn style regression/classification interface in which a large pool of non-linear transformations of the inputs (e.g., polynomial, logarithmic and rational combinations) is constructed and then aggressively

pruned using correlation filters and L1-regularised linear models with stability selection. The aim is to retain a small set of analytic features that make a linear model competitive while remaining globally interpretable, with optional use of physical units to filter dimensionally inconsistent expressions. OpenFE (Zhang et al., 2023a) follows an expand–and–reduce strategy on tabular data: it enumerates candidate features using a library of unary and binary operators (including aggregations and group-by operators), and evaluates them via the FeatureBoost procedure together with a coarse-to-fine pruning scheme driven by gradient boosting decision trees (GBDTs) and tree-based feature attributions. In both cases, the search space is predefined, the optimisation is driven purely by predictive performance, and the final predictor is either a linear model (AutoFeat) or a tree ensemble (OpenFE).

**Rule-based feature generation.**   Methods such as RuleFit (Friedman & Popescu, 2008) generate features in the form of decision rules. An ensemble of decision trees (e.g., gradient-boosted trees) is first trained on the original features, then each internal path is converted into a binary rule feature indicating whether a sample satisfies the corresponding conjunction of threshold conditions. A sparse linear model (typically with L1 regularisation) is then fitted on both the original features and these rule-based features. This yields an interpretable model that can capture non-linearities and feature interactions through a relatively small set of weighted rules derived from trees, sitting at the intersection of rule-based AFE and linear modelling.

**LLM-based feature generation.**   Recent work combines AFE with large language models. CAAFE (Hollmann et al., 2023) prompts an LLM with a natural-language description of the dataset and prediction task and asks it to generate executable Python code that adds new columns, together with free-form textual justifications of their utility. The proposed transformations are accepted only if they improve validation performance of downstream classifiers such as logistic regression, random forests or TabPFN. FeatLLM (Han et al., 2024) targets few-shot tabular learning: an LLM is prompted with a task description and a small labelled set and asked to produce per-class rule sets (e.g., conjunctions of simple threshold conditions) that are converted into binary features. A linear model is then trained on these rule indicators, yielding both a prediction rule and a ranking of human-readable rules by their learned weights. OCTree (Nam et al., 2024) views feature engineering as optimisation over rule-based column generators. An LLM proposes rules for new columns, which are iteratively refined using two feedback channels: validation scores of prediction models such as XGBoost and MLPs, and natural-language decision-tree "reasoning" extracted from CART models fitted on the current dataset. Compared to classical AFE, these LLM-based methods replace a hand-crafted operator library with a language prior and use natural-language context (column names, descriptions, decision-tree paths) as additional guidance, while still primarily optimising black-box predictive performance on tabular benchmarks.

**SGED as structure-guided feature engineering for symbolic equations.**   Viewed through the AFE lens, SGED also uses an LLM to propose new "features" $\psi_j(\mathbf{x})$ and keeps them only if they improve a downstream model. However, there are two key differences to the methods above. First, SGED constrains the downstream model to be a sparse linear equation

$$g(\mathbf{x}) = \sum_{j=1}^{m} w_j\,\psi_j(\mathbf{x}),$$

which is itself the object of interest: the set of basis functions and their weights are meant to be read as a compact, global symbolic model of the system under study rather than as a feature set for an arbitrary classifier. In contrast to typical rule-generation algorithms that operate over simple threshold rules, SGED's LLM-driven basis function proposal is not limited to rules and can express complex symbolic compositions of inputs, including non-linear operations and domain-specific transformations that would be difficult to enumerate by hand. The subsequent linear combination $\sum_j w_j\psi_j(\mathbf{x})$ maintains global interpretability, while the influence-guided pruning ensures that only valuable, LLM-generated transformations are retained.

Second, the feedback that drives LLM refinement in SGED is explicitly structure-aware: the LLM sees per-term influence scores $\Delta_j$ that quantify the change in global mean-squared error when each term is removed from the equation. This complements tree-based feature importance or SHAP-style attributions, which operate on pre-defined features and often describe local contributions to individual predictions rather than the global role of a term in a closed-form equation. In other words,

SGED can be viewed as a structure-guided, neurosymbolic AFE mechanism whose output is a single interpretable equation, not just an augmented feature matrix.

**Relationship to tree-based models and explainability.** Strong tree ensembles such as XG-Boost (Chen & Guestrin, 2016) and LightGBM (Ke et al., 2017) remain among the best-performing predictors for tabular data and underpin many AFE systems, including OpenFE and OCTree. In these methods, interpretability is typically provided either at the feature level (via split counts or impurity-based importance) or through local explanation methods such as SHAP values computed on the engineered features. RuleFit also leverages tree ensembles, but distils their structure into a sparse linear model over rule indicators, bringing it closer in spirit to SGED's linear formulation while still relying on tree-based rule extraction. SGED instead optimises a single sparse linear equation and reports global term-level contributions through the coefficients $w_j$ and influence scores $\Delta_j$. This sacrifices some expressive power relative to deep ensembles of trees, but yields a model class in which the full predictive mechanism can be inspected analytically and related directly to scientific priors (e.g., additivity, saturation, or known limiting behaviours). Conceptually, SGED is therefore complementary to tree-based pipelines: SGED can be seen as a structure-guided feature generator whose discovered terms may also be fed into tree ensembles, while tree-based methods provide powerful black-box baselines against which the fidelity of SGED's symbolic approximations can be assessed.

## A.7 ITERATIVE REFINEMENT AND SEARCH STRATEGIES IN COMPLEX SPACES

Discovering optimal symbolic equations is a challenging search problem over a vast and complex space.

**Iterative Refinement and Feedback:** Many AI systems employ iterative refinement, where solutions are progressively improved based on feedback. The nature and granularity of this feedback are crucial. SGED's iterative propose-and-prune cycle, guided by per-term influence scores, provides a structured mechanism for self-improvement. The LLM learns from its past decisions (which terms were kept/dropped and their impact) and the specific utility of each component in the current proposal. This contrasts with approaches where feedback is only a single scalar loss, offering less guidance for targeted improvement. LLM self-correction and refinement, often through in-context learning, is an active area of research (Madaan et al., 2023; Shinn et al., 2023; Pan et al., 2023). SGED provides a domain-specific instantiation of this principle with a highly structured feedback signal.

**Monte Carlo Tree Search (MCTS):** For more systematic exploration, SGED can integrate its propose-and-prune cycle into an MCTS framework. MCTS is a heuristic search algorithm that has achieved remarkable success in domains like game playing (e.g., AlphaGo/AlphaZero (Silver et al., 2016; 2017)) and other optimization problems. In SGED, MCTS treats the equation discovery process as navigating a tree where nodes represent (sets of) basis functions and edges represent propose/prune actions. MCTS balances exploration of new equation structures with exploitation of promising ones. While MCTS has been explored for program synthesis and, more recently, symbolic regression (Khalil et al., 2022; Li et al., 2023b; Chen et al., 2024; Ashok et al., 2020), SGED's novelty lies in combining MCTS with LLM-driven proposal/pruning stages that are informed by the detailed influence-based feedback. The work Li et al. (2023b) also uses MCTS with transformers for symbolic regression, but their feedback and reward structure within MCTS differs from SGED's influence-score-driven pruning by an LLM agent.

By leveraging detailed, interpretable feedback and powerful search strategies, SGED aims to make significant strides in the automated discovery of accurate and understandable symbolic models from data.

# B METHOD DETAILS

This section provides further details on the Structure Guided Equation Discovery (SGED) framework, complementing the main description in Section 2. We elaborate on the core components, including the definition and calculation of influence scores, and the specifics of the Monte Carlo Tree Search (MCTS) implementation.

## B.1 OVERVIEW OF SGED

As outlined in Section 2 and depicted in Figure 1, SGED operates as an iterative symbolic modeling pipeline driven by Large Language Models (LLMs). The framework employs two primary LLM agents:

- A **"Propose" Agent**: This LLM agent is responsible for generating new candidate basis functions $\psi_j(\mathbf{x})$. It receives contextual information about the dataset, the scientific problem, the current set of basis functions, the best equation discovered so far, and a history of past decisions and outcomes to guide its suggestions.

- A **"Prune" Agent**: After candidate terms are evaluated, this LLM agent refines the set of basis functions. It is provided with the candidate terms, their fitted weights $w_j$, and crucially, their per-term influence scores $\Delta_j$. Based on this granular feedback and its general knowledge, it decides which terms to keep or discard.

This propose-and-prune cycle forms the core iterative step. SGED can execute these cycles in a linear iterative refinement loop or integrate them within a Monte Carlo Tree Search (MCTS) framework for a more systematic and robust exploration of the equation space. The overall objective is to discover accurate, parsimonious, and interpretable models of the form $f(\mathbf{x}) = \sum_j w_j \psi_j(\mathbf{x})$. The hyperparameters related to LLM prompting, such as `terms_per_round` (default: 5), `first_round_n_candidates` (default: 10), and `keep_n_terms` (default: 6, but can be disabled to allow keeping *any* number of terms), guide the LLM agents' behavior during term generation and pruning.

## B.2 INFLUENCE SCORE ($\Delta_j$) DETAILS

The per-term influence score, $\Delta_j$, is a cornerstone of the SGED framework, providing the granular feedback necessary for the "Prune" LLM agent to make informed decisions.

**Definition**: For a linear model $f(\mathbf{x}) = \sum_{k=1}^{M} w_k \psi_k(\mathbf{x})$ with $M$ basis functions, the influence score $\Delta_j$ for a specific term $\psi_j(\mathbf{x})$ is defined as the change in the model's Mean Squared Error (MSE) on the validation set if that term were removed from the model, *while all other term weights $w_k$ ($k \neq j$) remain fixed at their originally fitted values*.

Let $\mathbf{\Phi}_{\text{val}}$ be the design matrix evaluated on the validation set using all $M$ candidate basis functions, and $\mathbf{y}_{\text{val}}$ be the corresponding true target values. Let $\mathbf{w} = (w_1, w_2, \ldots, w_M)^T$ be the vector of weights obtained by fitting the full model to the training data (e.g., using Ordinary Least Squares, OLS). The prediction of the full model on the validation set is $\hat{\mathbf{y}}_{\text{val}} = \mathbf{\Phi}_{\text{val}} \mathbf{w}$. The MSE of the full model on the validation set is $\text{MSE}_{\text{full}} = \text{mean}((\mathbf{y}_{\text{val}} - \hat{\mathbf{y}}_{\text{val}})^2)$.

Now, consider removing term $\psi_j$. This is equivalent to setting its weight $w_j$ to zero. The predictions of this reduced model, $f_{-j}(\mathbf{x})$, on the validation set are $\hat{\mathbf{y}}_{\text{val},-j} = \hat{\mathbf{y}}_{\text{val}} - \phi_{\text{val},j} w_j$, where $\phi_{\text{val},j}$ is the $j$-th column of $\mathbf{\Phi}_{\text{val}}$ (i.e., the evaluations of $\psi_j(\mathbf{x})$ on the validation set). The MSE of this reduced model is $\text{MSE}_{-j} = \text{mean}((\mathbf{y}_{\text{val}} - \hat{\mathbf{y}}_{\text{val},-j})^2)$. The influence score $\Delta_j$ is then:

$$\Delta_j = \text{MSE}_{-j} - \text{MSE}_{\text{full}}$$

A higher positive $\Delta_j$ indicates that removing term $\psi_j$ increases the validation MSE, implying that the term is important for the model's predictive accuracy on unseen data. A $\Delta_j \approx 0$ suggests the term has little unique contribution to reducing validation MSE under the current model. This calculation is performed for each target variable if the model is multi-output.

For a rigorous derivation of the influence score on validation data (where the orthogonality of residuals assumed in the simplified OLS formula does not strictly hold) and an empirical comparison against refit-aware alternatives, please refer to Appendix E.9.

**Calculation**: The calculation steps are:

1. Fit the full linear model (e.g., OLS) using the training data $(\mathbf{\Phi}_{\text{train}}, \mathbf{y}_{\text{train}})$ to obtain the weights $\mathbf{w}$.

2. Calculate the predictions $\hat{\mathbf{y}}_{\text{val}}$ and the baseline $\text{MSE}_{\text{full}}$ on the validation set $(\mathbf{\Phi}_{\text{val}}, \mathbf{y}_{\text{val}})$.

3. For each term $\psi_j$ (from $k = 1$ to $M$):
   (a) Calculate the predictions of the model without term $j$ (effectively $w_j = 0$, other $w_k$ fixed): $\hat{\mathbf{y}}_{\text{val},-j} = \hat{\mathbf{y}}_{\text{val}} - \phi_{\text{val},j} w_j$.
   (b) Calculate $\text{MSE}_{-j}$ using $\hat{\mathbf{y}}_{\text{val},-j}$.
   (c) Compute $\Delta_j = \text{MSE}_{-j} - \text{MSE}_{\text{full}}$.

**Computational Cost**:

- Fitting the initial OLS model: Typically $O(N_{\text{train}} M^2 + M^3)$ if $N_{\text{train}} > M$ or $O(N_{\text{train}}^2 M)$ if $M > N_{\text{train}}$.
- Calculating $\hat{\mathbf{y}}_{\text{val}}$: $O(N_{\text{val}} M)$.
- Calculating $\text{MSE}_{\text{full}}$: $O(N_{\text{val}})$ (assuming multi-output $m$ is small, otherwise $O(N_{\text{val}} m)$).
- For each of the $M$ terms, calculating $\Delta_j$:
  - Prediction $\hat{\mathbf{y}}_{\text{val},-j}$ adjustment: $O(N_{\text{val}})$.
  - $\text{MSE}_{-j}$ calculation: $O(N_{\text{val}})$.
  - Total for all $\Delta_j$: $M \cdot O(N_{\text{val}})$.

The calculation of influence scores is therefore efficient once the initial model is fitted and its predictions on the validation set are obtained.

**Justification and Relation to Other Measures**:

- **Direct Relevance to Predictive Performance**: $\Delta_j$ directly quantifies how much a term contributes to reducing error on unseen (validation) data, which is a primary goal.
- **Efficiency**: Calculating $\Delta_j$ by fixing other weights is much more computationally efficient than refitting the model $M$ times (once for each term's removal). This makes it practical for iterative refinement loops with many candidate terms.
- **Interpretability for LLM Guidance**: The concept of "change in error if term is removed" is intuitive and can be effectively communicated to an LLM, especially with heuristics like "$\Delta_j \approx 0 \implies$ drop".
- **Differentiation from other importance measures**:
  - It is a form of "leave-one-out" importance but applied to model terms (basis functions) rather than individual data points (like statistical influence functions, e.g., Cook's distance).
  - It differs from SHAP values (Lundberg & Lee, 2017), which explain the contribution of features to individual predictions rather than the global impact of a term on overall model MSE.
  - It is distinct from feature importance measures derived from tree-based ensembles (e.g., Gini importance or permutation importance on raw input features), as $\Delta_j$ is specific to the contribution of pre-defined or LLM-proposed basis functions $\psi_j(\mathbf{x})$ within a linear model structure.
  - By not refitting the model for each term removal, $\Delta_j$ measures the unique contribution of a term *given the current set of other terms and their weights*. This is a deliberate choice to assess the marginal utility of a term in the specific context of the current full model. If terms are highly collinear, this score might be low for some of them, aiding in pruning redundant terms.

The use of validation data for calculating $\Delta_j$ is critical for assessing generalization and mitigating overfitting, guiding the LLM to select terms that are robustly beneficial.

We further explore and experimentally investigate alternative approaches to computing influence scores in Appendix E.9.

### B.3 MONTE CARLO TREE SEARCH (MCTS) IMPLEMENTATION DETAILS

When SGED employs a tree-based search strategy, it utilizes a Monte Carlo Tree Search (MCTS) algorithm to navigate the complex hypothesis space of symbolic equations. This approach allows for a more structured exploration than simple iterative refinement.

- **State (Node Representation)**: Each node in the MCTS tree represents a specific state in the equation discovery process. This state is defined by the current set of selected basis functions $\{\psi_j(\mathbf{x})\}$, the corresponding fitted weights $\{w_j\}$ that form the current best equation for that path, and the history of decisions (proposals, prunings, and feedback) that led to this state.

- **Action (Transition)**: An action involves transitioning from a parent node (current equation state) to a child node (a refined equation state). This transition is achieved by executing one full **propose-and-prune cycle** as described in Section 2.1. Specifically, from a selected leaf node, the "Propose" agent suggests new terms, these are evaluated, influence scores are calculated, and the "Prune" agent refines the term set. This results in a new equation that defines a child node.

- **Expansion**: During the expansion phase of MCTS, if a selected leaf node is not a terminal state (e.g., maximum depth not reached), child nodes are generated. SGED can generate multiple distinct successor states (child nodes) from a single parent node by repeatedly invoking the propose-and-prune cycle. The hyperparameter `n_successors` (default: 5) controls how many child nodes are attempted to be generated from a parent node during a single expansion step. Variation between these successors arises from the inherent stochasticity in the LLM's responses to the propose and prune prompts, even when given the same historical context up to the parent node. Each such generated child represents a distinct path for exploration.

- **Reward and Value Estimation (Simulation/Rollout Phase)**: The quality or *value* of a node (equation) needs to be estimated to guide the MCTS search.

  - In the default SGED configuration, referred to as **Heuristic MCTS** (when the hyperparameter `rollout_is_just_node_reward` is set to `True`, which is the default), no explicit rollout (simulation beyond the newly expanded node) is performed. Instead, the reward for a newly expanded child node is its immediate, directly computed quality. This quality is typically the negative validation Mean Squared Error ($-\text{MSE}_{\text{val}}$) of the equation associated with that child node (after the pruning step). A higher value (lower MSE) indicates a better node. This immediate reward is then directly backpropagated up the tree.

  - If `rollout_is_just_node_reward` is set to `False`, a simulation phase (rollout) would be executed from the newly expanded node for a number of steps defined by `rollout_depth` (default: 1, see Appendix E.3 for an investigation). The reward from the end of this simulated trajectory would then be backpropagated. However, the primary results in this paper use the Heuristic MCTS approach.

  - Other reward signals beyond simply the negated validation set MSE could be used, though are not explored here, e.g. incorporating alternative accuracy metrics or accounting for equation parsimony (e.g. as $R = -\text{MSE}_{\text{val}} - \alpha \cdot \textit{Complexity}$, where complexity may be defined as the number of terms, operator count, etc.).

- **Backpropagation**: After a node's value is determined (either through direct evaluation in Heuristic MCTS or via a rollout), this value is backpropagated up the tree from the expanded node to the root. The visit counts and average reward (or value) of all visited nodes along the path are updated.

- **Selection (Tree Policy)**: The MCTS algorithm uses the Upper Confidence Bound applied to Trees (UCT) formula to balance exploration and exploitation when selecting which node to traverse down the tree. From a current node, the child $i$ selected is the one that maximizes:

$$\text{UCT}(i) = Q(i) + c\sqrt{\frac{\ln N(p)}{N(i)}}$$

where:

  - $Q(i)$ is the current estimated average reward (exploitation term) for child $i$.
  - $N(p)$ is the number of times the parent node $p$ has been visited.
  - $N(i)$ is the number of times child node $i$ has been visited.
  - $c$ is the exploration constant (hyperparameter, default: $\sqrt{2}$). A higher $c$ encourages more exploration of less-visited nodes.

- **Termination**: The MCTS process continues until a predefined computational budget is exhausted. This is primarily controlled by the `total_budget` hyperparameter (default: 30), which defines the total number of MCTS iterations (selection, expansion, simulation/evaluation, backpropagation cycles). A `depth_limit` (default: 10) can also restrict the maximum depth of the tree.

- **Output**: After the MCTS process completes, the equation corresponding to the node that yielded the best validation MSE throughout the entire search is typically selected as the final discovered model.

- **Computational Cost**: The main computational costs in the MCTS-based SGED are the LLM API calls (for term proposal and pruning at each expansion, potentially multiple times if `n_successors` > 1), the evaluation of basis functions on data, and the fitting of linear models. The total number of MCTS iterations (`total_budget`) is the primary driver of the overall computational expense.

This MCTS framework allows SGED to systematically explore diverse pathways of equation refinement, potentially avoiding local optima that a simpler iterative approach might encounter.

B.4 SGED Pseudocode

Below is a high-level pseudocode outlining the SGED framework. Algorithm 1 describes the overall MCTS-driven search. Algorithm 2 details the core propose-and-prune cycle (`PAPC`).

For brevity, Algorithm 1 illustrates the Heuristic MCTS case, but this can be easily adapted to the rollout with `rollout_depth` case.

---

**Algorithm 1** Structure Guided Equation Discovery (SGED) - MCTS Loop

---

1: **Input:** Dataset $\mathcal{D}$ (containing train, validation, test sets), Problem Description $P_d$, MCTS Budget $B_{\text{MCTS}}$, LLM Agents ($LLM_P$, $LLM_R$), MCTS Parameters (exploration const $c$, num successors $n_s$, depth limit $d_{lim}$)
2: **Output:** Best discovered symbolic equation $f_{\text{best}}$
3: Initialize $f_{\text{best}} \leftarrow$ None
4: Initialize $best\_val\_mse \leftarrow \infty$
5: Initialize MCTS Tree $T$ with a root node (representing an initial empty state)
6: **for** $iteration = 1$ to $B_{\text{MCTS}}$ **do**
7:    $selected\_node \leftarrow$ SelectPolicy$(T, c)$ {Select a node using UCT}
8:    **if** $selected\_node$ is suitable for expansion (e.g., not terminal by $d_{lim}$, and not yet fully expanded for $n_s$ children) **then**
9:       {Attempt to expand $selected\_node$ by generating up to $n_s$ children}
10:       **for** $i = 1$ to $n_s$ **do**
11:          $PAPC\_Inputs \leftarrow$ GatherInputsForPAPC$(selected\_node, P_d, \mathcal{D}, LLM_P, LLM_R, f_{\text{best}})$

12:          $equation, val\_mse, new\_terms, history \leftarrow$ ProposeAndPruneCycle$(PAPC\_Inputs)$
13:          **if** $equation$ was successfully generated **then**
14:             $child\_node \leftarrow$ AddChildNode$(T, selected\_node, new\_terms, equation, val\_mse, history)$

15:             $current\_reward \leftarrow -val\_mse$ {Heuristic reward for UCT}
16:             BackpropagateValue$(child\_node, current\_reward, T)$
17:             **if** $val\_mse < best\_val\_mse$ **then**
18:                $best\_val\_mse \leftarrow val\_mse$
19:                $f_{\text{best}} \leftarrow equation$
20:             **end if**
21:          **end if**
22:       **end for**
23:       Update expansion status of $selected\_node$ in $T$.
24:    **else**
25:       $current\_reward \leftarrow -selected\_node.stored\_val\_mse$ {Use node's known value}
26:       BackpropagateValue$(selected\_node, current\_reward, T)$
27:    **end if**
28: **end for**
29: **if** $f_{\text{best}}$ is not None **then**
30:    Evaluate $f_{\text{best}}$ using test set from $\mathcal{D}$
31: **end if**
32: **return** $f_{\text{best}}$

---

---

**Algorithm 2** Propose-and-Prune Cycle

---

1: **Input:** $current\_terms_{in}$, $history_{in}$, $best\_equation\_so\_far$, $P_d$, $\mathcal{D}_{\text{train}}$, $\mathcal{D}_{\text{val}}$, $LLM_P$, $LLM_R$

2: **Output:** $equation_{out}$, $val\_mse_{out}$, $final\_terms_{out}$, $history_{out}$

3:

4: *// Term Generation Phase*

5: Prompt $LLM_P$ with $P_d$, current active terms $current\_terms_{in}$, $best\_equation\_so\_far$, $history_{in}$, and data preview.

6: $newly\_proposed\_terms \leftarrow LLM_P$.generate_terms()

7: $candidate\_terms \leftarrow current\_terms_{in} \cup newly\_proposed\_terms$

8:

9: *// Candidate Evaluation and Feedback Preparation*

10: Evaluate all $\psi_j \in candidate\_terms$ on $\mathcal{D}_{\text{train}}$ to get design matrix $\mathbf{\Phi}_{\text{train}}$.

11: Fit linear model: $\mathbf{y}_{\text{train}} \approx \mathbf{\Phi}_{\text{train}}\mathbf{w}_{cand}$ to get candidate weights $\mathbf{w}_{cand}$.

12: Evaluate all $\psi_j \in candidate\_terms$ on $\mathcal{D}_{\text{val}}$ to get $\mathbf{\Phi}_{\text{val}}$.

13: Calculate full validation MSE: $\text{MSE}_{\text{val,cand}}$ using $\mathbf{w}_{cand}$ and $(\mathbf{\Phi}_{\text{val}}, \mathbf{y}_{\text{val}})$.

14: For each $\psi_j \in candidate\_terms$, calculate influence score $\Delta_j$ on validation set (as per Appendix B.2).

15:

16: *// Term Pruning Phase*

17: Prompt $LLM_R$ with $P_d$, $candidate\_terms$, their weights $w_j$, influence scores $\Delta_j$, $\text{MSE}_{\text{val,cand}}$, $history_{in}$, and current equation form.

18: $pruning\_decisions \leftarrow LLM_R$.decide_terms_to_keep_drop() {keep: [...], drop: [...]}

19: $final\_terms_{out} \leftarrow$ terms marked "keep" in $pruning\_decisions$

20:

21: *// Final Model Fitting and State Update*

22: Evaluate $final\_terms_{out}$ on $\mathcal{D}_{\text{train}}$ to get $\mathbf{\Phi}_{\text{train,final}}$.

23: Fit final linear model: $\mathbf{y}_{\text{train}} \approx \mathbf{\Phi}_{\text{train,final}}\mathbf{w}_{\text{final}}$ to get $\mathbf{w}_{\text{final}}$.

24: Form $equation_{out}$ using $final\_terms_{out}$ and $\mathbf{w}_{\text{final}}$.

25: Calculate $val\_mse_{out}$ for $equation_{out}$ on $\mathcal{D}_{\text{val}}$.

26: Record this cycle's details (terms before/after pruning, decisions, MSEs, equation) into $history_{out}$.

27:

28: **return** $equation_{out}$, $val\_mse_{out}$, $final\_terms_{out}$, $history_{out}$

---

This pseudocode provides a conceptual blueprint. Actual implementations would involve detailed prompt engineering, error handling, and specific choices for the linear model fitting procedure (e.g., OLS, Ridge).

## B.5 LLM DETAILS

Unless otherwise specified (e.g., in Appendix E.5 which details experiments across a range of different Large Language Models), the primary LLM employed for the core experiments presented in this work was a version of GPT-4o. Specific model versions were used as available at the time of experimentation. For example, one of the models used in comparative evaluations was:

- GPT-4o: model identifier `gpt-4o`, version `2024-11-20`.

## B.6 PROMPT DETAILS

The interaction with the LLM is central to the SGED framework. We employ two primary types of interactions, mediated by distinct LLM agents: one for proposing new candidate basis functions (the "Propose" agent) and another for refining the set of terms based on evaluation feedback (the "Prune" agent). Below, we detail the structure of these prompts and provide illustrative examples of LLM interactions.

### B.6.1 TERM GENERATION PHASE: THE "PROPOSE" AGENT

In this phase, the LLM is tasked with generating new candidate basis functions $\psi_j(\mathbf{x})$. The prompt provides comprehensive context, including the problem description, available features, the current set of basis functions (if any), the best equation found so far, and a history of previous interactions to facilitate learning from past attempts.

**Prompt Template for "Propose" Agent:** The following is a representative template for the prompt provided to the "Propose" agent. Specific details such as dataset descriptions, feature lists, and historical performance are dynamically inserted.

Listing 1: Prompt Template for "Propose" Agent

```
You are an automated assistant for proposing linear terms for the equations in a symbolic regression pipeline.

Your proposed terms will be:
1. Concatenated with the current candidate terms.
2. Sent to a LLM term pruner agent that will use various computed signals to decide which terms to keep and
    which to drop.

# Instructions:
Given below information, propose some candidate terms. The terms can be any valid numpy expressions.
Make use of the dataset and problem description to propose relevant terms.
Make sure to use the learnings from the history of previous rounds.

Return these between triple backticks and one term on each line.
The first backticks must be prepended with TERMS

Example output:

TERMS
```
x1
x2**2
np.sin(x3)
```

NOTES:
* Propose around {terms_per_round} terms, generally not too many unless this is the first round.
* If this *is* the first round, propose around {first_round_n_candidates} terms.
* You may propose no terms (nothing between the TERMS backticks) if you think it is appropriate.

DATASET AND PROBLEM DESCRIPTION
-----------------------------
{dataset_and_problem_description}

CURRENT TERMS:
-----------------------------
{current_terms}

# Current equation:
{current_equation}

HISTORY
-----------------------------
{history}

=========
The input data and target variable(s) preview:
{input_preview}
```

The prompt template for the "Propose" agent dynamically incorporates several pieces of information. Key placeholders include:

**terms_per_round**: Hyperparameter specifying how many new basis functions the LLM should suggest in the current round. Default value: 5.

**first_round_n_candidates**: Hyperparameter specifying how many new basis functions the LLM should suggest in the first round. Default value: 10.

**dataset_and_problem_description**: Provides the LLM with context about the scientific problem and data, e.g.:

```
Prediction of Treatment Response for Combined Chemo- and Radiation-Therapy
for Non-Small Cell Lung Cancer Patients Using a Bio-Mathematical Model

Here you must model the state differential of **cancer_volume**, and **chemo_concentration**, which are **
    dv_dt** and **dc_dt**, driven by the input actions **chemo_dosage** and **radiotherapy_dosage**,
    together with relevant
laboratory and coagulation markers listed below.

Description of the variables
```

```
* cancer_volume         : Volume of the tumour (cm^3)
* chemo_concentration   : Plasma concentration of vinblastine (mg m^-3)
* chemo_dosage          : Administered vinblastine dose rate (mg m^-3 day^-1)
* radiotherapy_dosage   : Delivered external-beam RT dose rate (Gy day^-1)

Time unit: **days**

Typical value ranges
────────────────────
* cancer_volume         : 0.01433 - 1170.861 cm^3
* chemo_concentration   : 0 - 9.9975 mg m^-3
* chemo_dosage          : 0 - 5 mg m^-3 day^-1
* radiotherapy_dosage   : 0 - 2 Gy day^-1

Dataset summary
───────────────
The training dataset consists of **1000** patients, each
followed for 60 days with daily resolution (delta_t = 1 day). Continuous variables are recorded once per day
    unless otherwise specified.
```

**current_terms**: Informs the LLM about terms already part of the active model in iterative steps, e.g.:

```
['cancer_volume', 'chemo_dosage', 'cancer_volume * chemo_concentration', 'cancer_volume * radiotherapy_dosage
    ', 'np.log(cancer_volume + 1)', 'np.sqrt(cancer_volume)']
```

**current_equation**: Shows the LLM the current best-performing model to potentially build upon or deviate from, e.g.:

```
dv_dt = 0.009194 cancer_volume - 0.0004414 chemo_dosage - 0.02797 cancer_volume * chemo_concentration -
    0.04775 cancer_volume * radiotherapy_dosage - 1.214 np.log(cancer_volume + 1) + 1.202 np.sqrt(
    cancer_volume)
dc_dt = - 0.004518 cancer_volume + 0.9776 chemo_dosage - 0.001191 cancer_volume * chemo_concentration - 3.691e
    -06 cancer_volume * radiotherapy_dosage - 1.442 np.log(cancer_volume + 1) + 0.5209 np.sqrt(cancer_volume
    )
```

**history**: Gives a history of terms decisions, and performance, enabling in-context learning, e.g.

```
Round node_0_0:  KEEP=['cancer_volume', 'cancer_volume * chemo_concentration', 'cancer_volume *
    radiotherapy_dosage', 'np.log(cancer_volume + 1)', 'np.sqrt(cancer_volume)', 'chemo_dosage']  | DROP=['
    chemo_concentration', 'radiotherapy_dosage', 'chemo_dosage * radiotherapy_dosage', 'cancer_volume *
    chemo_dosage', 'cancer_volume * radiotherapy_dosage', 'np.log(cancer_volume + 1)', 'np.sqrt(
    cancer_volume)', 'radiotherapy_dosage']  |  MSE before pruning=0.005453 (per-output
    =[0.010905755578988435, 6.33977252002678e-27])  |  MSE after pruning=0.900026 (per-output
    =[0.011005448429060318, 1.789047395904737])
Round node_0_0_1: ...
```

**Example LLM Response for "Propose" Agent:** Below is an example response from the LLM acting as the "Propose" agent, based on the template above.

Listing 2: Example LLM Response from "Propose" Agent

```
TERMS
```
cancer_volume
chemo_concentration
chemo_dosage
radiotherapy_dosage
np.log(cancer_volume + 1)
np.sqrt(cancer_volume)
chemo_dosage * radiotherapy_dosage
cancer_volume * chemo_concentration
radiotherapy_dosage / (cancer_volume + 1)
chemo_dosage / (chemo_concentration + 1)
```
```

### B.6.2    TERM PRUNING PHASE: THE "PRUNE" AGENT

After new terms are proposed and a model is fitted using the expanded set of basis functions, the "Prune" agent is invoked. This agent receives the full list of candidate terms, their fitted weights $w_j$, their per-term influence scores $\Delta_j$ (e.g., change in validation MSE if the term is removed), and the overall validation MSE. Its task is to decide which terms to keep or discard.

**Prompt Template for "Prune" Agent:** The following template illustrates the prompt given to the "Prune" agent.

## Listing 3: Prompt Template for "Prune" Agent

```
You are an equation-pruning assistant for symbolic regression.

========================================================
INPUT YOU RECEIVE
========================================================

* A table (or dictionary/json representation) where each row has:

| field     | meaning                                                              |
|-----------|----------------------------------------------------------------------|
| term      | Name of the symbolic basis function psi_k(x) (e.g. "x1", "sin(x2)", ...). |
| weight    | Fitted scalar coefficient w_k obtained by ordinary least squares (OLS).  |
| influence | Influence score for term k (see definition below).                   |

* The validation set MSE.

**Influence definition (no refit)**

delta_k is the increase in mean-squared-error (MSE) if the k-th weight is deleted while all other weights stay
      fixed. For OLS this is

   delta_k = (w_k^2 / n) * sum phi_k(x_i)^2  (always >= 0).

* Note that the influence values are computed on the validation (rather than training) set, and thus may not
      always be >= 0.

Hence:
* If influence is large -> the term is important (its removal hurts the loss a lot).
* If influence ~= 0 -> the term is useless (its removal makes no noticeable difference).

========================================================
YOUR TASK
========================================================
1. **Inspect every row**.
2. **Decide "keep" or "drop"** for each term using the rule:

* Use the heuristic: "delta_k ~= 0 -> drop", "large delta_k -> keep" and your own judgement.

3. **Return** a python dictionary after "DECISION" with exactly the two keys

DECISION
```
{{
    "keep":  ["term_a", "term_b", ...],
    "drop":  ["term_c", "term_d", ...]
}}
````

Place each term name in either **keep** or **drop** - never both, never neither.

**IMPORTANT:**
* Make use of the dataset and problem description to make the best decision.
* Make sure to use the learnings from the history of previous rounds.
* (!) You must consider the generalization beyond the validation set and make decisions accordingly.
* (!) You should also consider BOTH the weights and the influence of the terms.
* (!) You MUST keep {keep_n_terms} terms at most, to keep the model interpretable.

________________________________________________
CONVENTIONS & NOTES
________________________________________________

# Important notes
* Treat terms independently; no need to refit or update weights.
* Note that everything was evaluated on the validation set to avoid overfitting.
* If there are multiple outputs (targets), you will see multiple tables, one for each target.
* Use all the information available, but keep in mind that you must only return one keep/drop decision even if
      there are multiple outputs.
* Keep only the most important terms for each output.

# Output format
* Feel free to comment briefly (<= 30 chars) about each decision, but keep the python dictionary in the right
      format.
* The dictionary MUST be provided between triple backticks, otherwise it cannot be parsed.
* It must be prepended with "DECISION", otherwise it cannot be parsed.

________________________________________________
EXAMPLE
________________________________________________

# INPUT TABLE(S):

y_1:
| term   | weight | influence |
| ------ | ------ | --------- |
| x1     | 3.00   | 12.21     |
| x2     | -1.96  | 5.14      |
| x1**2  | 0.53   | 0.91      |
| sin    | 0.93   | 0.52      |
| cos    | -0.05  | 0.0009    |

y_2:
```

```
| term   | weight | influence |
| ------ | ------ | --------- |
| x1     | 3.00   | 12.21     |
| x2     | -1.96  | 5.14      |

MSE (per-output): [0.217, 0.145]
MSE overall: 0.181

# Output:

DECISION
```
{{
    "keep": ["x1", "x2", "x1**2", "sin"],
    "drop": ["cos"]
}}
```

That's it - perform the keep/drop decision based on the information provided.

=======================================================
DATASET AND PROBLEM DESCRIPTION
=======================================================
{dataset_and_problem_description}

=======================================================
CURRENT TERMS
=======================================================
{current_terms}

# Current equation:
{current_equation}

=======================================================
HISTORY
=======================================================
{history}

=======================================================
INPUT YOU RECEIVE
=======================================================

INPUT TABLE(S):

{input}

MSE (per-output): [..., ...]
MSE overall: ...
```

The prompt template for the "Prune" agent is populated with the following key pieces of information to guide the pruning decision:

**keep_n_terms**: Hyperparameter specifying the maximum number of terms to keep. Default value: 6.

**dataset_and_problem_description**: Analogous to the "Propose" agent.

**current_terms**: Analogous to the "Propose" agent.

**current_equation**: Analogous to the "Propose" agent.

**history**: Analogous to the "Propose" agent.

**input**: This is a critical input, formatted as a table, showing each candidate basis function $\psi_j$, its fitted weight $w_j$, and its calculated influence score $\Delta_j$. For example:

```
INPUT TABLE(S):

dv_dt:
| term                             |        weight |     influence |
|:---------------------------------|-------------:|-------------:|
| cancer_volume                    |  0.0092651    |    2.7752     |
| chemo_concentration              | -0.0035792    |    0.000354906 |
| chemo_dosage                     |  0.00120966   |    9.91133e-06 |
| radiotherapy_dosage              |  0.00212704   |   -7.52432e-07 |
| cancer_volume * chemo_concentration | -0.027962  |  638.689      |
| cancer_volume * chemo_dosage     | -1.14907e-05  |    9.16085e-05 |
| cancer_volume * radiotherapy_dosage | -0.0477543 |  197.164      |
| chemo_dosage * radiotherapy_dosage | -0.000750627 |    1.83055e-05 |
| np.log(cancer_volume + 1)        | -1.20511      |   17.425      |
| np.sqrt(cancer_volume)           |  1.19886      |  109.978      |

dc_dt:
| term                             |        weight |     influence |
```

```
|:----------------------------------|------------:|------------:|
| cancer_volume                     |  4.8914e-16 | -5.36642e-27 |
| chemo_concentration               | -0.5        |  6.62719    |
| chemo_dosage                      |  1          | 10.8058     |
| radiotherapy_dosage               | -6.55278e-15 |  4.93846e-28 |
| cancer_volume * chemo_concentration |  1.86065e-17 | -1.98136e-27 |
| cancer_volume * chemo_dosage      | -8.391e-18  |  7.54011e-28 |
| cancer_volume * radiotherapy_dosage | -9.34055e-17 |  3.47305e-27 |
| chemo_dosage * radiotherapy_dosage | -1.90453e-16 |  3.39238e-29 |
| np.log(cancer_volume + 1)         |  5.50181e-16 | -2.14874e-28 |
| np.sqrt(cancer_volume)            | -2.09539e-16 |  2.65579e-28 |

MSE (per-output): [0.010905755578988435, 6.33977252002678e-27]
MSE overall: 0.005453
```

**Example LLM Response for "Prune" Agent:**  An example of an LLM's response when acting as the "Prune" agent is shown below.

### Listing 4: Example LLM Response from "Prune" Agent

```
To make the equation interpretable and preserve only the most meaningful terms, I will review the terms in
    both outputs ('dv_dt' and 'dc_dt') based on their influence and weight. I'll aim to retain at most 6
    terms.

---

### **Analysis for 'dv_dt':**
1. **Highly influential terms**:
    - 'cancer_volume * chemo_concentration' (delta = 638.509) is critical.
    - 'cancer_volume * radiotherapy_dosage' (delta = 197.209) is highly impactful.
    - 'np.sqrt(cancer_volume)' (delta = 110.587) has substantial influence.
    - 'np.log(1 + cancer_volume)' (delta = 17.6803).

2. **Moderately influential / less relevant**:
    - 'cancer_volume' (delta = 2.7081): core variable, but delta suggests moderate importance.
    - 'chemo_dosage', 'radiotherapy_dosage': limited delta values.

3. **Negligible**:
    - Terms involving weighted interactions: e.g., 'chemo_dosage / (1 + cancer_volume)' (delta ~= 0). These
        can be ignored safely.

---

### **Analysis for 'dc_dt':**
1. **Highly influential terms**:
    - 'np.log(1 + cancer_volume)' (delta = 22.3107): critical.
    - 'chemo_dosage' (delta = 18.8115): essential for modeling 'dc_dt'.
    - 'np.sqrt(cancer_volume)' (delta = 15.8203): quite significant.

2. **Moderately influential**:
    - 'radiotherapy_dosage': important as part of the model.
    - 'cancer_volume': relevant, but has relatively small delta compared to other terms.

3. **Negligible terms**:
    - Interactions like 'radiotherapy_dosage / (1 + cancer_volume)' (small delta).

---

Considering both outputs, I aim to balance interpretability and prioritization based on influence across both
    equations. Here is the final decision:

---

DECISION
```
{
    "keep": [
    "cancer_volume * chemo_concentration",  # Very high delta for dv_dt
    "cancer_volume * radiotherapy_dosage",  # Very high delta for dv_dt
    "np.sqrt(cancer_volume)",               # Significant delta for both dv_dt and dc_dt
    "np.log(1 + cancer_volume)",            # Critical delta for both dv_dt and dc_dt
    "chemo_dosage",                         # Crucial delta for dc_dt
    "radiotherapy_dosage"                   # Balanced importance for dc_dt
    ],
    "drop": [
    "cancer_volume",                        # Moderate delta in both outputs
    "chemo_dosage * chemo_concentration",   # Insignificant in dv_dt (<0.0003)
    "radiotherapy_dosage * chemo_concentration",  # Similar to above
    "chemo_dosage / (1 + cancer_volume)",   # Negligible delta
    "radiotherapy_dosage / (1 + cancer_volume)"   # Negligible delta
    ]
}
```
```

## C  BENCHMARK DATASET AND EVALUATION DETAILS

Our evaluation utilizes several benchmark datasets, including three derived from a sophisticated biomedical Pharmacokinetic-Pharmacodynamic (PKPD) model, one from an epidemic simulator, one real-world clinical dataset, and one real-world biological dataset.

### C.1  CANCER PKPD SIMULATIONS

Three of our benchmark environments are based on a well-established biomedical Pharmacokinetic-Pharmacodynamic (PKPD) model of lung cancer tumor growth. This model simulates the combined effects of chemotherapy and radiotherapy and has been utilized in prior research (Geng et al., 2017; Seedat et al., 2022; Bica et al., 2020; Melnychuk et al., 2022; Holt et al., 2024b). We use this model to generate data for three distinct scenarios: **Lung Cancer** (no treatments), **Lung Cancer (with Chemo.)** (chemotherapy only), and **Lung Cancer (with Chemo. & Radio.)** (both chemotherapy and radiotherapy). Each scenario results in a separately sampled dataset. The comprehensive *Lung Cancer (with Chemo. & Radio.)* scenario, based on the general Cancer PKPD model, is described below, followed by how the other variations are derived.

**General Cancer PKPD Model Structure.** This model describes the tumor volume $x(t)$ (in cm$^3$) over time $t$ (in days) following diagnosis. It incorporates the effects of radiotherapy, $u_t^r$, and chemotherapy, $u_t^c$. The tumor dynamics are governed by the differential equation:

$$\frac{dx(t)}{dt} = \left( \rho \log \left( \frac{K}{x(t)} \right) - \beta_c C(t) - (\alpha_r d(t) + \beta_r d(t)^2) \right) x(t)$$

where the first term $\rho \log \left( \frac{K}{x(t)} \right)$ represents tumor growth dynamics, the term $\beta_c C(t)$ models the effect of chemotherapy, and $(\alpha_r d(t) + \beta_r d(t)^2)$ models the effect of radiotherapy. The patient-specific parameters $K, \rho, \beta_c, \alpha_r, \beta_r$ are adopted from Geng et al. (2017), and their values are summarized in Table 6.

Table 6: **Cancer PKPD model parameter values from Geng et al. (2017).**

| Model Component | Variable | Parameter Symbol | Value |
|---|---|---|---|
| Tumor Growth | Growth rate | $\rho$ | $7.00 \times 10^{-5}$ day$^{-1}$ |
| | Carrying capacity | $K$ | 30 cm$^3$ |
| Radiotherapy | Linear cell kill | $\alpha_r$ | 0.0398 Gy$^{-1}$ |
| | Quadratic cell kill | $\beta_r$ | Set such that $\alpha_r/\beta_r = 10$ Gy |
| Chemotherapy | Chemotherapy cell kill | $\beta_c$ | 0.028 (mg/m$^3$)$^{-1}$ day$^{-1}$ |

The concentration of the chemotherapy drug, $C(t)$, is modeled by an exponential decay with a one-day half-life:

$$\frac{dC(t)}{dt} = -0.5C(t)$$

A chemotherapy action $u_t^c$ corresponds to an increase in $C(t)$ by $5.0$ mg/m$^3$ of Vinblastine administered at time $t$. The radiotherapy term $d(t)$ represents the daily dose, with $u_t^r$ corresponding to $2.0$ Gy fractions of radiotherapy delivered at timestep $t$.

**Time-Dependent Confounding.** To introduce realistic complexities, the administration of chemotherapy and radiotherapy is modeled as Bernoulli random variables, with probabilities $p_c(t)$ and $p_r(t)$ dependent on the average tumor diameter $\bar{D}(t)$:

$$p_c(t) = \sigma \left( \frac{\gamma_c}{D_{\max}} (\bar{D}(t) - \delta_c) \right) \qquad p_r(t) = \sigma \left( \frac{\gamma_r}{D_{\max}} (\bar{D}(t) - \delta_r) \right)$$

where $\sigma(\cdot)$ is the sigmoid function, $D_{\max} = 13$ cm is the maximum tumor diameter, $\delta_c = \delta_r = D_{\max}/2$. The parameters $\gamma_c = \gamma_r = 2$ control the strength of this time-varying confounding.

**Dataset Generation.** For each of the three Cancer PKPD scenarios, we sample $N = 1,000$ patient trajectories. Initial tumor volumes $x(0)$ are drawn from a uniform distribution $\mathcal{U}(0, 1149)$ cm$^3$. Patient trajectories are simulated for 60 days using the PKPD model (Appendix C.1) and the specified

action policy, employing a Euler stepwise numerical solver. This procedure generates one dataset sample. We create distinct training, validation, and test sets ($\mathcal{D}_{\text{train}}, \mathcal{D}_{\text{val}}, \mathcal{D}_{\text{test}}$) by repeating this sampling process with different random seeds. For each run of a benchmark method requiring a random seed, these datasets are re-sampled. The **Lung Cancer** scenario omits both treatment terms (effectively $C(t) = 0, d(t) = 0$). The **Lung Cancer (with Chemo.)** scenario omits the radiotherapy term only ($d(t) = 0$). The **Lung Cancer (with Chemo. & Radio.)** scenario uses the full model as described.

## C.2 COVID-19 EPIDEMIC SIMULATION

We employ COVASIM, a detailed agent-based simulator for modeling COVID-19 epidemics (Kerr et al., 2021). COVASIM can simulate various non-pharmaceutical interventions (e.g., lockdowns, social distancing) and pharmaceutical interventions (e.g., vaccinations). In this model, each agent signifies an individual who can transition between states: susceptible, exposed, infectious, or recovered (which includes deaths).

We use COVASIM with its default parameters as provided in its open-source implementation[1]. To generate diverse epidemic trajectories, we simulate 24 distinct "countries" or populations. For each simulation, the population size is set to $1,000,000$ individuals, with each agent simulated individually (i.e., simulation rescaling is disabled). Each simulation starts with an initial number of infected individuals $I(0)$ sampled uniformly from $\mathcal{U}(10,000, 100,000)$, and the epidemic is simulated for 60 days.

This process is repeated with independent random seeds to generate training, validation, and test datasets ($\mathcal{D}_{\text{train}}, \mathcal{D}_{\text{val}}, \mathcal{D}_{\text{test}}$). For each benchmark method run that involves a random seed, these datasets are re-sampled.

## C.3 WARFARIN PHARMACOKINETICS DATASET

We utilize a real-world clinical trial dataset focused on Warfarin pharmacokinetics (PK), which is publicly available (Janssen et al., 2022). This dataset, known as the NOMEN dataset, can be accessed from `https://github.com/Janssena/SI-AIEP-paper`. It comprises data from 32 patients who received a single dose of Warfarin. The dataset contains a total of 251 Warfarin concentration measurements, with a median of six measurements per patient. Warfarin was administered at $t = 0$, and concentration measurements were taken at predefined time points: $t \in \{0.25, 0.5, 1.0, 2.0, 4.0, 6.0, 12.0, 24.0, 48.0, 72.0, 96.0, 120.0\}$ hours. Covariates available for each patient include weight, age, and sex.

We adhere to the original pre-processing scripts provided with the dataset. The data is split into training, validation, and test sets using proportions of 70%, 15%, and 15%, respectively. These splits are performed chronologically to maintain temporal causality. This dataset is released under a GPL-3.0 license.

## C.4 RNA POLYMERASE II PAUSING DATASET

This dataset, central to the case study presented in Section 5.1, focuses on predicting RNA Polymerase II (Pol II) pausing. It is derived from eNET-seq data that maps Pol II pause sites in human cells at single-nucleotide resolution (Fong et al., 2022). The primary objective is to predict the `pause_score`, a continuous variable between 0 and 1, where a higher score indicates increased Pol II pausing at a specific genomic location. The pause score is calculated as the ratio of sequencing reads at the pause site to the total reads in a 200bp window surrounding it.

The dataset comprises 263 features, including:

- **Genomic Coordinates and Context:** Features such as `start` (current position), `gene_start`, `gene_end`, `exon_intron_start`, `exon_intron_end`, distance to nearest downstream (`down_nuc_dist`) and upstream (`up_nuc_dist`) nucleosomes.
- **Chromatin Features:**

---

[1]COVASIM is available at `https://github.com/InstituteforDiseaseModeling/covasim`.

- Nucleosome occupancy signal from MNase-seq (e.g., `SIGNAL_MNase_CONDITION_WT`).
- Histone modification signals from ChIP-seq, such as H3K4me3 (e.g., `SIGNAL_ChIPseqH3K4me3_CONDITION_CDK7negWT`, where higher signal implies more H3K4me3 modification) and H3K36me3 under various conditions (e.g., `SIGNAL_ChIPseqH3K36me3_CONDITION_negU170K`, `SIGNAL_ChIPseqH3K36me3_CONDITION_posU170K`, where higher signal indicates more H3K36me3 modification).
- **RNA Structure Features:** DMS signal (e.g., `SIGNAL_DMS_CONDITION_WT_STRAND_neg`, where lower signal implies more RNA structure) and RNA structure scores (e.g., `SIGNAL_StructureScore_CONDITION_WT_STRAND_neg`, where higher score indicates more structure).
- **DNA Sequence Context:** One-hot encoded nucleotides at positions -20 to +20 relative to the potential pause site (e.g., `seq_neg1__A`, `seq_0__G`, `seq_1__T`). Possible nucleotide categories are `A`, `C`, `G`, `T`, and `N` (unknown).
- **Gene Region Annotations:** Categorical features like `chrom__<category>` (chromosome, e.g., `chr1`, `chrX`), `strand__<category>` (pos or neg), `gene_region__<category>` (one of `TSS`, `body`, `termination`), and `exon_intron__<category>` (one of `exon`, `intron`, `missing`). Categorical columns are one-hot encoded, with column names in the format `<column_name>__<category>`.

The target variable is `pause_score`. The dataset used for the experiments described in Section 5.1 was balanced, containing approximately 48,000 pause sites and a similar number of control sites (where pause score = 0), and was split into training, validation, and test sets. For each benchmark method run involving a random seed, this training/validation/test split was resampled.

## C.5 BENCHMARK METHOD DETAILS

Our comparative evaluation includes several established benchmark methods to contextualize the performance of SGED. These methods span both black-box and white-box modeling paradigms.

### C.5.1 WHITE-BOX NON-LLM BASELINES

**DyNODE** (Dynamical Neural Ordinary Differential Equations) is a method that learns the underlying dynamics of a system from observed data by parameterizing the derivative function of an ordinary differential equation (ODE) with a neural network (Chen et al., 2018). For systems involving external actions or interventions, the DyNODE framework can be extended to incorporate these action inputs directly into the learned dynamics, allowing it to model how treatments or other external factors influence the system's evolution over time (Alvarez et al., 2020). This approach offers a flexible black-box model for continuous-time dynamical systems.

**SINDy** (Sparse Identification of Nonlinear Dynamics) is a white-box method designed to discover governing differential equations directly from time-series data (Brunton et al., 2016). It operates by constructing a library of candidate nonlinear functions of the state variables and then employs sparse regression techniques (typically sequentially thresholded least-squares or Lasso) to identify a minimal set of active terms that best describe the observed dynamics. The result is an interpretable, parsimonious differential equation model.

**GPLearn** (Stephens, 2015) is a genetic programming approach for symbolic regression that algorithmically discovers mathematical expressions to model a given dataset. It works by evolving a population of candidate equations over a number of generations, applying genetic operators such as crossover and mutation to iteratively refine solutions based on a fitness metric, typically related to prediction accuracy. In the experiments, unless stated otherwise, GPLearn was configured with a `population_size` of 1000, run for 30 `generations`, and a `parsimony_coefficient` of 1.0. The `parsimony_coefficient` of 1.0 was selected to apply a considerable penalty to the complexity of the evolved expressions. This encourages the discovery of more concise equations, which aligns with the general aim of finding parsimonious models (e.g., around six terms in the primary SGED experiments), thus making shorter programs preferable during the evolutionary search.

### C.5.2 WHITE-BOX LLM BASELINES

To ensure fair comparison, we set the maximum number of terms allowed to be six (unless otherwise stated) in SGED, and ensured this was comparably set in all other corresponding methods. We use the same underlying LLM across all LLM-based methods (`GPT-4o` version `2024-11-20`, unless otherwise stated).

**ZeroShot** represents a baseline LLM-driven approach where the Large Language Model is prompted to generate a symbolic equation model in a single pass, based solely on the provided problem description and dataset characteristics. This method does not involve any iterative refinement or feedback based on the model's performance on actual data. Consequently, the parameters of the equation generated by the ZeroShot method are used as proposed by the LLM without any subsequent optimization against the training dataset. The number of terms in the equation generated by the LLM was constrained via prompting, typically to a maximum of six terms (unless stated otherwise), to align with the parsimony goals of SGED.

**ZeroOptim** builds directly upon the ZeroShot approach. An LLM first generates an initial symbolic equation model in a zero-shot manner, based on the problem description. However, unlike the pure ZeroShot baseline, the structural form of this LLM-proposed equation is then taken, and its constituent parameters (weights $w_j$) are subsequently optimized by fitting them to the training data. This optimization process is performed analogously to the final model fitting stage within the SGED framework, aiming to find the best parameter values for the LLM's proposed equation structure. As with ZeroShot, the LLM was prompted to generate equations with a limited number of terms, typically up to six (unless stated otherwise), for comparability.

**ICL - Basic Feedback** is an LLM-based iterative equation discovery method that serves as a key ablation and comparator to SGED. In this approach, the LLM iteratively proposes candidate equations (or modifications to existing ones). After each proposal, the equation's parameters are optimized analogously to SGED, and its performance is evaluated, typically yielding a scalar metric such as the overall Mean Squared Error (MSE) on a validation set. This scalar MSE, along with a history of previously attempted equations and their outcomes, is now provided back to the LLM to guide its proposal for the next iteration, facilitating in-context learning. Crucially, ICL - Basic Feedback lacks the granular, per-term influence scores ($\Delta_j$) that are central to SGED. Furthermore, it does not incorporate SGED's dedicated term pruning phase, where an LLM agent refines the equation structure based on these influence scores. For fair comparison with SGED, the maximum number of terms in the equations generated by the LLM was limited via prompting, with a default of six terms in the experiments unless specified otherwise.

**D3-white-box** (Holt et al., 2024b) refers to the Data-Driven Discovery framework when specifically configured to discover interpretable, white-box dynamical system models. This approach leverages Large Language Models (LLMs) within an iterative cycle involving three core agents: Modeling, Feature Acquisition, and Evaluation. For white-box discovery, the Modeling Agent is tasked by the LLM to propose and refine closed-form equation models. These equations are represented as executable Python code, typically PyTorch modules. After a model is proposed, its parameters are optimized against training data. The Evaluation Agent then assesses the model, providing feedback that includes quantitative metrics like validation Mean Squared Error (MSE) and qualitative verbal reflections on model structure and plausibility. This feedback informs the LLM for subsequent iterations of model generation and refinement. The Feature Acquisition Agent can also propose additional relevant features to incorporate, further guiding the discovery process, although for specific benchmark comparisons, it might be applied to a fixed set of predefined features. The overall D3 framework aims to autonomously navigate the model space, identify relevant system variables, and converge on accurate, interpretable equations. The implementation details and hyperparameters as per (Holt et al., 2024b) were used for this benchmark. The maximum number of input features in the white-box model was constrained via prompting, with a default of six, unless specified otherwise.

**LaSR** (Symbolic Regression with a Learned Concept Library, Grayeli et al., 2024) is a framework that enhances traditional genetic algorithms for symbolic regression by incorporating a learned library of abstract textual concepts. It uses an LLM to discover and evolve high-level concepts (e.g., "exponential decay") from successful equations. These concepts then guide the mutation and crossover operations in the search, biasing it towards scientifically plausible structures rather than refining the equations directly. The complexity hyperparameter `maxsize` of 30 was used to allow

for equations at least as complex as the six-term limit used elsewhere. The total number of cycles was set to 75. Other hyperparameters used were as per Grayeli et al. (2024).

**LLM-SR** (Scientific Equation Discovery via Programming with Large Language Models, Shojaee et al., 2025a) frames equation discovery as program synthesis. It leverages an LLM's scientific prior knowledge and code generation capabilities to propose equation skeletons as Python programs. These programs are then combined with an evolutionary search, where data-driven feedback on the overall program's fit is used to guide the iterative refinement process. The complexity hyperparameter `max_nparams` was set to 8, in excess of SGED's six term limit. The rest of the hyperparameters were used as per Shojaee et al. (2025a).

**ICSR** (In-Context Symbolic Regression, Merler et al., 2024) employs an LLM as an optimizer within an iterative refinement loop. The LLM is prompted with a set of previously evaluated equations and their corresponding scalar fitness scores (which typically combine accuracy and a complexity penalty). Through in-context learning, the LLM is tasked with generating a new candidate equation that is expected to achieve a better score, effectively navigating the solution space based on a history of successes and failures. The hyperparameter from Merler et al. (2024) were used, and `max_nodes` complexity parameter was set to 30 to allow for equations at least as complex as the six-term limit used in SGED and elsewhere.

### C.5.3 BLACK-BOX BASELINES

**RNN** (Recurrent Neural Network) models are a class of neural networks well-suited for sequential data, including time-series (Rumelhart et al., 1986). Standard RNN architectures, such as those using LSTM (Long Short-Term Memory) (Hochreiter & Schmidhuber, 1997) or GRU (Gated Recurrent Unit) (Cho et al., 2014) cells, maintain an internal hidden state that captures information from past inputs, enabling them to model temporal dependencies. For prediction tasks, the RNN processes input sequences (e.g., trajectories of state variables and actions) to predict future states or system outputs. These are generally considered black-box models due to the complexity of their internal representations.

**Transformer** models, originally introduced for natural language processing tasks (Vaswani et al., 2017), have demonstrated strong performance on a wide variety of sequential data, including time-series. The core mechanism of Transformers is the attention mechanism, particularly self-attention, which allows the model to weigh the importance of different elements in the input sequence when making predictions. This enables them to capture long-range dependencies effectively. Like RNNs, Transformer models are typically considered black-box due to their intricate architectures and large number of parameters.

### C.6 EVALUATION DETAILS

We assess the performance of all benchmark methods using the mean squared error (MSE) on a held-out test dataset, denoted as $\mathcal{D}_{\text{test}}$. The MSE is calculated based on the model's predictions against the true target values in this test set. Where there are multiple target variables, the mean MSE across all targets is used.

For each method, given per-seed test MSEs $x_1, \ldots, x_n$, we report two-sided 95% confidence intervals for the mean across seeds as $\bar{x} \pm t_{0.975,n-1} \frac{s}{\sqrt{n}}$, where $\bar{x}$ is the sample mean, $s$ is the unbiased sample standard deviation, and $t_{0.975,n-1}$ is the Student's $t$ critical value (see, e.g., Wasserman, 2004). Replicates are independent training runs (random seeds). We use the appropriate degrees of freedom for each method's $n$.

When a simulator is used to generate data (e.g., for the Cancer PKPD and COVID-19 benchmarks as described in Appendix C), new training, validation, and test datasets ($\mathcal{D}_{\text{train}}, \mathcal{D}_{\text{val}}, \mathcal{D}_{\text{test}}$) are independently generated for each random seed. Unless specified otherwise, the validation and test datasets are generated to contain a comparable number of samples or trajectories as the training set.

Each benchmark model is trained using its respective training dataset ($\mathcal{D}_{\text{train}}$). For methods that support it, early stopping (at patience 10) or default model selection heuristics are applied using the validation dataset ($\mathcal{D}_{\text{val}}$) to prevent overfitting and guide the learning process. For SGED, the validation set is critically used for calculating per-term influence scores and guiding the pruning

decisions, as detailed in Section 2. Maximum number of iterations or generations is set to 30 for all methods where this is applicable unless stated otherwise; SGED maximum node expansion budget for MCST is similarly set set to 30. For a complementary experiment comparing methods at a fixed computational (LLM token) budget, see Appendix H.3. The final reported performance for all methods is evaluated on the unseen test dataset ($\mathcal{D}_{\text{test}}$). This entire procedure of data generation (if applicable), training, validation-guided refinement (if applicable), and testing is repeated for each random seed to ensure robust and reliable comparisons. We use 25 random seeds for SGED results and other LLM-based method results, using the same underlying LLM. Given the evaluation is exactly the same as the baseline and datasets for some results of the baselines, we use previous values averaged over 10 random seeds for the non-LLM baselines from paper Holt et al. (2024b).

# D  RNA POLYMERASE II PAUSING CASE STUDY – FURTHER DISCUSSION

In this section, we provide SHAP plots for the case study experiment (referred to as **Experiment 1** here), an additional experiment (referred to as **Experiment 2** here) conducted to further investigate the use of SGED with the RNA Polymerase II pausing problem, and a discussion given both sets of results. The aim is to illustrate how SGED can be used in scientific discovery and hypothesis generation process (e.g. note the extension to Experiment 2 from Experiment 1, given the results of the former). **Experiment 3** follows Experiment 2's problem setup, but uses additional biological signal features – resulting in a novel hypothesis regarding DNA methylation impact on pausing, subsequently confirmed in a wet-lab experiment, Appendix D.3. This case study is one of many potential use cases for SGED, illustrating the potential power of this method.

**Configuration Note:** The case study Experiments 1-3 were performed with default SGED hyperparameters except: `keep_n_terms=15` and LLM backbone of `o3` (version `2025-04-16`).

## D.1  EXPERIMENT 1 SHAP PLOTS

To further interrogate the interpretable equation discovered by SGED for RNA Polymerase II pausing (presented in Section 5.1), $f(\mathbf{x}) = \sum_j w_j \psi_j(\mathbf{x})$, we employed SHapley Additive exPlanations (SHAP) (Lundberg & Lee, 2017). SHAP is a game theoretic approach that explains the output of a model by assigning an importance value (SHAP value) to each of its input features for every individual prediction. In this application, the "features" provided to SHAP are the evaluated basis functions $\psi_j(\mathbf{x})$ that form the terms of the SGED equation. The SHAP values therefore quantify how much each term $w_j \psi_j(\mathbf{x})$ contributes to pushing the model's output (the pause score) from its base (average) prediction to the actual predicted value for a given sample. This analysis allows us to visualize the magnitude, variability, and directional impact of each constituent term in the discovered equation.

While the SGED equation $f(\mathbf{x}) = \sum_j w_j \psi_j(\mathbf{x})$ is already structured for interpretability as a weighted sum of basis functions, SHAP analysis provides a standardized framework to:

- Visualize the distribution of contributions for each basis function term $\psi_j(\mathbf{x})$ across all samples.
- Understand how the specific value of a basis function $\psi_j(\mathbf{x})$ (which itself is derived from the original input features like $\text{signal}_{\text{MNase}}$ or sequence indicators) influences its contribution to the final pause score.
- Confirm the relative importance and consistent impact of terms that were selected and weighted during the SGED discovery process.

Essentially, SHAP helps to decompose the prediction into contributions from each $\psi_j(\mathbf{x})$ term, providing insights at both a global (overall term importance) and local (individual prediction explanation) level.

We generated two types of SHAP plots to visualize these explanations, as shown in Figure 3: a beeswarm plot and a custom term influence bar plot.

The **beeswarm plot** (Figure 3a) summarizes the SHAP values for the most influential basis function terms $\psi_j(\mathbf{x})$ in the equation. Each point on the plot represents a single term's SHAP value for a specific sample (a potential pause site). The terms are ranked along the y-axis by their global importance (sum of absolute SHAP values across all samples). The x-axis shows the SHAP value,

indicating the term's impact on the model output; positive SHAP values contribute to a higher predicted pause score, while negative values contribute to a lower score. The color of each point corresponds to the *value of the basis function $\psi_j(\mathbf{x})$* for that sample (typically, red indicates high values of $\psi_j(\mathbf{x})$ and blue indicates low values). This coloring reveals how the magnitude of each basis function's evaluation influences its contribution to the pause score.

The **custom term influence bar plot** (Figure 3b) provides an alternative view of global term importance and the general direction of each term's influence. The length of each bar corresponds to the mean absolute SHAP value for that basis function term $\psi_j(\mathbf{x})$, signifying its overall importance in the equation. The color and direction of the bar indicate the predominant direction of the term's influence on the pause score. This directionality is determined by calculating the Spearman's rank correlation coefficient ($\rho$) between the *values of the basis function $\psi_j(\mathbf{x})$* for each sample and their corresponding SHAP values for that term. Spearman's correlation is employed here because:

- It assesses monotonic relationships. The SHAP value of a term $w_j\psi_j(\mathbf{x})$ will have a monotonic relationship with $\psi_j(\mathbf{x})$ (the direction determined by the sign of $w_j$). Spearman captures this robustly.
- It is robust to outliers in the evaluated basis function values or the SHAP values.
- It operates on the ranks of the data, suitable even if distributions are non-normal.

A positive Spearman correlation (typically shown in red, extending to the right) suggests that as the value of the basis function $\psi_j(\mathbf{x})$ increases, its SHAP value (and thus its effective contribution $w_j\psi_j(\mathbf{x})$ relative to its mean) tends to push the predicted pause score higher. Conversely, a negative correlation (typically shown in blue, extending to the left) suggests that an increasing value of $\psi_j(\mathbf{x})$ tends to push the pause score lower. This plot helps to quickly identify which terms in the SGED-discovered equation are most impactful and whether higher values of these terms generally promote or inhibit Pol II pausing.

### D.2 EXPERIMENT 2 AND DISCUSSION

**Experimental Setup.** Experiment 1 (discussed in Section 5.1) tasked SGED with predicting pause sites from an artificially balanced dataset comprising equal numbers of actual pause sites and control non-pause sites within genes. While this approach identified key features distinguishing pause sites from non-pause regions, a limitation is that pause sites are naturally much less frequent than non-pause sites. Furthermore, this setup primarily addresses the *presence* of a pause rather than the *strength* or characteristics of different pause sites relative to each other.

To delve deeper into the factors modulating the intensity of Pol II pausing, Experiment 2 adopted a different approach. Here, SGED was tasked with identifying terms that distinguish pause sites based on their varying pause scores, using a dataset consisting exclusively of identified pause sites (i.e., non-pause sites were excluded). This focuses the analysis on understanding what makes some pauses stronger or weaker than others, given that a pause event is already occurring. This also addresses any potential confounding effects of artifical balancing between pause and non-pause sites in Experiment 1.

**Results.** The equation discovered by SGED in this context is presented below.

$$
\begin{aligned}
\text{pause\_score} = \ & -0.01711 \ln(1 + \text{signal}_{\text{MNase}}) \\
& - 0.0003407 \, \text{signal}_{\text{H3K4me3}} \\
& - 0.00216 \ln(2 + \text{signal}_{\text{H3K4me3}}) \\
& - 0.0005086 \sum_{\text{cond}} \text{signal}_{\text{H3K36me3, cond}} \\
& + 0.01206 \ln(1 + \text{gene\_length}) \\
& - 0.08414 \, \mathbb{1}_{\{\text{gene\_region}=\text{TSS}\}} \\
& + 0.0003421 \, \mathbb{1}_{\{\text{gene\_region}=\text{TSS}\}} \cdot \text{signal}_{\text{H3K4me3}} \\
& + 0.02143 \, \mathbb{1}_{\{\text{seq}_0=G\}} \\
& + 0.0006817 \ln(1 + \text{down\_nuc\_dist}) \\
& + 0.01968 \ln(1 + \text{down\_nuc\_dist} + |\text{up\_nuc\_dist}|) \\
& + 0.0001679 \left( \text{signal}_{\text{H3K4me3}} - \sum_{\text{cond}} \text{signal}_{\text{H3K36me3, cond}} \right)
\end{aligned}
$$

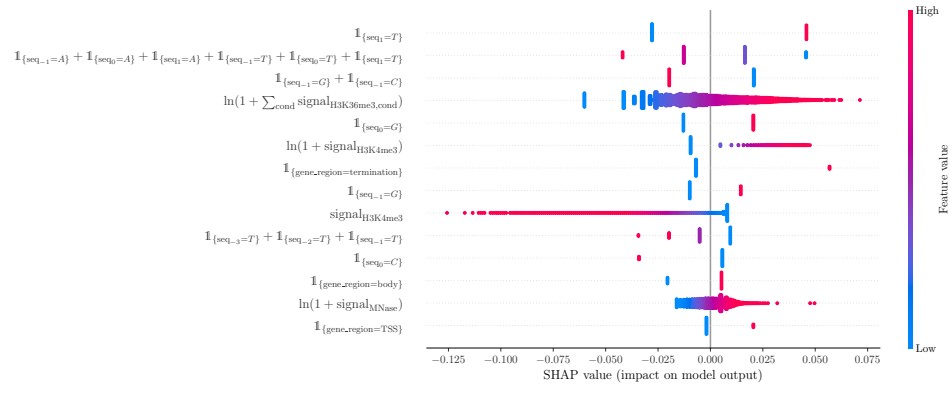

(a) SHAP Beeswarm Plot for Equation Terms

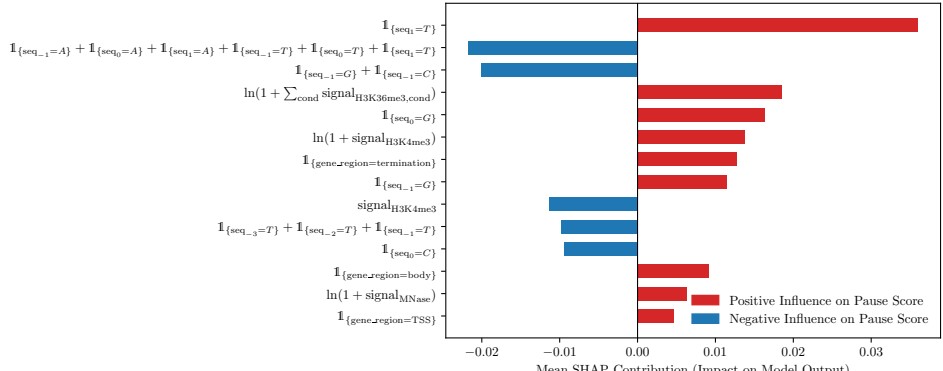

(b) SHAP Term Influence Bar Plot

Figure 3: SHAP analysis of the SGED-discovered equation for RNA Polymerase II pausing from **Experiment 1**. Each "feature" in the SHAP analysis corresponds to a basis function term $\psi_j(\mathbf{x})$ from the equation. (**a**) Beeswarm plot showing the distribution of SHAP values for the most important equation terms. Each point is a SHAP value for a term and a sample, colored by the value of the evaluated basis function $\psi_j(\mathbf{x})$ (red for high, blue for low). (**b**) Custom bar plot illustrating global term importance (mean absolute SHAP value) and the direction of influence. Red bars indicate a general positive correlation between the value of $\psi_j(\mathbf{x})$ and its contribution to the pause score (Spearman's $\rho > 0$), while blue bars indicate a negative correlation (Spearman's $\rho < 0$).

$$- 0.01338 \left( \ln(2 + \text{signal}_{\text{H3K4me3}}) - \ln\left( 2 + \sum_{\text{cond}} \text{signal}_{\text{H3K36me3, cond}} \right) \right)$$
$$+ 0.009823 \, \mathbb{1}_{\{\text{seq}_{-1}=G\}}$$
$$+ 0.01422 \left( \mathbb{1}_{\{\text{seq}_0=G\}} \cdot \mathbb{1}_{\{\text{seq}_1=T\}} \right)$$
$$- 0.01761 \left( \mathbb{1}_{\{\text{seq}_{-1}=G\}} + \mathbb{1}_{\{\text{seq}_{-1}=C\}} + \mathbb{1}_{\{\text{seq}_0=G\}} + \mathbb{1}_{\{\text{seq}_0=C\}} \right.$$
$$\left. + \mathbb{1}_{\{\text{seq}_1=G\}} + \mathbb{1}_{\{\text{seq}_1=C\}} \right),$$

The SHAP analysis for this equation (Figure 4) illustrates the relative effects of these different terms on the predicted pause score.

**Discussion.** Combining insights from both Experiment 1 and Experiment 2, SGED has revealed several features of potential functional importance to transcriptional pausing in human cells. A notable result emerging from this comparative analysis is that high H3K4me3 signal is generally associated with reduced pausing (i.e., lower pause scores in Experiment 2), consistent with a proposed a link between this mark and faster transcription (Wang et al., 2023). Regarding H3K36me3, Experiment 1 indicated its role as a significant predictor for the *presence* of pause sites. However, Experiment 2 suggests that higher levels of H3K36me3 are negatively related to the *strength* of these pause sites

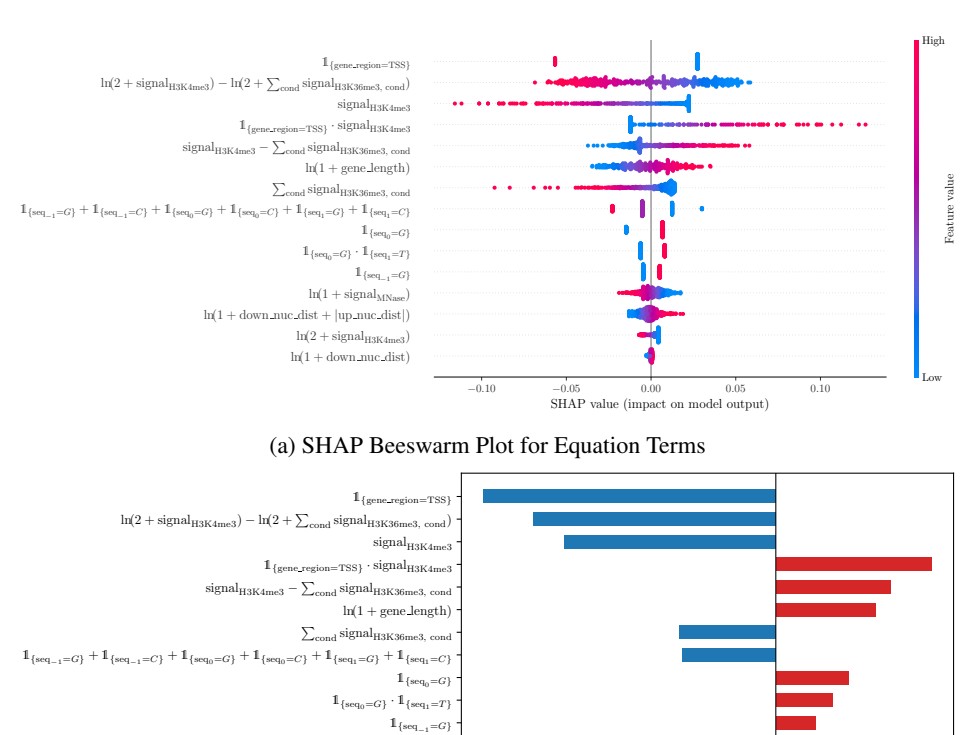

(a) SHAP Beeswarm Plot for Equation Terms

(b) SHAP Term Influence Bar Plot

Figure 4: SHAP analysis of the SGED-discovered equation for RNA Polymerase II pausing from **Experiment 2**. Analogous to Figure 3, but for **Experiment 2**.

(lower pause scores), a finding that aligns with some genetic studies (Lee et al., 2024; Wen et al., 2014). Furthermore, SGED's discovered equations highlight the importance of the relative levels of these two histone marks. Specifically, terms reflecting a high ratio of H3K36me3 to H3K4me3 (e.g., through difference terms like $\mathrm{signal}_{\mathrm{H3K4me3}} - \sum \mathrm{signal}_{\mathrm{H3K36me3, cond}}$ with a positive coefficient, or ratio-like terms in logarithmic form) are associated with higher pause scores, whereas a high H3K4me3 to H3K36me3 ratio appears linked to lower pause scores.

The analysis also found that the region downstream of genes (termination region) is associated with higher pausing scores relative to the gene body and the 5' end (TSS). This is consistent with the hypothesis that Pol II pausing plays a role in facilitating transcription termination (Gromak et al., 2006).

This work also refined our understanding of sequence specificity in pausing. Previous studies identified a G, T/C sequence element as enriched at the pause site (position 0) and the +1 position (Fong et al., 2022; Gajos et al., 2021). Our SGED models confirm that T at position +1 (often as part of a $\mathrm{G}_0-\mathrm{T}_{+1}$ motif) is a strong positive predictor for pause sites. Conversely, the models from both experiments, particularly when examining factors that disfavor pausing or reduce pause strength, identified new elements: C at position 0 and -1, and T at positions -1, -2, and -3 appear to disfavor pausing. Additionally, a cluster of G/C nucleotides at positions -1, 0, and +1 (captured by terms like $-(\mathbb{1}_{\{\mathrm{seq}_{-1}=G\}} + \cdots + \mathbb{1}_{\{\mathrm{seq}_1=C\}}))$ was also found to be associated with reduced pausing or lower pause scores.

These results, derived from the interpretable equations generated by SGED, provide new, data-driven hypotheses that will help guide future experimental studies into the complex mechanisms of sequence-dependent transcriptional pausing and its regulation by the chromatin environment.

### D.3 DISCOVERY VALIDATION: WET-LAB CONFIRMATION OF SGED HYPOTHESES

To rigorously assess the capability of SGED to generate novel, physically verifiable scientific knowledge – rather than merely fitting hold-out data – we conducted a testable study. This involved running a new experiment using SGED, analyzing the resulting interpretable equation to formulate a biological hypothesis, and subsequently verifying that hypothesis through a targeted "wet-lab" experiment involving cell treatment and sequencing.

#### D.3.1 EXPERIMENTAL SETUP: EXPERIMENT 3

We applied SGED to an extended version of the RNA Polymerase II pausing dataset. Like *Experiment 2*, this experiment (*Experiment 3*) focused on modeling the intensity of pausing (pause score) at pause sites, but unlike previous experiments, we included additional signal features, specifically `H3K27ac` (an acetylation mark) and `hg38LiftOver` (a mappability signal often correlated with sequence complexity and *methylation status*).

**The Discovered Equation:** SGED converged on the following model. Notably, the system incorporated the new signals (`H3K27ac` and `hg38LiftOver`) while discarding previous markers like `H3K4me3`, yet achieving a superior MSE of 0.0102 (compared to 0.0111 in prior experiments).

$$
\begin{aligned}
\text{pause\_score} = {}& 0.1430 \quad \text{(constant baseline term)} \\
& + 0.0878 \, \text{signal}_{\text{hg38LiftOver}} - 0.2199 \, \ln\!\left(1 + \text{signal}_{\text{hg38LiftOver}}\right) \\
& + 0.0103 \, \ln\!\left(1 + \text{down\_nuc} - \text{up\_nuc}\right) \\
& - 0.0745 \, \ln\!\left(1 + \text{signal}_{\text{MNase}}\right) + 0.0126 \\
& - 0.0923 \, \mathbb{1}_{\{\text{gene\_region}=\text{TSS}\}} + 0.0342 \, \mathbb{1}_{\{\text{gene\_region}=\text{TSS}\}} \cdot \ln\!\left(1 + \text{signal}_{\text{MNase}}\right) \\
& - 0.0732 \, \ln\!\left(1 + \text{signal}_{\text{H3K27ac}}\right) \\
& + 0.0172 \, \mathbb{1}_{\{\text{gene\_region}=\text{TSS}\}} \cdot \text{signal}_{\text{H3K27ac}} \\
& + 0.0474 \, \mathbb{1}_{\{\text{exon}\}} \cdot \ln\!\left(1 + \text{signal}_{\text{H3K27ac}}\right) \\
& + 0.0274 \, \mathbb{1}_{\{\text{seq}_0=G\}} - 0.0143 \sum_{pos \in \{-1,0,1\}} \left(\mathbb{1}_{\{\text{seq}_{pos}=G\}} + \mathbb{1}_{\{\text{seq}_{pos}=C\}}\right) \\
& + 0.0112 \, \ln\!\left(1 + \text{gene\_length}\right)
\end{aligned}
$$

#### D.3.2 FROM EQUATION TO HYPOTHESIS

Analysis of the SGED-discovered equation pointed to a significant role for DNA methylation. Specifically, SGED predicted that DNA CpG methylation terms (inferred via the `LiftOver` mappability signal and sequence contexts) accounted for approximately 50% of the explanatory power for pause site strength in this context.

Based on the signs of the coefficients, SGED suggests the novel hypothesis: **DNA methylation suppresses pausing within gene bodies.**

While methylation of promoter sequences is well known to suppress transcription initiation (Deaton & Bird, 2011), far less is known about the function of DNA methylation further downstream within gene bodies. Conflicting reports suggest that DNA methylation of the gene body can either suppress (Lorincz et al., 2004) or enhance gene expression (Yang et al., 2014). How transcription elongation is specifically affected by DNA methylation remains understudied. Based on genome-wide data, SGED suggests the novel hypothesis that \*\*DNA methylation suppresses pausing\*\*, a connection that has not previously received significant attention.

#### D.3.3 WET-LAB EXPERIMENTAL VERIFICATION

**Methodology:** We tested this hypothesis by treating HCT116 human colorectal carcinoma cells with 5-azacytidine (1 mM), a potent DNA hypomethylating agent approved for the treatment of certain

leukemias (Thoms et al., 2025). Cells were treated for 3 days followed by 2 days of recovery to deplete DNA methylation levels. We then examined transcriptional pausing using eNET-seq (two biological replicates).

**Results:** The experimental results revealed a **significant increase** in both the frequency and strength of RNA Polymerase II pauses throughout the length of transcribed genes in the treated (hypomethylated) cells compared to controls. This aligns perfectly with the SGED prediction: if methylation suppresses pausing (as predicted), removing methylation should increase it.

---

**Validation Conclusion**

**Hypothesis Confirmed:**
- **Prediction:** SGED predicted that methylation suppresses pausing (negative correlation in the discovered equation).
- **Observation:** Removing methylation (hypomethylation via 5-azacytidine) caused pausing to *increase*.
- **Conclusion:** This confirms the SGED-derived hypothesis that CpG methylation in gene bodies functions as a suppressor of pausing, thereby enhancing gene expression at the level of transcription elongation.

---

This result suggests a **previously unsuspected function for CpG methylation in gene bodies**. In summary, SGED provides a potent example of how an AI agent can collaborate with biologists to generate entirely original hypotheses that are experimentally verifiable.

# E  ADDITIONAL RESULTS

## E.1  IMPACT OF INFLUENCE FEEDBACK WITH MCTS

To further delineate the benefits of the per-term influence feedback mechanism, we conducted an additional comparative analysis focusing on scenarios where Monte Carlo Tree Search (MCTS) is employed as the overarching search strategy. In this setup, we compared SGED equipped with its full influence-based feedback against a variant that also uses MCTS but relies only on basic Mean Squared Error (MSE) for its pruning decisions (akin to the "MCTS + Basic Feedback" ablation in Table 4).

This comparison is particularly insightful as it highlights the value of granular feedback even when a powerful search algorithm like MCTS is exploring the equation space. While MCTS inherently provides a robust exploration framework, the detailed influence scores offer more directed guidance to the LLM agent during the pruning phase within each node expansion of the MCTS. This allows the search to more rapidly identify and prioritize promising branches in the search tree.

Figure 5 illustrates the expected convergence behavior in this MCTS context. The x-axis represents the number of nodes expanded in the MCTS, providing a measure of computational effort in the search process. As shown, SGED with full influence-based feedback converges more quickly to lower MSE values than the MCTS variant relying solely on basic feedback. This accelerated convergence underscores the efficiency gains afforded by providing the LLM with detailed, interpretable information about the contributions of individual equation components, even within a sophisticated tree search paradigm.

Both Figure 2 and Figure 5 are based on 25 seed runs for the Lung Cancer (with Chemo. & Radio.) dataset. In both figures, the best test MSE achieved so far as of # iteration or # node expansion is tracked.

## E.2  ILLUSTRATION OF SGED EQUATION DISCOVERY

To provide a more concrete understanding of how SGED discovers and refines equations, this section illustrates the process using two example runs (Run A and Run B) on the Lung Cancer (with Chemo. & Radio.) benchmark dataset. Figure 6 depicts the progression of the best Mean Squared Error (MSE) achieved on the test set as the Monte Carlo Tree Search (MCTS) expands more nodes. Each

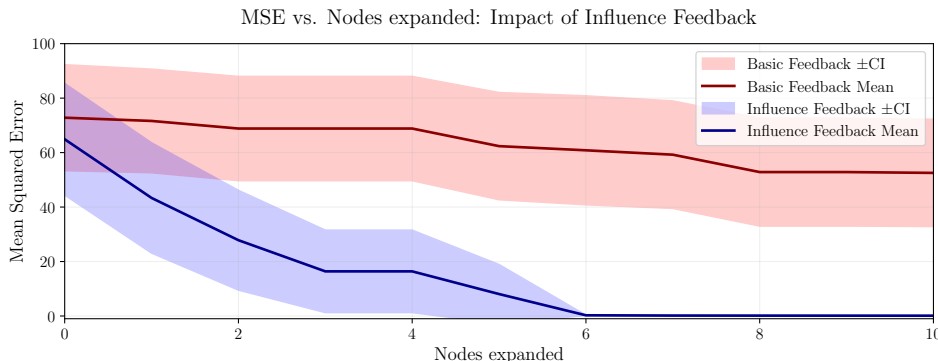

Figure 5: **MSE vs. Nodes Expanded (MCTS): Impact of Influence Feedback.** Validation MSE convergence as a function of MCTS nodes expanded. Compares SGED with full influence-based feedback against an MCTS variant using only basic MSE feedback on the Lung Cancer (with Chemo. & Radio.) dataset. Detailed influence feedback leads to faster convergence (fewer nodes expanded to reach a given MSE) and a lower final MSE within a given computational budget.
Shaded regions denote 95% confidence intervals, and solid lines show means. For equivalent comparison: both variants start with a candidate pool of 6 terms and 6 terms are kept after each propose-and-prune iteration; at the root node performance before pruning is shown to ensure equivalent starting point.

significant drop in MSE, indicating the discovery of a more accurate equation, is annotated on the plot (e.g., A0, B9). Below, we present the specific equations and their corresponding MSEs at these key points, showcasing the iterative improvements made by SGED. This visualization highlights how the framework navigates the equation space, modifying and selecting terms to progressively enhance model performance.

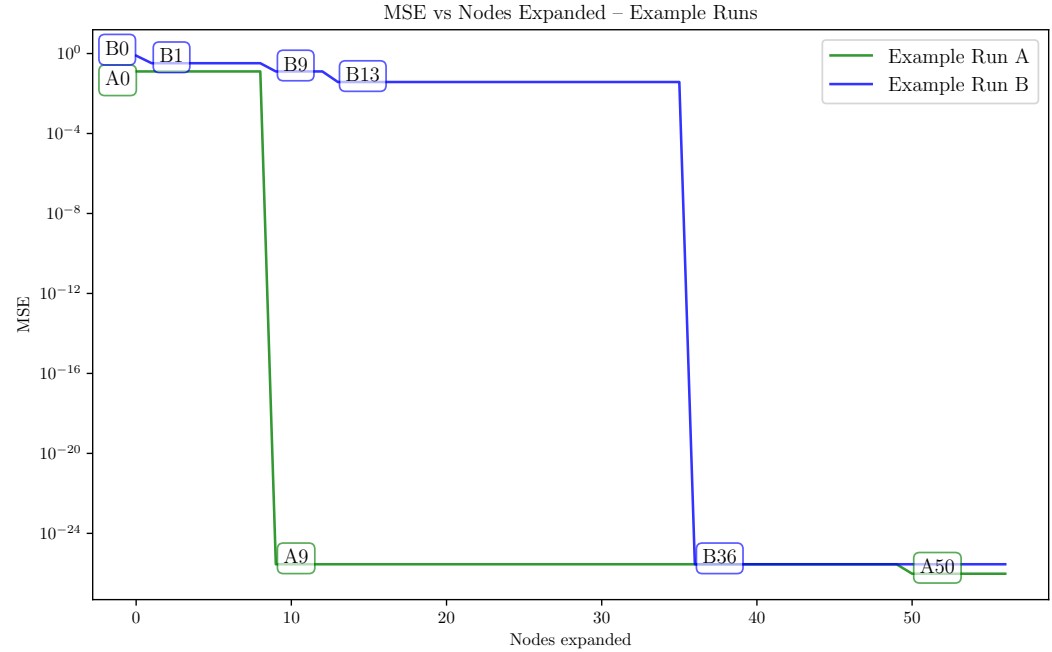

Figure 6: **Illustration of SGED Equation Discovery Process.** Test MSE achieved versus MCTS nodes expanded for two example SGED runs (Run A: green, Run B: blue) on the Lung Cancer (with Chemo. & Radio.) dataset. Points A0, A9, A50 for Run A, and B0, B1, B9, B13, B36 for Run B, indicate instances where a new, more accurate equation was discovered. The equations corresponding to these points are detailed below.

The equations discovered at pivotal moments during these SGED runs are as follows:

EXAMPLE RUN A

- **Point A0** (Node 0):
    - MSE: 0.127
    - Equations:

$$dv\_dt = 0.02425 \cdot \text{cancer\_volume} - 0.08865 \cdot \text{chemo\_concentration}$$
$$- 0.03812 \cdot \text{chemo\_dosage}$$
$$- 0.02755 \cdot \text{cancer\_volume} \cdot \text{chemo\_concentration}$$
$$- 0.04786 \cdot \text{cancer\_volume} \cdot \text{radiotherapy\_dosage}$$
$$+ 0.4913\sqrt{\text{cancer\_volume}}$$
$$dc\_dt = -0.5 \cdot \text{chemo\_concentration} + \text{chemo\_dosage}$$

- **Point A9** (Node 9):
    - MSE: $2.85 \times 10^{-26}$
    - Equations:

$$dv\_dt = 0.1389 \cdot \text{cancer\_volume}$$
$$- 0.028 \cdot \text{cancer\_volume} \cdot \text{chemo\_concentration}$$
$$- 0.04776 \cdot \text{cancer\_volume} \cdot \text{radiotherapy\_dosage}$$
$$- 0.01453 \cdot \text{cancer\_volume} \cdot \ln(\text{cancer\_volume})$$
$$dc\_dt = -0.5 \cdot \text{chemo\_concentration} + \text{chemo\_dosage}$$

- **Point A50** (Node 50):
    - MSE: $9.59 \times 10^{-27}$
    - Equations:

$$dv\_dt = 0.1389 \cdot \text{cancer\_volume}$$
$$- 0.028 \cdot \text{cancer\_volume} \cdot \text{chemo\_concentration}$$
$$- 0.01453 \cdot \text{cancer\_volume} \cdot \ln(\text{cancer\_volume})$$
$$- 0.02388 \cdot \text{cancer\_volume} \cdot (\text{radiotherapy\_dosage})^2$$
$$dc\_dt = -0.5 \cdot \text{chemo\_concentration} + \text{chemo\_dosage}$$

EXAMPLE RUN B

- **Point B0** (Node 0):
    - MSE: 0.785
    - Equations:

$$dv\_dt = 0.04326 \cdot \text{cancer\_volume} + 0.1558 \cdot \text{chemo\_concentration}$$
$$+ 0.1157 \cdot \text{chemo\_dosage}$$
$$- 0.02769 \cdot \text{cancer\_volume} \cdot \text{chemo\_concentration}$$
$$- 0.04836 \cdot \text{cancer\_volume} \cdot \text{radiotherapy\_dosage}$$
$$dc\_dt = -0.5 \cdot \text{chemo\_concentration} + \text{chemo\_dosage}$$

- **Point B1** (Node 1):
    - MSE: 0.328
    - Equations:

$$dv\_dt = 0.03643 \cdot \text{cancer\_volume} - 0.08611 \cdot \text{chemo\_concentration}$$
$$- 0.02803 \cdot \text{chemo\_dosage}$$
$$- 0.02736 \cdot \text{cancer\_volume} \cdot \text{chemo\_concentration}$$
$$- 0.04801 \cdot \text{cancer\_volume} \cdot \text{radiotherapy\_dosage}$$
$$+ 0.6666 \ln(\text{cancer\_volume} + 1)$$
$$dc\_dt = -0.5 \cdot \text{chemo\_concentration} + \text{chemo\_dosage}$$

- **Point B9** (Node 9):
    - MSE: 0.127
    - Equations:

$$dv\_dt = 0.02425 \cdot \text{cancer\_volume} - 0.08865 \cdot \text{chemo\_concentration}$$
$$- 0.03812 \cdot \text{chemo\_dosage}$$
$$- 0.02755 \cdot \text{cancer\_volume} \cdot \text{chemo\_concentration}$$
$$- 0.04786 \cdot \text{cancer\_volume} \cdot \text{radiotherapy\_dosage}$$
$$+ 0.4913\sqrt{\text{cancer\_volume}}$$
$$dc\_dt = -0.5 \cdot \text{chemo\_concentration} + \text{chemo\_dosage}$$

- **Point B13** (Node 13):
    - MSE: 0.0376

– Equations:

$$dv\_dt = 0.005864 \cdot \text{chemo\_concentration} + 0.007118 \cdot \text{chemo\_dosage}$$
$$- 0.02803 \cdot \text{cancer\_volume} \cdot \text{chemo\_concentration}$$
$$- 0.04733 \cdot \text{cancer\_volume} \cdot \text{radiotherapy\_dosage}$$
$$- 1.708 \ln(\text{cancer\_volume} + 1)$$
$$+ 1.544 \sqrt{\text{cancer\_volume}}$$
$$dc\_dt = -0.5 \cdot \text{chemo\_concentration} + \text{chemo\_dosage}$$

- **Point B36** (Node 36):
  - MSE: $2.85 \times 10^{-26}$
  - Equations:

$$dv\_dt = 0.1389 \cdot \text{cancer\_volume}$$
$$- 0.028 \cdot \text{cancer\_volume} \cdot \text{chemo\_concentration}$$
$$- 0.04776 \cdot \text{cancer\_volume} \cdot \text{radiotherapy\_dosage}$$
$$- 0.01453 \cdot \text{cancer\_volume} \cdot \ln(\text{cancer\_volume})$$
$$dc\_dt = -0.5 \cdot \text{chemo\_concentration} + \text{chemo\_dosage}$$

### E.3 INVESTIGATION OF MCTS ROLLOUT DEPTH

The Monte Carlo Tree Search (MCTS) component of SGED plays a crucial role in systematically exploring the vast space of possible equations. A key parameter within MCTS is the rollout depth, which determines how far into the future the search simulates potential sequences of actions (in our case, "propose then prune terms" equation refinement steps) to estimate the value of a given state (equation). Deeper rollouts can provide more accurate value estimates, potentially leading to better search decisions and ultimately more accurate discovered equations. However, they also incur a higher computational cost. This investigation aims to quantify the impact of varying MCTS rollout depths on the performance of SGED, helping to understand this trade-off.

We evaluated the effect of rollout depth on two distinct benchmark datasets: the simulated **Lung Cancer (with Chemo. & Radio.)** dataset and the real-world **RNA Polymerase** dataset. Three MCTS configurations were tested:

- **No Rollout (Heuristic MCTS)**: In this configuration, the MCTS does not perform any simulation. Instead, the value of a newly expanded node (representing an equation) is directly estimated using an immediate heuristic, which in our case is the negative validation Mean Squared Error (MSE) of the equation at that node. This reward is then directly backpropagated.
- **Rollout Depth 1**: After selecting a leaf node for expansion, the MCTS performs a simulation of one additional propose-and-prune cycle. The reward obtained from this single-step rollout is used for backpropagation.
- **Rollout Depth 2**: Similar to the above, but the simulation (rollout) extends for two propose-and-prune cycles.

For each configuration and dataset, performance was measured by the test MSE, averaged over 10 random seeds, with 95% confidence intervals reported.

The results are presented in Table 7.

Table 7: **Impact of MCTS Rollout Depth on SGED Performance.** Test MSE (mean ± 95% CI) on the Lung Cancer with Chemo. & Radio. (simulated) and RNA Polymerase (real-world) datasets for different MCTS rollout depths. Results are averaged over 10 seeds. Lower MSE is better. The best performing variation for each dataset is indicated in bold.

| MCTS Rollout Variation | Lung Cancer with Chemo. & Radio. MSE ↓ | RNA Polymerase MSE ↓ |
| --- | --- | --- |
| No Rollout (Heuristic MCTS) | $0.0571 \pm 0.0367$ | $0.0118 \pm 0.000445$ |
| Rollout Depth 1 | $0.0211 \pm 0.0131$ | $0.0114 \pm 0.000720$ |
| Rollout Depth 2 | $\mathbf{0.0130} \pm 0.0123$ | $\mathbf{0.0113} \pm 0.000523$ |

The findings from this investigation, as summarized in Table 7, indicate a trend consistent with expectations for MCTS performance. For both the simulated Lung Cancer with Chemo. & Radio.

dataset and the real-world RNA Polymerase dataset, increasing the rollout depth generally leads to improved performance (i.e., lower test MSE). Specifically, moving from no rollout to a rollout depth of 1, and further to a rollout depth of 2, resulted in progressively lower mean MSE values.

On the Lung Cancer with Chemo. & Radio. dataset, a rollout depth of 2 achieved the lowest MSE (0.0130), a noticeable improvement over no rollout (0.0571) and rollout depth 1 (0.0211). Similarly, for the RNA Polymerase dataset, rollout depth 2 yielded the best mean MSE (0.0113), compared to 0.0118 for no rollout and 0.0114 for rollout depth 1. While the confidence intervals show some overlap, particularly for the RNA Polymerase dataset where the improvements are more modest, the consistent reduction in mean MSE suggests that deeper rollouts allow the MCTS to make more informed decisions during the search process. This enhanced lookahead capability helps SGED to better navigate the complex equation space and identify more accurate symbolic models. This benefit, however, should be weighed against the increased computational time required for deeper rollouts in practical applications.

Elsewhere in this work, unless otherwise specified, we use the Heuristic MCTS configuration.

### E.4 CONVERGENCE EFFICIENCY

A critical aspect of any equation discovery algorithm is its efficiency in navigating the vast search space of potential mathematical expressions. Rapid convergence to accurate and parsimonious solutions is highly desirable, as it reduces computational cost and accelerates the scientific discovery process. This section investigates the convergence efficiency of SGED compared to a standard Genetic Programming (GP) approach, GPLearn. We aim to demonstrate that the structured, influence-based feedback mechanism within SGED, particularly when coupled with MCTS, leads to faster identification of high-quality symbolic models.

**Experimental Setup and Iteration Comparability:** To assess convergence efficiency, we tracked the best test Mean Squared Error (MSE) achieved against the number of "major evaluation iterations" for both SGED and GPLearn on the Lung Cancer (with Chemo. & Radio.) benchmark dataset. Experiments were conducted over 10 random seeds, and we report the mean MSE ± 95% confidence intervals.

For SGED, an "iteration" is defined as one MCTS node expansion. Each expansion involves the core propose-and-prune cycle: the LLM proposes candidate basis functions, these are evaluated, influence scores are calculated, and the LLM prunes the terms to form a new candidate equation. This new equation is then fully evaluated.

For GPLearn, an "iteration" corresponds to one generation. In each generation, a population of candidate equations undergoes genetic operations (e.g., crossover, mutation), and each new individual (equation) in the subsequent population is evaluated.

While the precise computational operations within an SGED MCTS expansion (involving LLM calls and influence score calculations) differ from those in a GPLearn generation (dominated by genetic operations and fitness evaluations of a population), both represent a fundamental step where a set of new candidate equations are generated, fully evaluated, and used to guide the subsequent search. By plotting performance against these respective "major evaluation iterations" or, more broadly, "number of candidate equations fully evaluated," we gain insight into how quickly each algorithm explores promising regions of the equation space and refines its solutions. This provides a broadly comparable measure of how efficiently each method utilizes its core search mechanism to improve solution quality.

The iteration count was set to 30 for both methods. GPLearn was run with the following parameters: `population_size = 1000`, `parsimony_coefficient = 0.01`, with the rest of the parameters set to their default values.

**Results:** The convergence behavior of SGED and GPLearn is illustrated in Figure 7. The plot clearly shows that SGED converges significantly faster to a substantially lower test MSE compared to GPLearn on this benchmark. SGED rapidly identifies high-performing equations within a smaller number of major evaluation iterations. Specifically, SGED reached a final mean test MSE of $0.0752 \pm 0.0295$. In contrast, GPLearn's convergence was slower, and it settled at a considerably

higher final mean test MSE of $28.5 \pm 9.34$ within a comparable or number of its own evaluation iterations.

This superior convergence efficiency highlights the effectiveness of SGED's approach. The detailed, per-term influence feedback allows the LLM to make more informed decisions during the pruning phase, leading to more targeted exploration. Coupled with the systematic search of MCTS, SGED is able to more quickly discard unpromising avenues and focus on refining equation structures that demonstrate high predictive accuracy. This contrasts with the more stochastic and population-based exploration of GPLearn, which, in this case, required more evaluations to achieve a less optimal solution.

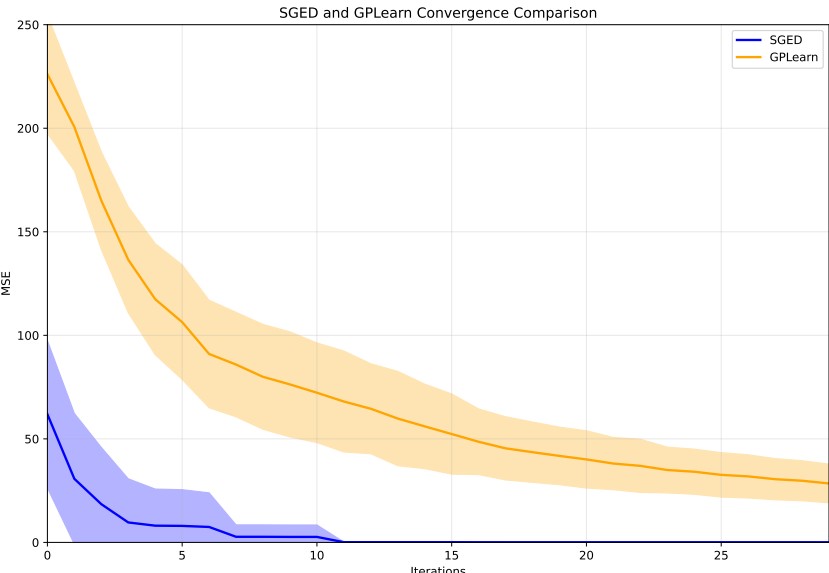

Figure 7: **Convergence Efficiency: SGED vs. GPLearn.** Test MSE (mean ± 95% CI over 10 seeds) versus number of major evaluation iterations for SGED (MCTS expansions) and GPLearn (generations) on the Lung Cancer (with Chemo. & Radio.) dataset. SGED demonstrates faster convergence to a lower MSE, indicating superior search efficiency.

This demonstrates SGED's ability not only to find accurate equations but to do so with greater efficiency in terms of the number of core search and evaluation steps.

### E.5 INVESTIGATION OF LLM SENSITIVITY

Understanding the robustness of SGED's advantages across different underlying Large Language Models (LLMs) is crucial for assessing the generalizability of our approach. This investigation aims to determine if the performance gains observed with SGED are primarily due to its structured, influence-based feedback mechanism and search strategy, rather than being contingent on the specific capabilities of a single LLM. Demonstrating such robustness would imply that the architectural innovations of SGED provide consistent benefits, offering practical value to users who may employ a variety of LLMs, including proprietary and open-weight models.

To evaluate this sensitivity, we conducted a comprehensive set of experiments comparing SGED (Ours) with the baseline LLM-driven methods: ZeroShot, ZeroOptim, and ICL (Basic Feedback). These four methods were evaluated on all six main benchmark datasets detailed in Section 4 and Appendix C: Lung Cancer, Lung Cancer (with Chemo.), Lung Cancer (with Chemo. & Radio.), COVID-19, RNA Polymerase, and Warfarin PK. For each method and dataset combination, we utilized a diverse suite of nine different base LLMs. The models used, their classification, and version notes are as follows:

- **Open-weight Models:**
    - `Llama-3.3-70B`: `Llama-3.3-70B-Instruct`, version 5.

- DeepSeek V3: `DeepSeek V3`, version `0324`.
- **Proprietary Models:**
    - GPT-4: `gpt-4`, version `turbo-2024-04-09`.
    - GPT-4o-Mini: `gpt-4o-mini`, version `2024-07-18`.
    - GPT-4o: `gpt-4o`, version `2024-11-20`.
    - GPT-4.1: `gpt-4.1`, version `2025-04-14`.
    - OpenAI o1: `o1`, version `2024-12-17`.
    - OpenAI o4-Mini: `o4-mini`, version `2025-04-16`.
    - OpenAI o3: `o3`, version `2025-04-16`.

All inference hyperparameters were set to their default values for each model throughout the experiments.

The performance was measured using the Mean Squared Error (MSE) on the respective test sets, consistent with our main evaluations. Each experiment was run with 25 seeds[2] to ensure reliability, and average MSE values are reported.

The results of this extensive evaluation are presented systematically. Figure 8 presents bar plots for all six datasets, showing the MSE outcomes. Corresponding numerical results are detailed in Table 8.

The bar plots presented in Figure 8 facilitate visual comparison of method performance across different LLMs. Each subfigure in Figure 8 corresponds to a specific dataset. Within each such dataset-specific plot, results for the four methods – ZeroShot, ZeroOptim, ICL (Basic Feedback), and SGED (Ours) – are displayed in separate panels. In every panel, the x-axis lists the different base LLMs, while the y-axis represents the Mean Squared Error (MSE), where lower bars indicate better performance. The ± 95% CI is shown as error bars. For improved clarity, the y-axis MSE values are presented on a logarithmic scale. Notably, due to its significantly MSE, the ZeroShot method is plotted with a y-axis range distinct from the other three methods within each dataset. Conversely, the y-axis ranges for ZeroOptim, ICL (Basic Feedback), and SGED (Ours) are all aligned with each other within the context of a single dataset to enable their direct comparison.

---

[2]Except `Llama-3.3-70B-Instruct`, where 10 seeds were used, due to limitations to model throughput that was available at the time of the experiments.

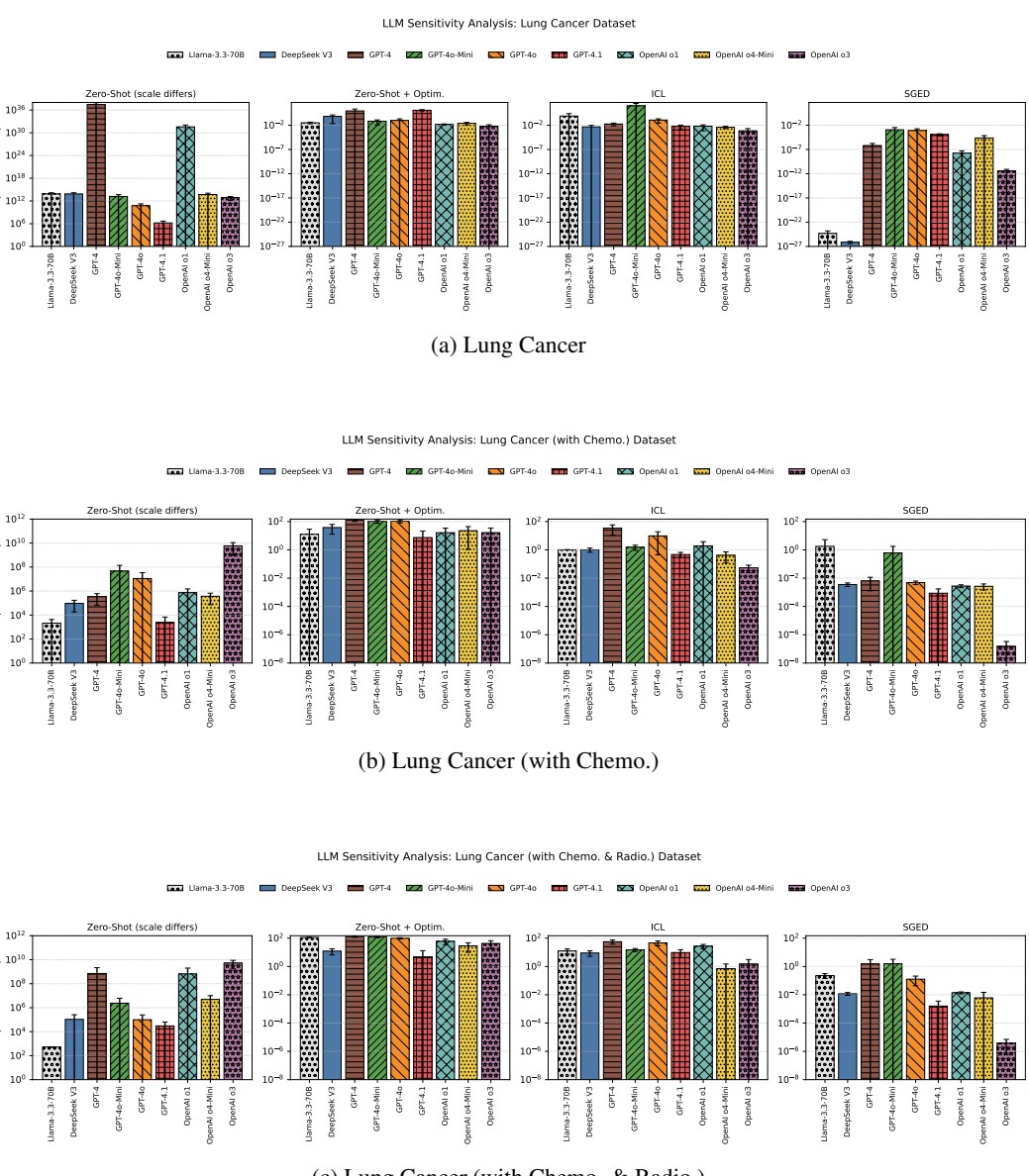

(a) Lung Cancer

(b) Lung Cancer (with Chemo.)

(c) Lung Cancer (with Chemo. & Radio.)

Figure 8: MSE (± 95% CI error bars) on benchmark datasets for ZeroShot, ZeroOptim, ICL (Basic Feedback), and SGED (Ours) across various base LLMs, including open-weight models. Each subfigure represents a different dataset. SGED outperforms the other methods across the datasets and base LLMs with few exceptions.
**Note:** The y axis (MSE) scale is logarithmic. For each dataset: the MSE range is different for the ZeroShot baseline, as its performance is significantly worse; the other methods are shown over matching ranges for clarity.
*(Figure continued to next page)*

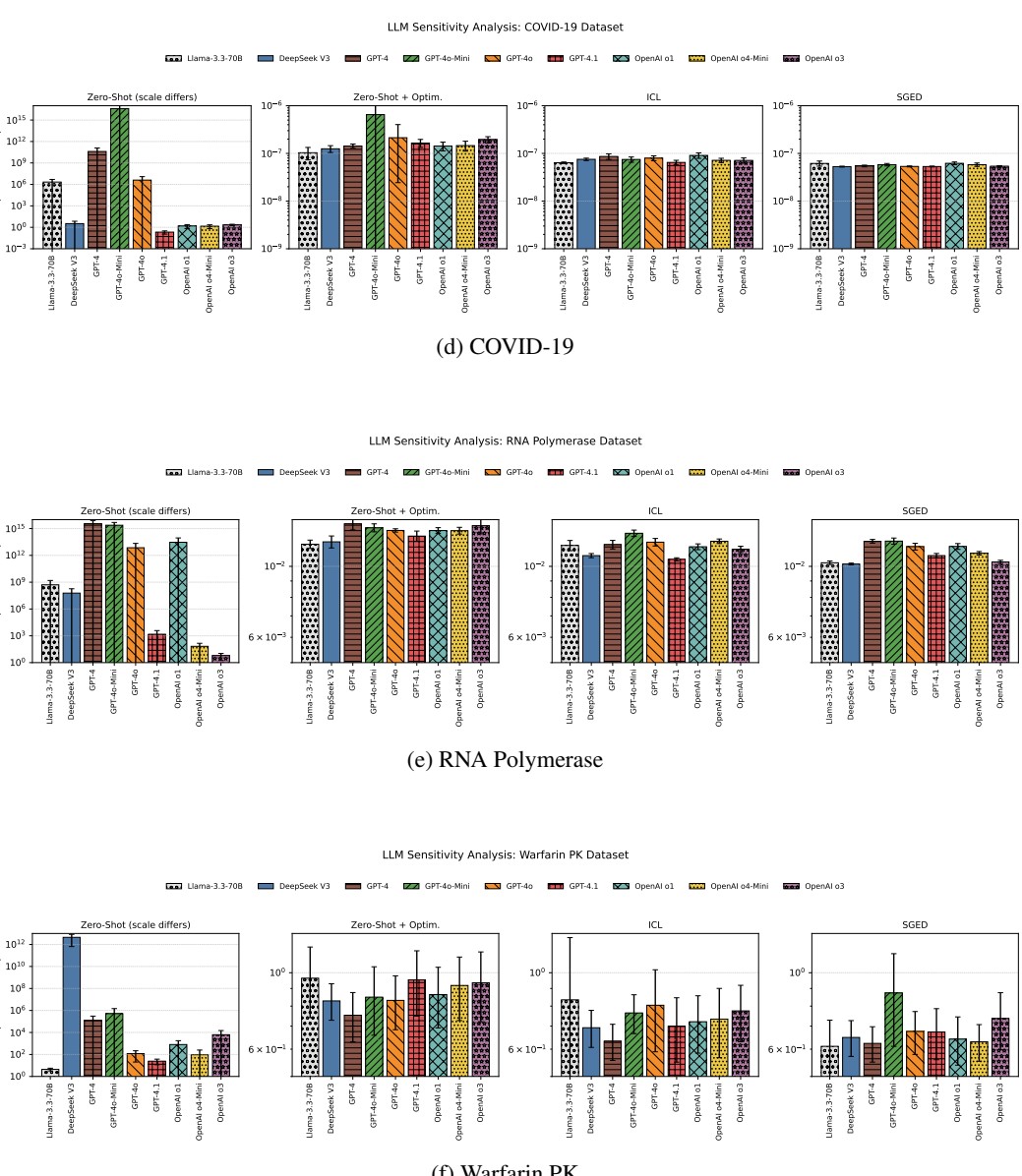

Figure 8: MSE (± 95% CI error bars) on benchmark datasets for ZeroShot, ZeroOptim, ICL (Basic Feedback), and SGED (Ours) across various base LLMs, including open-weight models. Each subfigure represents a different dataset. SGED outperforms the other methods across the datasets and base LLMs with few exceptions.

**Note:** The y axis (MSE) scale is logarithmic. For each dataset: the MSE range is different for the ZeroShot baseline, as its performance is significantly worse; the other methods are shown over matching ranges for clarity.

*(Figure completed)*

Table 8: Test MSE (mean ± 95% CI) for different methods and base LLMs across all benchmark datasets. The best performing method for each LLM is bolded. If the best method's confidence interval overlaps with any other method's CI for that LLM, only its mean is bolded; otherwise, both mean and CI are bolded.

(a) Lung Cancer

| Base LLM | ZeroShot | ZeroOptim | ICL (Basic Feedback) | SGED (Ours) |
|---|---|---|---|---|
| Llama-3.3-70B | 6.88e+13 ± 7.74e+13 | 0.0319 ± 0.0117 | 0.782 ± 1.72 | **4.89e-25** ± 1.09e-24 |
| DeepSeek V3 | 7.23e+13 ± 7.54e+13 | 0.679 ± 0.657 | 0.0046 ± 0.00468 | **7.55e-27** ± 3.76e-27 |
| GPT-4 | 2.85e+37 ± 5.88e+37 | 9.06 ± 13.1 | 0.0179 ± 0.0106 | **6.57e-07** ± 1.09e-06 |
| GPT-4o-Mini | 1.59e+13 ± 3.18e+13 | 0.0682 ± 0.0575 | 102 ± 210 | **0.00116** ± 0.00218 |
| GPT-4o | 5.75e+10 ± 1.15e+11 | 0.0971 ± 0.111 | 0.119 ± 0.0899 | **0.000792** ± 0.00105 |
| GPT-4.1 | 1.59e+06 ± 3.2e+06 | 12.4 ± 4.05 | 0.006 ± 0.00483 | **0.000118 ± 5.32e-05** |
| OpenAI o1 | 3.71e+31 ± 7.66e+31 | 0.0146 ± 0.00384 | 0.00629 ± 0.00558 | **1.74e-08** ± 3.49e-08 |
| OpenAI o4-Mini | 4.98e+13 ± 6.34e+13 | 0.025 ± 0.0133 | 0.00395 ± 0.00183 | **2.39e-05** ± 4.87e-05 |
| OpenAI o3 | 8.37e+12 ± 7.06e+12 | 0.00626 ± 0.00664 | 0.000717 ± 0.00131 | **3.54e-12 ± 4.4e-12** |

(b) Lung Cancer (with Chemo.)

| Base LLM | ZeroShot | ZeroOptim | ICL (Basic Feedback) | SGED (Ours) |
|---|---|---|---|---|
| Llama-3.3-70B | 2.13e+03 ± 2.14e+03 | 12.5 ± 16.4 | **1.01 ± 0.00448** | 1.76 ± 3.48 |
| DeepSeek V3 | 9.44e+04 ± 7.63e+04 | 37.8 ± 25 | 1.01 ± 0.329 | **0.00358 ± 0.00102** |
| GPT-4 | 3.38e+05 ± 2.73e+05 | 124 ± 20 | 34.7 ± 23.7 | **0.00644 ± 0.00516** |
| GPT-4o-Mini | 4.6e+07 ± 9.29e+07 | 101 ± 27.7 | 1.52 ± 0.608 | **0.603** ± 1.17 |
| GPT-4o | 1.11e+07 ± 2.28e+07 | 106 ± 25.1 | 9.74 ± 9.29 | **0.00489 ± 0.00139** |
| GPT-4.1 | 2.44e+03 ± 4.4e+03 | 7.45 ± 14.1 | 0.468 ± 0.193 | **0.000864 ± 0.000883** |
| OpenAI o1 | 7.39e+05 ± 7.73e+05 | 16.4 ± 18.3 | 1.89 ± 1.91 | **0.00277 ± 0.000628** |
| OpenAI o4-Mini | 3.7e+05 ± 2.97e+05 | 22.7 ± 21.6 | 0.417 ± 0.299 | **0.00267 ± 0.00123** |
| OpenAI o3 | 5.92e+09 ± 4.99e+09 | 16.5 ± 18.7 | 0.0533 ± 0.029 | **1.52e-07 ± 1.8e-07** |

(c) Lung Cancer (with Chemo. & Radio.)

| Base LLM | ZeroShot | ZeroOptim | ICL (Basic Feedback) | SGED (Ours) |
|---|---|---|---|---|
| Llama-3.3-70B | 524 ± 18.4 | 113 ± 0.92 | 12.5 ± 4.91 | **0.227 ± 0.0846** |
| DeepSeek V3 | 1.06e+05 ± 1.53e+05 | 12.2 ± 5.62 | 9 ± 3.8 | **0.0118 ± 0.00265** |
| GPT-4 | 7.07e+08 ± 1.46e+09 | 126 ± 10.3 | 56.4 ± 18.2 | **1.47** ± 1.6 |
| GPT-4o-Mini | 2.38e+06 ± 3.7e+06 | 118 ± 11.8 | 15.1 ± 3.56 | **1.57** ± 1.66 |
| GPT-4o | 9.92e+04 ± 1.5e+05 | 98.6 ± 10.3 | 46.3 ± 17.3 | **0.127 ± 0.083** |
| GPT-4.1 | 3.02e+04 ± 3.39e+04 | 4.76 ± 8.33 | 9.68 ± 5.6 | **0.0016** ± 0.00199 |
| OpenAI o1 | 6.51e+08 ± 1.34e+09 | 59.7 ± 24.4 | 27.1 ± 9.04 | **0.0137 ± 0.00223** |
| OpenAI o4-Mini | 4.84e+06 ± 5.33e+06 | 28.7 ± 18.6 | 0.708 ± 0.801 | **0.00608** ± 0.00867 |
| OpenAI o3 | 5.24e+09 ± 3.54e+09 | 42.7 ± 23.7 | 1.49 ± 1.61 | **3.82e-06 ± 3.22e-06** |

(d) COVID-19

| Base LLM | ZeroShot | ZeroOptim | ICL (Basic Feedback) | SGED (Ours) |
|---|---|---|---|---|
| Llama-3.3-70B | 2.13e+06 ± 2.84e+06 | 1.03e-07 ± 3.01e-08 | **6.38e-08 ± 2.41e-09** | 6.16e-08 ± 7.54e-09 |
| DeepSeek V3 | 3.3 ± 3.76 | 1.25e-07 ± 2.05e-08 | 7.52e-08 ± 4.33e-09 | **5.28e-08 ± 1.18e-09** |
| GPT-4 | 4.03e+10 ± 8.25e+10 | 1.43e-07 ± 1.44e-08 | 8.63e-08 ± 1.17e-08 | **5.45e-08 ± 2.28e-09** |
| GPT-4o-Mini | 4e+16 ± 8.26e+16 | 6.51e-07 ± 7.71e-07 | 7.47e-08 ± 9.47e-09 | **5.81e-08 ± 3.45e-09** |
| GPT-4o | 4e+06 ± 8.26e+06 | 2.13e-07 ± 1.89e-07 | 7.95e-08 ± 9.33e-09 | **5.33e-08 ± 1.27e-09** |
| GPT-4.1 | 0.213 ± 0.102 | 1.65e-07 ± 3.17e-08 | 6.47e-08 ± 6.96e-09 | **5.31e-08 ± 1.17e-09** |
| OpenAI o1 | 1.53 ± 0.857 | 1.43e-07 ± 2.99e-08 | 9.07e-08 ± 1.19e-08 | **6.2e-08 ± 4.14e-09** |
| OpenAI o4-Mini | 1.44 ± 0.791 | 1.46e-07 ± 3.45e-08 | 7.16e-08 ± 7.49e-09 | **5.81e-08 ± 4.81e-09** |
| OpenAI o3 | 2.16 ± 0.514 | 1.95e-07 ± 2.92e-08 | 7.13e-08 ± 9.05e-09 | **5.4e-08 ± 1.28e-09** |

(e) RNA Polymerase

| Base LLM | ZeroShot | ZeroOptim | ICL (Basic Feedback) | SGED (Ours) |
|---|---|---|---|---|
| Llama-3.3-70B | 5.08e+08 ± 1.05e+09 | 0.0117 ± 0.00035 | 0.0116 ± 0.000425 | **0.0103 ± 0.000129** |
| DeepSeek V3 | 6.01e+07 ± 1.24e+08 | 0.0119 ± 0.000514 | 0.0108 ± 0.000163 | **0.0102 ± 6.65e-05** |
| GPT-4 | 3.46e+15 ± 4.49e+15 | 0.0136 ± 0.000579 | 0.0117 ± 0.000362 | **0.012 ± 0.00016** |
| GPT-4o-Mini | 2.36e+15 ± 2.46e+15 | 0.0132 ± 0.000381 | 0.0127 ± 0.000295 | **0.012 ± 0.000268** |
| GPT-4o | 7.16e+12 ± 1.48e+13 | 0.013 ± 0.000156 | **0.0119 ± 0.000323** | 0.0115 ± 0.000279 |
| GPT-4.1 | 1.5e+03 ± 2.36e+03 | 0.0124 ± 0.00046 | **0.0105 ± 0.000118** | 0.0108 ± 0.000168 |
| OpenAI o1 | 2.91e+13 ± 6e+13 | 0.0129 ± 0.000275 | **0.0115 ± 0.00024** | 0.0115 ± 0.000242 |
| OpenAI o4-Mini | 66.9 ± 78.7 | 0.0129 ± 0.000315 | 0.012 ± 0.000196 | **0.011 ± 0.000137** |
| OpenAI o3 | 6.18 ± 4.42 | 0.0134 ± 0.000774 | 0.0113 ± 0.000258 | **0.0103 ± 0.000124** |

(f) Warfarin PK

| Base LLM | ZeroShot | ZeroOptim | ICL (Basic Feedback) | SGED (Ours) |
|---|---|---|---|---|
| Llama-3.3-70B | 4.42 ± 1.23 | 0.964 ± 0.223 | 0.834 ± 0.43 | **0.612 ± 0.117** |
| DeepSeek V3 | 4.46e+12 ± 3.82e+12 | 0.828 ± 0.1 | 0.692 ± 0.0853 | **0.649 ± 0.0773** |
| GPT-4 | 1.21e+05 ± 1.86e+05 | 0.753 ± 0.123 | 0.633 ± 0.076 | **0.624 ± 0.0734** |
| GPT-4o-Mini | 5.15e+05 ± 9.64e+05 | **0.85 ± 0.191** | 0.764 ± 0.0987 | 0.874 ± 0.262 |
| GPT-4o | 119 ± 98.5 | 0.831 ± 0.148 | 0.805 ± 0.215 | **0.675 ± 0.0961** |
| GPT-4.1 | 24.6 ± 13 | 0.953 ± 0.205 | 0.698 ± 0.148 | **0.673 ± 0.114** |
| OpenAI o1 | 798 ± 995 | 0.864 ± 0.173 | 0.721 ± 0.137 | **0.642 ± 0.103** |
| OpenAI o4-Mini | 94.1 ± 158 | 0.918 ± 0.192 | 0.734 ± 0.167 | **0.63 ± 0.0768** |
| OpenAI o3 | 6.24e+03 ± 8.68e+03 | 0.935 ± 0.214 | 0.775 ± 0.144 | **0.736 ± 0.139** |

Across all datasets and underlying LLM choices, SGED consistently outperforms ZeroShot, ZeroOptim, and ICL (Basic Feedback), with only a few exceptions (see Table 8). This strongly suggests that the structured feedback and systematic search strategy integral to SGED are key drivers of its enhanced equation discovery capabilities, providing a robust advantage irrespective of the specific foundational LLM employed. These findings underscore the value of SGED's methodology in leveraging LLMs for scientific discovery.

### E.6 Investigation of Robustness to a Large Number of Irrelevant Features

Scientific datasets, particularly in fields like genomics or high-throughput screening, can often present a high-dimensional feature space where many features may be irrelevant or noisy. A critical aspect of an effective equation discovery method is its ability to identify the truly influential variables amidst a large number of distractors. This section investigates the robustness of SGED to an increasing number of such irrelevant features.

We conducted experiments using the **Lung Cancer (with Chemo. & Radio.)** benchmark dataset. To simulate scenarios with varying degrees of feature space complexity, we augmented the original dataset by adding 5, 75, or 150 additional "distractor" features. These features were populated with values drawn from a standard normal distribution ($\mathcal{N}(0, 1)$) and were designed to have no causal relationship with the target variables. The performance of SGED (denoted as "**SGED (Ours)**" in the table) was compared against the LLM-based baselines: ZeroShot, ZeroOptim, and ICL (Basic

Feedback). All methods were evaluated based on their Mean Squared Error (MSE) on the test set, averaged over 25 random seeds, with 95% confidence intervals reported.

The results are presented in Table 9.

Table 9: **Performance with Increasing Numbers of Irrelevant Features.** Test MSE (mean ± 95% CI) on the Lung Cancer (with Chemo. & Radio.) dataset, augmented with varying numbers of normally distributed, irrelevant features. Results are averaged over 25 seeds. Lower MSE is better. "+0 Features" corresponds to the original dataset.

| Method | +0 Features (Original) | +5 Features | +75 Features | +150 Features |
|---|---|---|---|---|
| ZeroShot | 2.54e+03 ± 2.74e+03 | 1.48e+03 ± 2.35e+03 | 3.08e+03 ± 6.18e+03 | 1.57e+04 ± 2.18e+04 |
| ZeroOptim | 122 ± 6 | 142 ± 12.2 | 131 ± 27.8 | 132 ± 22.7 |
| ICL (Basic Feedback) | 63.3 ± 16.5 | 111 ± 32.6 | 118 ± 32.5 | 126 ± 22.5 |
| **SGED (Ours)** | **0.0521 ± 0.0178** | **0.0843 ± 0.0421** | **2.99 ± 5.95** | **4.88 ± 9.8** |

As evident from Table 9, SGED consistently achieves the lowest MSE across all conditions, even when a substantial number of irrelevant features are introduced. While the performance of SGED does show some degradation as the number of distractor features increases (MSE from 0.0521 with 0 extra features to 4.88 with 150 extra features), its accuracy remains significantly better—often by orders of magnitude—than the other LLM-based approaches. For instance, with 150 additional irrelevant features, SGED achieves an MSE of 4.88, whereas the next best LLM-based method, ICL (Basic Feedback), has an MSE of 126.

The other methods (ZeroShot, ZeroOptim, ICL) show varied responses to the additional features but consistently perform much worse than SGED. Their inability to effectively filter out the noise, even with optimization (ZeroOptim) or basic iterative feedback (ICL), highlights the challenge distractor features pose. These findings suggest that SGED's architecture, particularly its use of per-term influence scores and systematic search (with MCTS), equips the LLM with a more robust mechanism to discern and prioritize relevant features.

### E.7 INVESTIGATION OF SYNTHETIC MODEL BENCHMARK

To further assess the robustness and generalization capabilities of SGED, particularly its performance on model structures that may not be prevalent in its LLM's training data, we conducted evaluations on a procedurally generated synthetic model benchmark. By creating synthetic models with known ground-truth equations incorporating diverse mathematical operators and interactions, we can test the ability of SGED to discover accurate equations even when faced with potentially unfamiliar functional forms.

These synthetic datasets were derived by introducing specific modifications to the underlying structure of the previously described lung cancer model with chemotherapy and radiotherapy. The modifications included the incorporation of trigonometric operators, division operators, and novel interaction terms, resulting in five distinct synthetic models:

- **Synthetic 1 (inc. $\gamma \sin(\omega t)$):** Introduces a sinusoidal forcing term. The underlying differential equation is:
$$\frac{dx(t)}{dt} = \left( \rho \log \left( \frac{K}{x(t)} \right) - \beta_c C(t) - (\alpha_r d(t) + \beta_r d(t)^2) + \gamma \sin(\omega t) \right) x(t)$$

- **Synthetic 2 (inc. $-\delta I(t)$):** Incorporates an additional linear negative feedback term $I(t)$.
$$\frac{dx(t)}{dt} = \left( \rho \log \left( \frac{K}{x(t)} \right) - \beta_c C(t) - (\alpha_r d(t) + \beta_r d(t)^2) - \delta I(t) \right) x(t)$$

- **Synthetic 3 (inc. $\log(\frac{K}{x(t)+N(t)})$):** Modifies the growth term to include an additional variable $N(t)$ in the denominator of the logistic term.
$$\frac{dx(t)}{dt} = \left( \rho \log \left( \frac{K}{x(t) + N(t)} \right) - \beta_c C(t) - (\alpha_r d(t) + \beta_r d(t)^2) \right) x(t)$$

- **Synthetic 4 (inc. $\epsilon \cos(\phi t)$)**: Adds a cosine forcing term.

$$\frac{dx(t)}{dt} = \left( \rho \log \left( \frac{K}{x(t)} \right) - \beta_c C(t) - (\alpha_r d(t) + \beta_r d(t)^2) + \epsilon \cos(\phi t) \right) x(t)$$

- **Synthetic 5 (inc. $\theta C(t)d(t)$)**: Introduces a multiplicative interaction term between chemotherapy $C(t)$ and radiotherapy $d(t)$.

$$\frac{dx(t)}{dt} = \left( \rho \log \left( \frac{K}{x(t)} \right) - \beta_c C(t) - (\alpha_r d(t) + \beta_r d(t)^2) - \theta C(t)d(t) \right) x(t)$$

Here, $x(t)$ represents the tumor volume, $C(t)$ is the chemotherapy effect, $d(t)$ is the radiotherapy effect, and other parameters $(\rho, K, \beta_c, \alpha_r, \beta_r, \gamma, \omega, \delta, N(t), \epsilon, \phi, \theta)$ are constants or time-varying inputs specific to each synthetic model.

The performance of SGED was compared against the LLM-based baselines: ZeroShot, ZeroOptim, and ICL (Basic Feedback). The results, in terms of test Mean Squared Error (MSE), are presented in Table 10.

Table 10: **Synthetic Model Benchmark Performance.** Test MSE (mean ± 95% CI) on five procedurally generated synthetic datasets. Results are averaged over 25 seeds. Lower MSE is better. SGED (Ours) consistently outperforms other LLM-based methods.

| Method | Synthetic 1 | Synthetic 2 | Synthetic 3 | Synthetic 4 | Synthetic 5 |
|---|---|---|---|---|---|
| ZeroShot | 3.42e+03 ± 2.59e+03 | 1.14e+04 ± 2.17e+04 | 1.11e+03 ± 470 | 3.12e+05 ± 6.36e+05 | 7.8e+03 ± 5.55e+03 |
| ZeroOptim | 153 ± 13.5 | 65.3 ± 7.53 | 106 ± 14.6 | 147 ± 14.8 | 118 ± 12.3 |
| ICL (Basic Feedback) | 102 ± 17.7 | 48.3 ± 10.7 | 40.4 ± 11.3 | 89.2 ± 12 | 42.6 ± 16.3 |
| **SGED (Ours)** | **52.7 ± 0.307** | **0.189 ± 0.158** | **0.0272 ± 0.0153** | **52.5 ± 0.296** | **0.036 ± 0.0202** |

As shown in Table 10, SGED consistently achieves significantly lower MSE compared to ZeroShot, ZeroOptim, and ICL (Basic Feedback) across all five synthetic datasets. This strong performance on models with varied and potentially novel structures (such as trigonometric terms or modified logistic growth factors) underscores SGED's ability to effectively search the equation space and adapt its discovery process. The granular, influence-based feedback appears crucial in guiding the LLM to identify relevant terms and construct accurate models, even when the underlying system dynamics deviate from more common forms. These results further highlight the robustness of the SGED framework and its potential for discovering meaningful equations in diverse scientific domains.

### E.8 GENERALIZATION STUDY ON THE RNA POLYMERASE DATASET

A crucial test for any equation discovery method aiming for scientific relevance is its ability to generalize not just to held-out data from the same experiment, but to entirely new, independent measurements. A model that performs well on a new biological replicate is more likely to have captured true underlying biological principles rather than just fitting noise or artifacts specific to a single experiment.

To rigorously assess this, we obtained a second biological replicate for the RNA Polymerase II pausing measurements, representing a separate laboratory experiment (used as the target `pause_score`; henceforth, *"Replicate 2"*). We then took the final equations discovered by SGED and several LLM-based baselines – which were trained and selected using only the original dataset (*"Replicate 1"*) – and evaluated their predictive performance on this new, unseen replicate without any re-fitting. This provides a strong test of out-of-distribution generalization.

The results, summarized in Table 11, demonstrate the robust generalization of the SGED-discovered models. While all methods experienced a predictable and modest drop in performance when evaluated on the new replicate – an expected outcome given inter-experiment variability – SGED maintained its superior accuracy.

On *Replicate 2*, SGED achieved the lowest Mean Squared Error (MSE) of $0.00793 \pm 0.000214$ and the highest coefficient of determination ($R^2$) of $0.142 \pm 0.0232$. This indicates that the SGED-discovered equation explained the most variance in the independent dataset, outperforming the next-best method, ICL (Basic Feedback), which achieved an MSE of $0.00815$ and an $R^2$ of $0.119$.

The strong performance on a true biological replicate provides compelling evidence that the structured, influence-based feedback mechanism of SGED guides the discovery process towards equations that capture genuine, reproducible biological principles.

Table 11: **Generalization to an Independent Biological Replicate.** Performance of discovered equations on the original test set (Replicate 1) versus a new, unseen biological replicate (Replicate 2) for the RNA Polymerase dataset. Models were trained/selected using only Replicate 1 data. Results are mean $\pm$ 95% CI over 25 seeds. SGED shows the best generalization, maintaining the lowest MSE and highest $R^2$ on the new replicate.

| Method | Performance on Original Test Set (Replicate 1) | | | Performance on New Biological Replicate (Replicate 2) | | |
|---|---|---|---|---|---|---|
| | MSE $\downarrow$ | $R^2 \uparrow$ | NRMSE $\downarrow$ | MSE $\downarrow$ | $R^2 \uparrow$ | NRMSE $\downarrow$ |
| ZeroShot | 1.35e+05 $\pm$ 1.75e+05 | -9.80e+06 $\pm$ 1.28e+07 | 1.55e+03 $\pm$ 1.15e+03 | 1.40e+05 $\pm$ 1.72e+05 | -1.52e+07 $\pm$ 1.87e+07 | 1.97e+03 $\pm$ 1.42e+03 |
| ZeroOptim | 0.0130 $\pm$ 0.000287 | 0.0724 $\pm$ 0.0199 | 0.963 $\pm$ 0.0101 | 0.00887 $\pm$ 0.000159 | 0.0406 $\pm$ 0.0188 | 0.979 $\pm$ 0.00937 |
| ICL (Basic Feedback) | 0.0119 $\pm$ 0.000352 | 0.149 $\pm$ 0.0251 | 0.922 $\pm$ 0.0137 | 0.00815 $\pm$ 0.000246 | 0.119 $\pm$ 0.0262 | 0.938 $\pm$ 0.0140 |
| **SGED (Ours)** | **0.0115 $\pm$ 0.000312** | **0.176 $\pm$ 0.0226** | **0.907 $\pm$ 0.0125** | **0.00793 $\pm$ 0.000214** | **0.142 $\pm$ 0.0232** | **0.926 $\pm$ 0.0126** |

## E.9 INFLUENCE SCORE VARIANTS

In SGED, the per-term influence score, $\Delta_j$, is computed on a *validation* split by deleting a single term (setting $w_j \leftarrow 0$) while holding the remaining coefficients fixed. This is the explicit definition of $\Delta_j$ used throughout the method and its pruning phase. The corresponding pruning prompt includes, as an illustration, an OLS identity, $\Delta_k = \frac{w_k^2}{n} \sum_i \phi_k(x_i)^2$, with the comment "always $\geq 0$" and an instruction to "treat terms independently; no need to refit or update weights."

The applicability of these statements hinges on the OLS orthogonality that nullifies cross-terms, a property that holds for the *training* data at the optimum but does not necessarily extend to a disjoint validation set. To ensure this distinction is clear, the prompt contains an explicit note that validation-computed values "may not always be $\geq 0$".

Here, as part of an additional investigation, we also implement and evaluate two *refit-aware* influence score alternatives that align the definition more closely with extra-sum-of-squares logic.

INFLUENCE SCORE COMPUTATION

**Notation.** Let $X \in \mathbb{R}^{n \times p}$ and $y \in \mathbb{R}^n$ denote training data; $X_{\mathrm{val}} \in \mathbb{R}^{n_{\mathrm{val}} \times p}$, $y_{\mathrm{val}} \in \mathbb{R}^{n_{\mathrm{val}}}$ denote validation data; $W \in \mathbb{R}^{p \times m}$ the fitted full–model coefficients (one column per output). On validation, $\widehat{Y}_{\mathrm{val}} = X_{\mathrm{val}} W$, with residuals $R_{\mathrm{val}} = Y_{\mathrm{val}} - \widehat{Y}_{\mathrm{val}}$. We use $\mathrm{MSE}_{\mathrm{val}}(W)_j = \frac{1}{n_{\mathrm{val}}} \| y_{\mathrm{val}}^{(j)} - (X_{\mathrm{val}} W)_{:,j} \|_2^2$.

**(A) No–refit** (default in SGED). Deleting term $k$ is implemented by $\widehat{y}_{\mathrm{val},-k}^{(j)} = (X_{\mathrm{val}} W)_{:,j} - \varphi_k^{\mathrm{val}} w_{k,j}$, where $\varphi_k^{\mathrm{val}} = X_{\mathrm{val}} e_k$. Expanding the squares yields the exact validation–split change

$$\Delta_{k,j}^{\mathrm{val}} = \frac{2 w_{k,j}}{n_{\mathrm{val}}} (\varphi_k^{\mathrm{val}})^\top r_{\mathrm{val}}^{(j)} + \frac{w_{k,j}^2}{n_{\mathrm{val}}} \| \varphi_k^{\mathrm{val}} \|_2^2, \quad r_{\mathrm{val}}^{(j)} = y_{\mathrm{val}}^{(j)} - (X_{\mathrm{val}} W)_{:,j}.$$

The cross–term vanishes on the *training* split at the OLS optimum (residuals orthogonal to columns of $X$), making the classic "$\geq 0$" identity a good approximation there; on validation the cross–term generally does not vanish and $\Delta_{k,j}^{\mathrm{val}}$ can be slightly negative. This approach is how SGED computes $\Delta_j$ before pruning, by default.

**(B) Refit–aware (full refit).** Define the refit–aware influence as the *validation* MSE change after *refitting on train* with column $k$ removed:

$$\Delta_{k,j}^{\mathrm{refit}} = \mathrm{MSE}_{\mathrm{val}}(W^{(-k)})_j - \mathrm{MSE}_{\mathrm{val}}(W)_j, \qquad W_{:,j}^{(-k)} = \arg \min_{u \in \mathbb{R}^{p-1}} \frac{1}{n} \| y^{(j)} - X_{(-k)} u \|_2^2$$

(or ridge adds $\lambda \|u\|_2^2$). This aligns with extra-sum-of-squares on train and yields a clean validation readout.

**(C) Refit–aware (efficient)**, compatible with OLS and ridge. Let $A = X^\top X$ (OLS) or $A = X^\top X + \lambda I$ (ridge), $B = A^{-1}$. Partition index $k$ from the rest and denote $\alpha_k = B_{kk}$, $\beta_{-k} = B_{-k,k}$. Writing $W = \begin{bmatrix} W_{-k,:} \\ W_{k,:} \end{bmatrix}$,

$$W_{-k,:}^{(-k)} = W_{-k,:} - \frac{\beta_{-k}}{\alpha_k} W_{k,:}$$

$$\widehat{Y}_{\mathrm{val}}^{(-k)} = X_{\mathrm{val}}^{(-k)} W_{-k,:}^{(-k)}$$

$$\Delta_{k,j}^{\mathrm{refit}} = \frac{1}{n_{\mathrm{val}}} \left\| y_{\mathrm{val}}^{(j)} - \widehat{y}_{\mathrm{val}}^{(j)(-k)} \right\|_2^2 - \mathrm{MSE}_{\mathrm{val}}(W)_j.$$

This avoids an explicit fit–loop, reusing $B$ to update all $p$ leave–one–out solutions in $O(p^2)$ per term (after a one–time $O(p^3)$ factorization), and works columnwise for multi–output.

**Where this is used in SGED.** The pipeline computes $\Delta_j$ on validation to guide pruning (Section 2, Appendix B.2); the prompt receives the per–term weights $w_j$ and validation–computed $\Delta_j$ to guide its term keep/drop decisions.

ABLATION ACROSS SIX BENCHMARKS: ACCURACY VS. COST

We compared the three influence score variants – (i) No–refit, (ii) Refit–aware (full), and (iii) Refit–aware (efficient) – across the six benchmark datasets described in Tables 12 and 13. To ensure a fair comparison of test performance, the same set of 15 random seeds was used for each variant.

Table 12: Test MSE (mean $\pm$ 95% CI) across datasets for the three influence variants.

| Variant | COVID-19 | Lung Cancer (with Chemo. & Radio.) | Lung Cancer (with Chemo.) | Lung Cancer | RNA Polymerase | Warfarin PK |
|---|---|---|---|---|---|---|
| No–refit | 5.35e-08 $\pm$ 1.93e-09 | 0.0311 $\pm$ 0.027 | 0.00304 $\pm$ 0.00194 | 3.55e-09 $\pm$ 5.3e-09 | 0.0114 $\pm$ 0.000398 | 0.597 $\pm$ 0.0821 |
| Refit–aware (full) | 5.24e-08 $\pm$ 1.67e-09 | 0.0337 $\pm$ 0.0064 | 0.00903 $\pm$ 0.0043 | 0.0065 $\pm$ 0.00561 | 0.0118 $\pm$ 0.000373 | 0.622 $\pm$ 0.0943 |
| Refit–aware (efficient) | 5.14e-08 $\pm$ 1.97e-09 | 0.0476 $\pm$ 0.0226 | 0.00756 $\pm$ 0.00169 | 0.0028 $\pm$ 0.00192 | 0.0119 $\pm$ 0.000391 | 0.639 $\pm$ 0.0888 |

Table 13: Wall–clock optimization time (mean $\pm$ 95% CI; seconds).

| Variant | COVID-19 | Lung Cancer (with Chemo. & Radio.) | Lung Cancer (with Chemo.) | Lung Cancer | RNA Polymerase | Warfarin PK |
|---|---|---|---|---|---|---|
| No–refit | 0.752 $\pm$ 0.0103 | 4.46 $\pm$ 0.564 | 5.54 $\pm$ 0.712 | 2.83 $\pm$ 0.193 | 13.5 $\pm$ 1.17 | 0.482 $\pm$ 0.0415 |
| Refit–aware (full) | 0.891 $\pm$ 0.0159 | 10.6 $\pm$ 1.56 | 10.7 $\pm$ 1.22 | 9.87 $\pm$ 1.14 | 23.1 $\pm$ 1.59 | 0.607 $\pm$ 0.0848 |
| Refit–aware (efficient) | 0.779 $\pm$ 0.0139 | 4.55 $\pm$ 0.512 | 4.31 $\pm$ 0.61 | 5.04 $\pm$ 0.662 | 18 $\pm$ 1.69 | 0.53 $\pm$ 0.0753 |

A repeated-measures ANOVA confirmed that there was *no significant difference* in test MSE among the three variants on any of the six datasets (all $p \geq 0.09$).[3] As expected, the refit–aware variants incur a non–trivial extra cost (Table 13) without providing a corresponding gain in accuracy (Table 12).

DISTRIBUTION OF INFLUENCE VALUES

We examined the empirical distribution of $\Delta$ values encountered during runs for *Lung Cancer (with Chemo. & Radio.)* and *RNA Polymerase* across all three variants. Two representative histograms (symlog $x$–axis for readability; outliers lightly clipped at extreme quantiles) are included in Fig. 9–10.

**Key summaries:**

| **Lung Cancer (with Chemo. & Radio.)**: share of influences (%) | | | | |
|---|---|---|---|---|
| **Variant** | $\Pr(\Delta < 0)$ | $\Pr(\Delta = 0)$ | $\Pr(\Delta > 0)$ | $n$ |
| No–refit | 18.19 | – | 81.81 | 21,912 |
| Refit–aware (full) | 24.59 | 0.05 | 75.37 | 21,590 |
| Refit–aware (efficient) | 29.53 | 0.14 | 70.33 | 22,398 |

---

[3]We conducted a one-factor repeated-measures ANOVA with Greenhouse-Geisser correction for each dataset. The statistics ($F(2, 18)$, $p_{\mathrm{GG}}$, partial $\eta^2$) were: Lung Cancer ($F = 0.19, p = 0.72, \eta_p^2 = 0.02$); Lung Cancer (with Chemo.) ($F = 2.17, p = 0.17, \eta_p^2 = 0.20$); Lung Cancer (with Chemo. & Radio.) ($F = 1.74, p = 0.22, \eta_p^2 = 0.16$); Warfarin PK ($F = 2.78, p = 0.09, \eta_p^2 = 0.24$); RNA Polymerase ($F = 1.49, p = 0.26, \eta_p^2 = 0.14$); and COVID-19 ($F = 1.05, p = 0.34, \eta_p^2 = 0.10$).

| **RNA Polymerase**: share of influences (%) | | | | |
|---|---|---|---|---|
| **Variant** | $\Pr(\Delta < 0)$ | $\Pr(\Delta = 0)$ | $\Pr(\Delta > 0)$ | $n$ |
| No–refit | 4.71 | – | 95.29 | 7,190 |
| Refit–aware (full) | 9.08 | 1.40 | 89.52 | 6,696 |
| Refit–aware (efficient) | 8.75 | 2.28 | 88.97 | 6,526 |

**Observations:** (i) negative values are infrequent, and where present their magnitudes are typically small compared to the bulk positive mass (see symlog histograms); (ii) the *shape* of the distribution is broadly similar across the three variants; (iii) taken together with the within–seed ANOVA analysis, these results suggest the validation–computed $\Delta$ is *stable enough* for pruning decisions even without per–term refits.

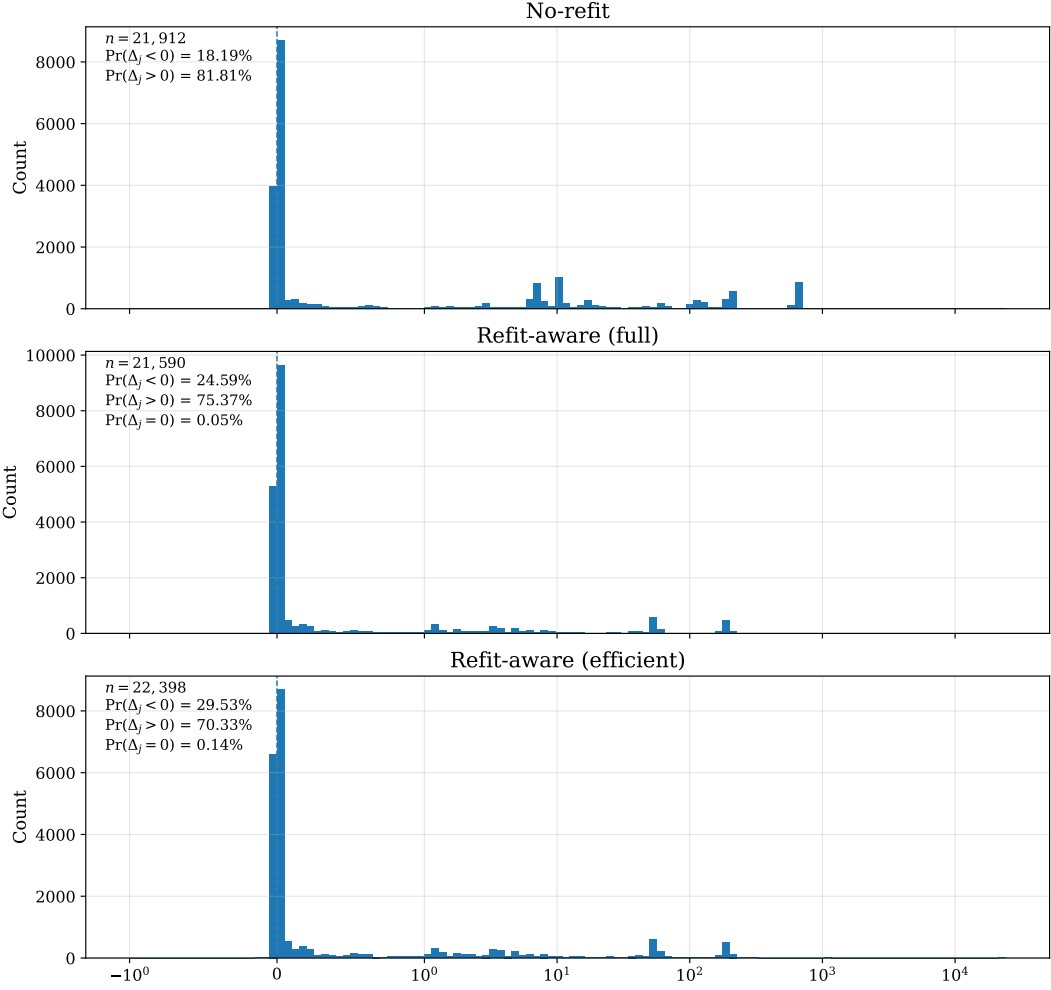

Figure 9: Distribution of influence values $\Delta$ on **Lung Cancer (with Chemo. & Radio.)** (symlog $x$–axis; extreme outliers clipped at the 0.01% tails for legibility).

SUMMARY OF FINDINGS

- **The Nature of the Influence Score.** Because the influence score, $\Delta$, is computed on a validation set *without* refitting the model, it is not constrained to be non-negative. Occasional small negative values are an expected result of this design and indicate that a feature has low or redundant influence in the context of the full model.

- **Performance vs. Cost.** Our experiments across six diverse datasets show that the test accuracy of the default, no-refit, and refit-aware variants is statistically indistinguishable. Given

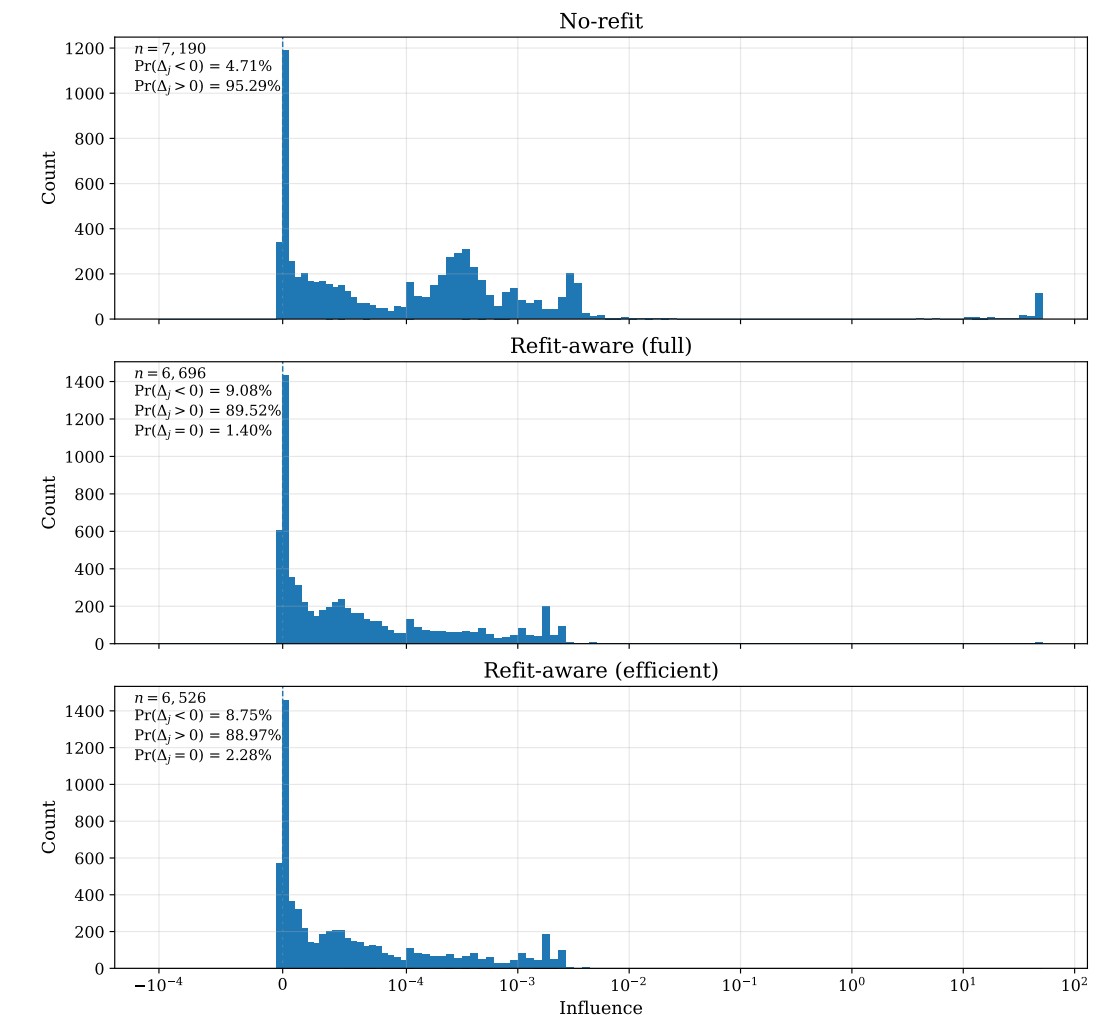

Figure 10: Distribution of influence values $\Delta$ on **RNA Polymerase** (symlog $x$–axis; extreme outliers clipped).

    that the refit-aware variants are more computationally expensive, this finding establishes the default **no-refit** method as an efficient approach that does not compromise test accuracy.

- **Implementation Flexibility.** For applications where a refit-aware analysis is specifically required, two alternatives are available via configuration. Setting `refit_aware=True` performs a full refit for each term, while `refit_aware_efficient=True` uses a faster computational update. Both methods provide the extra-sum-of-squares style score, $\Delta^{\mathrm{refit}}$, evaluated on the validation set.

### E.10 TERM-LOCAL OPTIMIZATION

To enhance SGED's flexibility, we introduce a variant that allows for the discovery of optimal scalar constants *within* the basis functions proposed by the LLM. This extension, which we refer to as SGED with Term-Local Optimization (SGED-TLO), addresses scenarios where an equation's true functional form involves specific constants that an LLM is unlikely to guess *a priori* (e.g., a decay rate in an exponential term or a frequency in a sinusoidal term). SGED-TLO integrates a dedicated optimization step to fine-tune these constants, enabling the discovery of more precise and potentially more accurate symbolic models.

The core principle is to augment the standard SGED loop: the "Propose" agent can now suggest basis functions containing tunable parameters, which are subsequently optimized before the "Prune" agent performs its influence-guided term selection.

**Parametric Basis Functions**    In this variant, the "Propose" LLM agent can include tunable scalar constants in its proposed terms using the syntax `c(init)`, where `init` is an initial value for the parameter. For example, an agent might propose a term like `np.sin(c(1.0) * x1)`. Each basis function $\psi_j$ can thus be a parametric function $\psi_j(\mathbf{x}; \theta_j)$, where $\theta_j$ is a vector of the tunable constants within that term. The collection of all such constants across all candidate terms is denoted by $\theta$.

To maintain the model's primary structure as a linear combination of basis functions, the use of `c()` is disallowed as a direct outer multiplier (e.g., `c(1.5) * x1`), as the per-term linear coefficient $w_j$ is already learned by the outer model. The "Propose" agent's prompt in SGED-TLO variant contains clear instructions and examples to reflect this setup.

**Term-Local Optimization Objective**    After the "Propose" agent suggests a new set of candidate terms, an additional optimization step is introduced before pruning. This step tunes the vector of all constants $\theta$ by minimizing an objective function $J(\theta)$. Crucially, this is a nested optimization problem. For any given set of constants $\theta$, the outer linear model weights $W$ are first re-computed by fitting on the training data. The objective $J(\theta)$ is then the Mean Squared Error (MSE) evaluated on the validation set using these optimal weights $W(\theta)$. This ensures that the constants are optimized for generalization performance.

Formally, for a given $\theta$, the design matrix on a data split $S \in \{\text{train, val}\}$ is $\Phi_S(\theta)$, where $[\Phi_S(\theta)]_{ik} = \psi_k(\mathbf{x}_i; \theta_k)$. The optimal outer weights are a function of $\theta$:

$$W(\theta) = \arg\min_{W'} \frac{1}{n_{\text{train}}m} \|Y_{\text{train}} - \Phi_{\text{train}}(\theta)W'\|_F^2 \ (+ \lambda\|W'\|_F^2 \text{ for ridge})$$

The objective for the inner-term constants is then to minimize the validation loss:

$$J(\theta) = \frac{1}{n_{\text{val}}m} \big\|Y_{\text{val}} - \Phi_{\text{val}}(\theta)\, W(\theta)\big\|_F^2$$

This objective is minimized with respect to $\theta$ using a quasi-Newton method (L-BFGS-B by default), with gradients estimated via finite differences.

**Dynamic Re-optimization and Agent Interaction**    A key feature of this variant is that the constants are not optimized once and then fixed. After optimization, the symbolic representation of the terms retains the `c(value)` markers, where `value` is now the optimized value. These symbolic forms are what the "Prune" agent sees in its feedback tables.

In each subsequent iteration of the SGED loop, the constants within *all* surviving and newly proposed terms are re-optimized together. This dynamic re-optimization allows the ideal value for a constant in one term to adapt to the presence or absence of other terms in the model, preserving maximal flexibility throughout the discovery process. The "Prune" agent is explicitly prompted that these constants are not static and will be re-tuned in subsequent rounds.

**Algorithm Flow**    The SGED-TLO propose-and-prune cycle proceeds as follows:

1. The LLM Propose agent suggests new terms, which may contain `c(init)` syntax. These are combined with surviving terms from the previous iteration.
2. The set of all constants $\theta$ from all current candidate terms is optimized by minimizing $J(\theta)$.
3. After optimization, the resulting terms are evaluated on the data splits (train, validation, test).
4. The standard pruning phase proceeds. The Prune agent receives the terms (with their optimized `c(value)` markers), their fitted outer weights $w_j$, and their per-term influence scores $\Delta_j$.
5. The agent returns keep/drop decisions. Surviving terms, retaining their `c()` markers, are passed to the next iteration.

This entire cycle is embedded within either the linear iterative refinement or the MCTS search strategy, just as in the standard SGED framework.

### E.10.1 PROOF OF CONCEPT EXPERIMENT

To provide a clear illustration of the specific advantage offered by term-local optimization, we conducted a proof-of-concept experiment on a simple synthetic dataset. The data was generated from the ground-truth equation $y = 1/(0.123 + x_1^2)$, which contains a non-trivial constant, $0.123$, that an LLM is highly unlikely to propose spontaneously. We ran both the standard SGED and the SGED-TLO variant on this dataset, using their non-tree-based iterative refinement modes for simplicity.

As hypothesized, SGED-TLO was uniquely capable of recovering the exact ground-truth equation. During its iterative search, the "Propose" agent eventually suggested the correct functional form with a tunable constant: `1 / (x_1**2 + c(init))`. The subsequent optimization step successfully tuned the constant to match the ground truth value of $0.123$. In the final pruning round, the feedback provided to the "Prune" agent was unambiguous. The correct parametric term had an influence score of $17.59$, while all other candidate terms had negligible influence (on the order of $10^{-17}$ or less). Guided by this overwhelming signal, the agent correctly kept only the single correct term and discarded all others, resulting in the exact solution with a final MSE of effectively zero ($1.07 \times 10^{-29}$). The final discovered equation was:

$$y = 1 / (x\_1**2 + 0.123)$$

By contrast, the standard SGED variant was unable to discover the true equation. Without the ability to tune the constant, it could not find a single basis function to accurately model the data. Instead, it was forced to approximate the target function by constructing a linear combination of multiple, non-parametric basis functions. After ten iterations, the best model it found was a complex five-term approximation:

```
y = 17.33 x_1 + 102.2 np.exp(-x_1) + 38.73 x_1 * np.exp(-x_1)
      - 18.34 np.abs(x_1) - 186.4 * 1 / (1 + np.exp(x_1))
```

While this model achieved a respectable test MSE of $0.0372$, it failed to capture the simple, parsimonious structure of the underlying data-generating process. In this experimental setting, with a 10 iteration budget, SGED-TLO was able to discover the exact expression in 10/25 runs, while vanilla SGED was able discover it in 0/25 runs (never). The experiment clearly demonstrates that for problems where precise constants are integral to the model's form, the term-local optimization capability of SGED-TLO is essential for discovering the correct symbolic solution.

### E.10.2 BENCHMARK RESULTS WITH TERM-LOCAL OPTIMIZATION

To assess the practical impact of term-local optimization, we compared the performance of SGED-TLO against the vanilla SGED framework across our six benchmark datasets. The results, presented in Table 14, show that the inclusion of tunable constants yields mixed outcomes depending on the dataset's characteristics. However, it is important to note that SGED-TLO's performance remains highly competitive, and is often superior, when compared to the broader set of baselines presented in Table 3.

Table 14: **Comparison of SGED vs. SGED-TLO.** Test MSE (mean $\pm$ 95% CI) across all benchmark datasets. Results are averaged over 25 seeds. Lower is better. Bold indicates the better-performing variant for each dataset (if confidence intervals overlap, only the mean of the better-performing variant is bold.)

| Benchmark Dataset | SGED | SGED-TLO |
|---|---|---|
| Lung Cancer | $0.0033 \pm 0.0035$ | $\mathbf{7.59e\text{-}06 \pm 1.33e\text{-}05}$ |
| Lung Cancer (with Chemo.) | $0.0054 \pm 0.00107$ | $\mathbf{0.000877 \pm 0.000548}$ |
| Lung Cancer (with Chemo. & Radio.) | $\mathbf{0.0521 \pm 0.0178}$ | $0.239 \pm 0.127$ |
| COVID-19 | $\mathbf{5.32e\text{-}08} \pm 1.35e\text{-}09$ | $5.33e\text{-}08 \pm 1.77e\text{-}09$ |
| RNA Polymerase | $\mathbf{0.0115} \pm 0.000312$ | $0.0119 \pm 0.000124$ |
| Warfarin PK | $\mathbf{0.646} \pm 0.105$ | $0.663 \pm 0.111$ |

For the **Lung Cancer** and **Lung Cancer (with Chemo.)** datasets, SGED-TLO demonstrates a marked improvement in accuracy. This is likely attributable to the increased flexibility afforded

by parametric terms, which can more closely approximate the underlying system dynamics. In the discovered equations for these datasets, we observe the selection of basis functions with optimized constants, such as those with the functional form $w_j \log(x + \theta)$ or $w_j \exp(\theta \cdot x)$. For instance, a high-performing equation discovered for the Lung Cancer (with Chemo.) task includes such adaptive terms:

$$
\begin{aligned}
\text{dv\_dt} = & -0.0002279 \cdot \text{chemo\_concentration} + 0.0001333 \cdot \text{chemo\_dosage} \\
& -2.358 \cdot \text{np.log}(\text{cancer\_volume} + \mathbf{4.06858}) \\
& -0.02799 \cdot \text{cancer\_volume} \cdot \text{chemo\_concentration} \\
& +1.736 \cdot \text{np.sqrt}(\text{cancer\_volume}) \\
& +1.945 \cdot \text{np.exp}(\mathbf{-0.00875506} \cdot \text{cancer\_volume}) \\
\text{dc\_dt} = & -0.5 \cdot \text{chemo\_concentration} + \text{chemo\_dosage}
\end{aligned}
$$

Conversely, on datasets like **Lung Cancer (with Chemo. & Radio.)** and **RNA Polymerase**, the performance of SGED-TLO is slightly worse than that of vanilla SGED. We speculate that this may be due to the significantly more complex and potentially non-convex optimization landscape introduced by the tunable constants $\theta$. The L-BFGS-B optimizer, while effective, may converge to local minima, particularly if the LLM's initial guesses for the constants are far from an optimal region. This could result in a set of basis functions that are locally optimal with respect to their internal constants but globally suboptimal for the final linear model, leading to a higher test MSE compared to the simpler, non-parametric terms found by the standard SGED. Performance on the COVID-19 and Warfarin PK datasets was comparable between the two variants.

The promising results on several datasets indicate that SGED-TLO is a valuable extension of the core framework. The approach and its variants warrant further investigation in future work, which could explore more sophisticated global optimization algorithms or improved heuristics for initializing the tunable constants. We provide the implementation of SGED-TLO as described in this section with the codebase for this work.

### E.11 INVESTIGATION IN THE CONTEXT OF AUTOMATED FEATURE ENGINEERING

As discussed in Appendix A.6, SGED shares conceptual parallels with Automated Feature Engineering (AFE). While SGED focuses on discovering a concise, interpretable symbolic equation, the basis functions $\psi_j(\mathbf{x})$ can be viewed as engineered features that transform the input space to make it linearly separable. A relevant question, therefore, is how SGED compares to established AFE frameworks and whether the restriction to a linear downstream model limits predictive performance compared to tree-based predictors.

To investigate this, we conducted a comparative study against *OpenFE* (Zhang et al., 2023a), a state-of-the-art non-LLM AFE method known for its effectiveness on tabular data. We chose OpenFE as a robust representative of the AFE landscape because many recent LLM-based AFE tools (e.g., CAAFE (Hollmann et al., 2023)[4], FeatLLM (Han et al., 2024)[5]) are primarily intended for classification tasks or lack standardized regression implementations.

**Experimental Setup.** We evaluated four configurations:

1. **SGED:** The standard SGED framework where features are discovered by the LLM and the predictor is a linear model.

2. **SGED + XGBoost:** The features $\psi_j(\mathbf{x})$ discovered by SGED are frozen and used as inputs to an XGBoost regressor. This tests whether a non-linear predictor can extract more value from the discovered symbolic features.

3. **OpenFE + Linear Regression:** OpenFE is used to generate and select features, followed by a linear predictor.

4. **OpenFE + XGBoost:** OpenFE is used to generate and select features, followed by an XGBoost predictor.

To ensure fair comparison, we constrained OpenFE to select the top 6 features, matching the parsimony constraint imposed on SGED (`keep_n_terms = 6`). We utilized the *Lung Cancer (with*

---

[4]Implementation at https://github.com/noahho/CAAFE
[5]Implementation at https://github.com/Sungwon-Han/FeatLLM

*Chemo. & Radio.)* dataset (complex dynamics, low dimensionality) and the *RNA Polymerase* dataset (high dimensionality, 263 features). Performance is reported using Normalized Mean Squared Error (NMSE) as defined in Shojaee et al. (2025b), over 10 seeds.

**Results on Complex Dynamics (Cancer Dataset).** As shown in Table 15, SGED significantly outperforms the OpenFE baselines on the Cancer dataset. OpenFE, which relies on expanding a predefined set of operators, struggles to reconstruct the specific functional forms (e.g., specific interaction terms or logarithmic growth laws) required to model the tumor dynamics accurately.

Interestingly, SGED with a linear backend outperforms SGED + XGBoost. Since the influence scores $\Delta_j$ used during the discovery process explicitly optimize for the reduction of linear least-squares error, the discovered basis functions tend to linearize the problem representation. In this context, the additional complexity of XGBoost does not yield a performance gain and may lead to slight overfitting compared to the linear analytic form.

Table 15: **AFE Comparison on Lung Cancer (with Chemo. & Radio.).** NMSE and Wall Clock time averaged over 10 seeds. SGED outperforms standard AFE baselines by orders of magnitude. Replacing the linear solver with XGBoost does not improve performance, confirming that SGED successfully discovers features that linearize the dynamics.

| Method (FE) | Predictor | NMSE ↓ | Wall Clock (s) ↓ |
|---|---|---|---|
| OpenFE | XGBoost | $0.183 \pm 0.038$ | $17.8 \pm 0.4$ |
| OpenFE | Linear Reg. | $0.202 \pm 0.041$ | $\mathbf{17.7} \pm 0.4$ |
| SGED | XGBoost | $0.0084 \pm 0.0005$ | $340.4 \pm 56.3$ |
| SGED | Linear Reg. | $\mathbf{0.0013} \pm \mathbf{0.0004}$ | $336.3 \pm 98.7$ |

Table 16: **AFE Scalability Comparison on RNA Polymerase (263 features).** NMSE and Wall Clock time averaged over 10 seeds. Traditional AFE (OpenFE) suffers from combinatorial explosion and times out. SGED leverages LLM priors to navigate the high-dimensional space efficiently.

| Method (FE) | Predictor | NMSE ↓ | Wall Clock (s) ↓ |
|---|---|---|---|
| OpenFE | XGBoost | DNF | DNF |
| OpenFE | Linear Reg. | DNF | DNF |
| SGED | XGBoost | $\mathbf{0.773} \pm 0.030$ | $318.9 \pm 22.1$ |
| SGED | Linear Reg. | $0.791 \pm 0.039$ | $\mathbf{310.0} \pm 37.5$ |

DNF: Did Not Finish within the allocated 3-hour limit.

**Scalability on High-Dimensional Data (RNA Pol II).** A critical limitation of traditional AFE methods is the combinatorial explosion of the search space as the number of input features increases. The RNA Polymerase dataset contains 263 input features. As detailed in Table 16, OpenFE failed to complete a single run within a 3-hour cutoff. Analysis of the logs revealed that OpenFE attempted to evaluate a pool of $716,320$ candidate features, rendering it computationally intractable for this regime.

In contrast, SGED leverages the LLM's semantic priors and the influence-based pruning mechanism to navigate this large feature space efficiently, converging in approximately 5 minutes. The performance of SGED (with default Linear Regression) and SGED + XGBoost was comparable on this dataset, suggesting that the discovered features captured the bulk of the signal regardless of the downstream predictor.

**Summary.** This investigation highlights two key advantages of SGED over traditional AFE pipelines for scientific discovery:

1. **Scalability via Priors:** SGED avoids the combinatorial explosion typical of generative feature engineering by using LLM priors to propose semantically relevant features, making it viable for high-dimensional datasets where traditional methods fail.

2. **Inherent Interpretability:** By optimizing features specifically for a linear backend, SGED produces a model that is fully interpretable ($f(\mathbf{x}) = \sum w_j \psi_j(\mathbf{x})$). The results show that this interpretability does not come at the cost of accuracy.

## E.12 INVESTIGATION OF PROPOSED TERMS COUNT

The `terms_per_round` hyperparameter controls the breadth of the search during the term generation phase. It dictates how many new candidate basis functions the "Propose" LLM agent suggests in each iteration of the propose-and-prune cycle. A higher number implies a wider exploration of the local equation space per step, potentially increasing the likelihood of finding correct terms, but also increasing the cognitive load on the LLM (context length) and the computational cost of evaluation.

To quantify this trade-off and justify our default configuration, we evaluated SGED on the *Lung Cancer (with Chemo. & Radio.)* dataset while varying `terms_per_round` across the set $\{1, 3, 5, 10, 20\}$. All other hyperparameters were maintained at their default values (including a maximum sparsity of 6 terms). We report the Normalized Mean Squared Error (NMSE, defined and implemented as per Shojaee et al., 2025b) on the test set, along with computational costs (Total Tokens and Wall Clock time), averaged over 25 independent seeds.

Table 17: **Impact of `terms_per_round` on Performance and Cost.** Evaluation on the *Lung Cancer (with Chemo. & Radio.)* dataset. Results are mean $\pm$ 95% confidence intervals over 25 seeds. Increasing the number of proposed terms improves predictive accuracy (lower NMSE) up to a saturation point, at the expense of increased computational cost. The default setting of 5 represents an efficient trade-off.

| Terms Per Round | NMSE ↓ | Total Tokens ↓ | Wall Clock (s) ↓ |
|---|---|---|---|
| 1 | $0.00223 \pm 0.000535$ | $271{,}785 \pm 34{,}154$ | $216.0 \pm 28.5$ |
| 3 | $0.00197 \pm 0.000809$ | $299{,}619 \pm 43{,}488$ | $248.4 \pm 33.2$ |
| **5 (Default)** | $\mathbf{0.00107 \pm 0.000483}$ | $\mathbf{303{,}615 \pm 42{,}967}$ | $\mathbf{280.5 \pm 39.2}$ |
| 10 | $0.000742 \pm 0.000435$ | $324{,}178 \pm 34{,}697$ | $349.9 \pm 41.8$ |
| 20 | $0.000877 \pm 0.000445$ | $343{,}885 \pm 59{,}708$ | $351.1 \pm 51.5$ |

The results are summarized in Table 17. The data reveals a clear trend: increasing the proposal width generally leads to improved model accuracy. Moving from suggesting 1 term to 5 terms per round reduces the NMSE by approximately half (from 0.00223 to 0.00107). This improvement highlights the benefit of providing the "Prune" agent with a sufficiently diverse pool of candidates; the influence-based feedback mechanism ($\Delta_j$) is most effective when the candidate set contains high-quality options to select from.

However, diminishing returns are observed at higher values. While increasing the proposal count to 10 yields the lowest mean NMSE (0.000742), this represents a smaller marginal gain compared to the jump from 1 to 5. Furthermore, proposing 20 terms does not improve performance beyond the 10-term setting, potentially due to the increased difficulty for the LLM to attend to and reason about a larger list of candidates within the prompt context or simply due to saturation of the search space.

In terms of computational efficiency, costs scale with the number of terms. The setting of 10 terms increases the wall-clock time by approximately 25% compared to the setting of 5 terms. The default value of `terms_per_round = 5` was thus selected as the operational "sweet spot" for SGED. It achieves a substantial performance improvement over lower values (nearly halving the error compared to 3 terms) with only a modest increase in runtime, ensuring the method remains computationally efficient while maintaining high discovery capability. Acknowledging that the optimal configuration is contingent upon specific dataset characteristics and computational constraints, we recommend 5 to 10 terms per round as a robust heuristic for general application.

## E.13 INVESTIGATION OF PROMPT VARIATIONS

The reliance of SGED on Large Language Models raises the question of sensitivity to prompt engineering: to what extent does performance depend on the specific, detailed instructions and context provided in the default prompts? Furthermore, the context window length and associated

token costs are practical concerns, particularly when dealing with high-dimensional datasets where providing data previews can consume significant bandwidth.

To address these questions, we investigated the performance of SGED using a set of *Simplified* prompts. We compared this variant against the *Default* prompt configuration described in Appendix B.6. The simplified prompts differ from the default prompts in the following key aspects:

- **Data Preview Removal:** In the "Propose" phase, the snippet of raw input data and target variable preview was removed. This is the primary driver of context length reduction, particularly for datasets with many features.
- **Instruction Compression:** For both "Propose" and "Prune" agents, verbose instructions and detailed task descriptions were collapsed into concise lists.
- **Pruning Simplification:** The extensive "Input You Receive" and "Your Task" sections, along with the example input/output tables, were removed from the "Prune" agent prompt. Only the core input tables (terms, weights, influence), history, and essential output formatting constraints were retained.

We evaluated both variants on the *Lung Cancer (with Chemo. & Radio.)* dataset (representing complex dynamics) and the *RNA Polymerase* dataset (representing high-dimensional data). Experiments were conducted over 10 random seeds. We report the Normalized Mean Squared Error (NMSE) on the test set, along with the total token consumption and wall-clock time.

The predictive performance results are presented in Table 18. While the *Default* prompts yield the lowest mean NMSE on both datasets, the *Simplified* prompts remain competitive. On the RNA Polymerase dataset, the performance gap is relatively narrow (0.824 vs 0.854), and the confidence intervals overlap significantly. On the Lung Cancer dataset, the *Simplified* variant exhibits higher variance and a higher mean error (0.0219 vs 0.0013), suggesting that for complex dynamical systems, the detailed instructions and examples in the *Default* prompt help stabilize the LLM's reasoning.

Table 18: **Impact of Prompt Variations on Predictive Accuracy.** NMSE (mean $\pm$ 95% CI) on test sets over 10 seeds. *Default* prompts achieve lower mean error, but *Simplified* prompts demonstrate reasonable robustness.

| Dataset | SGED (Default) | SGED (Simplified) |
|---|---|---|
| Lung Cancer (with Chemo. & Radio.) | **0.00127** $\pm$ 0.00087 | 0.0219 $\pm$ 0.0439 |
| RNA Polymerase | **0.824** $\pm$ 0.031 | 0.854 $\pm$ 0.009 |

However, the trade-off becomes distinct when analyzing computational costs, as shown in Table 19. The *Simplified* variant achieves a substantial reduction in token usage. For the high-dimensional RNA Polymerase dataset (263 features), removing the data preview reduced the average token consumption from $\approx$1.68 million to $\approx$140,000. This also translated to a wall-clock speedup of approximately 2.5$\times$.

Table 19: **Computational Cost of Prompt Variations.** Comparison of total token usage and wall-clock time (seconds) averaged over 10 seeds. The *Simplified* prompts offer significant efficiency gains, particularly for high-dimensional data (RNA Polymerase).

| Dataset | Metric | SGED (Default) | SGED (Simplified) |
|---|---|---|---|
| Lung Cancer (with Chemo. & Radio.) | Total Tokens | 291,540 $\pm$ 45,848 | 116,301 $\pm$ 19,301 |
| | Wall Clock (s) | 271.8 $\pm$ 39.9 | 153.7 $\pm$ 16.0 |
| RNA Polymerase | Total Tokens | 1,675,638 $\pm$ 151,933 | 139,835 $\pm$ 8,714 |
| | Wall Clock (s) | 296.5 $\pm$ 28.2 | 112.0 $\pm$ 11.6 |

These findings suggest that SGED is reasonably robust to variations in prompt detail. While detailed instructions and examples (as in the *Default* configuration) maximize discovery accuracy and stability, the core efficacy of the method is driven by the influence-based feedback mechanism rather than prompt engineering alone. For resource-constrained scenarios or high-dimensional datasets, simplified prompts offer a viable, highly efficient alternative that retains the majority of the method's performance capabilities.

### E.14 TERM CANDIDATE DIVERSITY IN SEARCH

To quantify the exploration capability of the SGED framework, specifically within the stochastic term generation phase of the Monte Carlo Tree Search (MCTS), we analyzed the structural diversity of the candidate basis functions proposed by the LLM. High diversity among candidate terms implies that the search is effectively exploring the hypothesis space rather than collapsing into a narrow mode of similar expressions.

**Methodology.** We define a *Diversity Index* based on the symbolic composition of the proposed terms. For a given candidate term $t$, we parse its abstract syntax tree to extract a "bag-of-symbols" set $S(t)$, comprising all unique variables, operators, functions, and constants utilized in the expression (e.g., $S(3x + \sin(y)) = \{3, x, y, +, \sin\}$).

For a set of unique candidate terms $\mathcal{T}$ generated at a specific search node, we compute the mean pairwise Jaccard similarity $\bar{s}$:

$$\bar{s} = \frac{1}{|\mathcal{P}|} \sum_{(t_i, t_j) \in \mathcal{P}} \frac{|S(t_i) \cap S(t_j)|}{|S(t_i) \cup S(t_j)|}$$

where $\mathcal{P}$ is the set of all unique pairs in $\mathcal{T}$. The *Diversity Index* is defined as $D = 1 - \bar{s}$. A score of $1.0$ indicates disjoint symbol sets (maximum diversity), while $0.0$ indicates identical symbol usage. We also track the *Distinct Ratio*, defined as the number of unique symbolic strings divided by the total number of proposed terms, to measure repetition.

**Results.** We evaluated this metric across all expansion points in the MCTS for two representative dataset: *Lung Cancer (with Chemo. & Radio.)*, low-dimensional, 4 state/action variables, and *RNA Polymerase*, high-dimensional, 263 features. All default SGED (MCTS variant) hyperparameters were used, with early stopping disabled to obtain a larger pool on node expansions. The results are summarized in Table 20.

Table 20: **Candidate Term Diversity Statistics.** Metrics are averaged across all MCTS expansion nodes (mean $\pm$ 95% CI). The *Diversity Index* measures structural variation ($1 - $ Jaccard), while *Distinct Ratio* measures string uniqueness.

| Dataset | Diversity Index ($D$) | Distinct Ratio |
|---|---|---|
| Lung Cancer (with Chemo. & Radio.) | $0.812 \pm 0.020$ | $0.683 \pm 0.088$ |
| RNA Polymerase | $0.987 \pm 0.021$ | $0.384 \pm 0.060$ |

**Analysis.** The *Lung Cancer* dataset exhibits a robust diversity index of approximately $0.81$. Given the small number of available variables in this problem domain, some overlap in symbol usage (e.g., repeated use of the variable `cancer_volume`) is structurally inevitable; essentially, terms must share variables to be relevant. Despite this constraint, the high index confirms that the operators and functional transformations proposed are varied.

In contrast, the *RNA Polymerase* dataset achieves a near-maximal diversity index of $\approx 0.99$. This higher diversity is expected and desirable: the dataset contains 263 input features, allowing the LLM to propose candidate terms involving disjoint subsets of features (e.g., different nucleotide positions or histone markers) with very little symbolic overlap.

These results indicate that the stochastic sampling parameters used in the MCTS induce a high degree of structural variety in the candidate generation phase, ensuring broad coverage of the search space.

## F EVALUATION ON LLM-SRBENCH

**Benchmark Overview:** Existing symbolic regression benchmarks often rely on well-known physical equations (e.g., Feynman benchmark) that are likely present in the pre-training corpora of modern Large Language Models. This allows models to potentially solve problems via memorization rather than true scientific reasoning and data-driven discovery. LLM-SRBench (Shojaee et al., 2025b) addresses this by introducing a suite of 239 challenging problems designed to prevent trivial

memorization. It includes transformed representations of physical models and synthetic, discovery-driven problems. These problems span four distinct scientific domains: Physics, Biology, Material Science, and Chemistry. Evaluating on LLM-SRBench ensures that SGED's performance reflects genuine data-driven discovery rather than the retrieval of memorized equations.

**Experimental Setup:** Due to the significant computational resources and time required to execute multiple iterative, multi-agent LLM-based frameworks across the entire suite of 239 problems, we conducted our evaluation on a representative subset. We randomly selected 3 problems from each of the four scientific domains, resulting in a total of 12 diverse benchmark problems. This subset allows us to gauge performance across different disciplines while maintaining a manageable computational budget.

To ensure a strictly comparable computational footprint across all evaluated methods, we enforced a fixed budget of approximately 300,000 tokens per run, with iteration counts set in excess to ensure this token limit served as the active termination criterion. All experiments were conducted over 10 random seeds using `gpt-4o-mini` (version `2024-07-18`) as the underlying LLM, while all other method-specific hyperparameters were maintained consistent with the rest of the experiments in this work, see Appendix C.5.

**Evaluation Metrics:** To provide a comprehensive view of model performance, we report metrics covering predictive accuracy, out-of-distribution generalization, and symbolic structural correctness.

**Performance Metrics:** We first report standard numerical accuracy metrics:

- *Normalized Mean Squared Error (NMSE)*: We compute this metric as defined in (Shojaee et al., 2025b). It is evaluated for **both** in-distribution (ID) and out-of-distribution (OOD) test sets.

- *Accuracy to Tolerance 0.1 (*$\text{Acc}_{0.1}$*)*: We compute this metric as defined in (Shojaee et al., 2025b), representing the fraction of runs achieving an NMSE below 0.1. This is also evaluated for **both** ID and OOD test sets.

**Symbolic Metrics:** While LLM-SRBench typically utilizes an *LLM-assisted Symbolic Accuracy (SA)* metric, we found this measure insufficient for discerning relative performance in this specific experimental setting. When computed following the benchmark's protocol (using `gpt-4o` as the judge), *all evaluated methods attained a Symbolic Accuracy of 0.0* on this subset within the fixed computational budget. To quantify the symbolic proximity of the discovered equations to the ground truth and distinguish between suboptimal solutions, we instead introduce and report the following additional structural metric:

- *Term Recall*: To evaluate the recovery of underlying dynamics independent of parameter precision, we define Term Recall. Let $E_{gt}$ and $E_{pred}$ be the ground truth and predicted expressions, decomposed into additive term sets. We define a skeletonization mapping, $\phi(\tau)$, which reduces a term $\tau$ to its canonical form by removing top-level scalar coefficients and replacing internal floating-point parameters with unity, while preserving integer exponents that define topology (e.g., $\phi(0.42\sin(1.01t)) \rightarrow \sin(t)$). Denoting the set of skeletonized terms as $S(E) = \{\phi(\tau) \mid \tau \in \text{terms}(E)\}$, the metric is defined as the normalized intersection $\text{TR} = |S(E_{gt}) \cap S(E_{pred})|/|S(E_{gt})|$. This strictly penalizes missing physical terms while remaining robust to numerical approximation.

To enable these structural comparisons across methods with differing output formats – which ranges from raw symbolic strings to executable Python functions or class definitions we implemented a standardization step, where we utilized a high-capability LLM (`GPT-5`) to translate each proposed solution into a canonical, SymPy-parsable format, ensuring a consistent representation for subsequent metric computation.

The aggregate results (average rank) are presented in Table 21, followed by detailed per-problem results in Tables 22 through 33.

Table 21: Average method ranks across LLM-SRBench problems tested (lower is better). Best result shown in **bold**, second best result shown underlined. Note that the Symbolic Accuracy metric (Shojaee et al., 2025b) was a *tie* at the value 0.0 across all methods tested.

| Method | ID Test Set NMSE | ID Test Set $\text{Acc}_{0.1}$ | OOD Test Set NMSE | OOD Test Set $\text{Acc}_{0.1}$ | Symbolic Metrics Term Recall |
|---|---|---|---|---|---|
| SGED | 2.50 | 2.12 | 2.00 | 2.25 | **1.92** |
| SGED (top-K) | **1.08** | **1.58** | **1.42** | **1.88** | 2.12 |
| ICSR | 3.42 | 3.75 | 3.25 | 3.42 | 4.46 |
| LLM-SR | 3.00 | 3.17 | 3.58 | 3.25 | 3.71 |
| D3-white-box | 5.00 | 4.38 | 4.75 | 4.21 | 2.79 |

On the numerical performance benchmarks, SGED and its non-LLM pruning variant consistently outperform all baseline methods across both In-Distribution (ID) and Out-Of-Distribution (OOD) evaluations. As shown in Table 21, the SGED variants secure the top two average ranks for both NMSE and $\text{Acc}_{0.1}$. Crucially, this performance advantage is maintained on the OOD test sets, where SGED and its variant achieve average NMSE ranks of 2.00 and 1.42 respectively, significantly surpassing the best-performing baseline (Rank 3.25). This robust generalization capability indicates that the models discovered by SGED are not merely overfitting to the training domain.

In terms of symbolic recovery, SGED and its non-LLM pruning variant achieve the highest performance on Term Recall (Avg. Rank 1.92 and 2.12), demonstrating a superior ability to correctly identify the essential basis functions governing the ground truth dynamics. While the stricter Symbolic Accuracy metric yielded 0.0 across all evaluated methods, reflecting the challenge of exact skeletal recovery under these constraints, the strong Term Recall results validate that SGED successfully isolates the correct mechanistic components.

Table 22: LLM-SRBench results for BPG16. Best metric in **bold**, overlapping 95% CIs in gray.

| Method | ID Test Set NMSE ↓ | ID Test Set $\text{Acc}_{0.1}$ ↑ | OOD Test Set NMSE ↓ | OOD Test Set $\text{Acc}_{0.1}$ ↑ | Symbolic Metrics Symbolic Accuracy ↑ | Term Recall ↑ |
|---|---|---|---|---|---|---|
| SGED | 5.84e-07 ± 1.62e-07 | **1** ± 0 | 0.0234 ± 0.00913 | **1** ± 0 | 0.000 ± 0.000 | **0.450** ± 0.113 |
| SGED (top-K) | **2.84e-07 ± 9.83e-08** | 1 ± 0 | **0.0066** ± 0.00637 | 1 ± 0 | 0.000 ± 0.000 | 0.450 ± 0.113 |
| ICSR | 0.0026 ± 0.00234 | 0.6 ± 0.369 | 10.3 ± 10.9 | 0.5 ± 0.377 | 0.000 ± 0.000 | 0.050 ± 0.113 |
| LLM-SR | 0.23 ± 0.211 | 0.1 ± 0.226 | 236 ± 145 | 0.2 ± 0.302 | 0.000 ± 0.000 | 0.150 ± 0.173 |
| D3-white-box | 1.19 ± 0.957 | 0 ± 0 | 775 ± 363 | 0 ± 0 | 0.000 ± 0.000 | 0.200 ± 0.185 |

Table 23: LLM-SRBench results for BPG8. Best metric in **bold**, overlapping 95% CIs in gray.

| Method | ID Test Set NMSE ↓ | ID Test Set $\text{Acc}_{0.1}$ ↑ | OOD Test Set NMSE ↓ | OOD Test Set $\text{Acc}_{0.1}$ ↑ | Symbolic Metrics Symbolic Accuracy ↑ | Term Recall ↑ |
|---|---|---|---|---|---|---|
| SGED | 7.17e-09 ± 7.76e-09 | **1** ± 0 | 2.11e-06 ± 2.43e-06 | **1** ± 0 | 0.000 ± 0.000 | 0.300 ± 0.075 |
| SGED (top-K) | **3.75e-11** ± 3.96e-11 | 1 ± 0 | 7.39e-09 ± 2.13e-09 | 1 ± 0 | 0.000 ± 0.000 | 0.300 ± 0.075 |
| ICSR | 0.00139 ± 0.000944 | 0 ± 0 | 10 ± 15.5 | 0 ± 0 | 0.000 ± 0.000 | 0.100 ± 0.115 |
| LLM-SR | 0.00757 ± 0.00889 | 0.1 ± 0.226 | 13 ± 24.5 | 0.1 ± 0.226 | 0.000 ± 0.000 | 0.200 ± 0.123 |
| D3-white-box | 0.459 ± 0.371 | 0 ± 0 | 245 ± 307 | 0 ± 0 | 0.000 ± 0.000 | **0.333** ± 0.000 |

Table 24: LLM-SRBench results for BPG9. Best metric in **bold**, overlapping 95% CIs in gray.

| Method | ID Test Set NMSE ↓ | ID Test Set $\text{Acc}_{0.1}$ ↑ | OOD Test Set NMSE ↓ | OOD Test Set $\text{Acc}_{0.1}$ ↑ | Symbolic Metrics Symbolic Accuracy ↑ | Term Recall ↑ |
|---|---|---|---|---|---|---|
| SGED | 2.82e-09 ± 3.17e-09 | **1** ± 0 | 6.65e-06 ± 7.85e-06 | **1** ± 0 | 0.000 ± 0.000 | **0.667** ± 0.000 |
| SGED (top-K) | **9.83e-11** ± 1.48e-10 | 1 ± 0 | **8.8e-09** ± 4.4e-09 | 1 ± 0 | 0.000 ± 0.000 | 0.667 ± 0.000 |
| ICSR | 0.00049 ± 0.000424 | 0 ± 0 | 0.944 ± 0.58 | 0 ± 0 | 0.000 ± 0.000 | 0.000 ± 0.000 |
| LLM-SR | 0.304 ± 0.229 | 0.1 ± 0.226 | 241 ± 183 | 0.1 ± 0.226 | 0.000 ± 0.000 | 0.533 ± 0.201 |
| D3-white-box | 0.305 ± 0.35 | 0.1 ± 0.226 | 261 ± 303 | 0.1 ± 0.226 | 0.000 ± 0.000 | 0.433 ± 0.161 |

Table 25: LLM-SRBench results for CRK10. Best metric in **bold**, overlapping 95% CIs in gray.

| Method | ID Test Set | | OOD Test Set | | Symbolic Metrics | |
| | NMSE ↓ | $Acc_{0.1}$ ↑ | NMSE ↓ | $Acc_{0.1}$ ↑ | Symbolic Accuracy ↑ | Term Recall ↑ |
|---|---|---|---|---|---|---|
| SGED | 9.66e-09 ± 2.07e-08 | 0.9 ± 0.226 | 8.88e-05 ± 0.000195 | 0.9 ± 0.226 | 0.000 ± 0.000 | **0.400** ± 0.151 |
| SGED (top-K) | **9.85e-12** ± 9.5e-12 | **1** ± 0 | **1.62e-07** ± 2.54e-07 | **1** ± 0 | 0.000 ± 0.000 | 0.200 ± 0.185 |
| ICSR | 0.00016 ± 0.000152 | 0 ± 0 | 14.9 ± 10 | 0 ± 0 | 0.000 ± 0.000 | 0.000 ± 0.000 |
| LLM-SR | 2.9e-07 ± 7.35e-08 | 0.1 ± 0.226 | 0.00352 ± 0.00281 | 0 ± 0 | 0.000 ± 0.000 | 0.000 ± 0.000 |
| D3-white-box | 0.719 ± 0.252 | 0 ± 0 | 5.27e+04 ± 1.98e+04 | 0 ± 0 | 0.000 ± 0.000 | 0.000 ± 0.000 |

Table 26: LLM-SRBench results for CRK28. Best metric in **bold**, overlapping 95% CIs in gray.

| Method | ID Test Set | | OOD Test Set | | Symbolic Metrics | |
| | NMSE ↓ | $Acc_{0.1}$ ↑ | NMSE ↓ | $Acc_{0.1}$ ↑ | Symbolic Accuracy ↑ | Term Recall ↑ |
|---|---|---|---|---|---|---|
| SGED | 0.00184 ± 0.00233 | 0 ± 0 | 4.69e+03 ± 6.66e+03 | 0 ± 0 | 0.000 ± 0.000 | **0.467** ± 0.123 |
| SGED (top-K) | **0.000121** ± 7.76e-05 | 0.1 ± 0.226 | 263 ± 388 | 0 ± 0 | 0.000 ± 0.000 | 0.467 ± 0.123 |
| ICSR | 0.000825 ± 0.000846 | 0.1 ± 0.226 | **73.3** ± 57.9 | 0 ± 0 | 0.000 ± 0.000 | 0.200 ± 0.123 |
| LLM-SR | 0.00045 ± 0.000405 | 0 ± 0 | 150 ± 249 | 0 ± 0 | 0.000 ± 0.000 | 0.300 ± 0.176 |
| D3-white-box | 0.741 ± 0.179 | 0 ± 0 | 8.59e+04 ± 5.24e+04 | 0 ± 0 | 0.000 ± 0.000 | 0.433 ± 0.115 |

Table 27: LLM-SRBench results for CRK36. Best metric in **bold**, overlapping 95% CIs in gray.

| Method | ID Test Set | | OOD Test Set | | Symbolic Metrics | |
| | NMSE ↓ | $Acc_{0.1}$ ↑ | NMSE ↓ | $Acc_{0.1}$ ↑ | Symbolic Accuracy ↑ | Term Recall ↑ |
|---|---|---|---|---|---|---|
| SGED | 4.65e-06 ± 4.11e-06 | 0.6 ± 0.369 | 0.000316 ± 0.000472 | 1 ± 0 | 0.000 ± 0.000 | 0.650 ± 0.173 |
| SGED (top-K) | **4.41e-07** ± 5.07e-07 | **0.8** ± 0.302 | **3.68e-05** ± 4.42e-05 | 1 ± 0 | 0.000 ± 0.000 | **0.700** ± 0.302 |
| ICSR | 0.00156 ± 0.000511 | 0.1 ± 0.226 | 0.0141 ± 0.00954 | 1 ± 0 | 0.000 ± 0.000 | 0.100 ± 0.151 |
| LLM-SR | 0.000106 ± 0.000138 | 0.1 ± 0.226 | 0.0595 ± 0.0901 | 0.9 ± 0.226 | 0.000 ± 0.000 | 0.300 ± 0.185 |
| D3-white-box | 0.0148 ± 0.00403 | 0 ± 0 | 0.181 ± 0.233 | 0.8 ± 0.302 | 0.000 ± 0.000 | 0.500 ± 0.000 |

Table 28: LLM-SRBench results for MatSci20. Best metric in **bold**, overlapping 95% CIs in gray.

| Method | ID Test Set | | OOD Test Set | | Symbolic Metrics | |
| | NMSE ↓ | $Acc_{0.1}$ ↑ | NMSE ↓ | $Acc_{0.1}$ ↑ | Symbolic Accuracy ↑ | Term Recall ↑ |
|---|---|---|---|---|---|---|
| SGED | 7.36e-06 ± 1.29e-05 | **1** ± 0 | 0.00179 ± 0.00315 | **1** ± 0 | 0.000 ± 0.000 | 0.325 ± 0.086 |
| SGED (top-K) | **2.87e-10** ± 4.24e-10 | 1 ± 0 | **3.05e-07** ± 4.57e-07 | 1 ± 0 | 0.000 ± 0.000 | 0.200 ± 0.075 |
| ICSR | 0.000325 ± 0.000238 | 0.1 ± 0.226 | 0.0459 ± 0.0304 | 1 ± 0 | 0.000 ± 0.000 | 0.250 ± 0.000 |
| LLM-SR | 4.28e-08 ± 7.05e-08 | 1 ± 0 | 1.1e-05 ± 1.96e-05 | 1 ± 0 | 0.000 ± 0.000 | 0.300 ± 0.141 |
| D3-white-box | 0.701 ± 0.114 | 0 ± 0 | 23.2 ± 2.96 | 0 ± 0 | 0.000 ± 0.000 | **0.475** ± 0.057 |

Table 29: LLM-SRBench results for MatSci22. Best metric in **bold**, overlapping 95% CIs in gray.

| Method | ID Test Set | | OOD Test Set | | Symbolic Metrics | |
| | NMSE ↓ | $Acc_{0.1}$ ↑ | NMSE ↓ | $Acc_{0.1}$ ↑ | Symbolic Accuracy ↑ | Term Recall ↑ |
|---|---|---|---|---|---|---|
| SGED | 1.37e-06 ± 9.75e-07 | **1** ± 0 | **0.000184** ± 0.000146 | **1** ± 0 | 0.000 ± 0.000 | 0.250 ± 0.000 |
| SGED (top-K) | **7.65e-07** ± 5.58e-07 | 1 ± 0 | 0.000248 ± 0.000231 | 1 ± 0 | 0.000 ± 0.000 | 0.250 ± 0.000 |
| ICSR | 0.00167 ± 0.000915 | 0.1 ± 0.226 | 0.792 ± 0.493 | 1 ± 0 | 0.000 ± 0.000 | 0.250 ± 0.000 |
| LLM-SR | 1.59e-06 ± 1.25e-06 | 1 ± 0 | 0.000898 ± 0.000658 | 1 ± 0 | 0.000 ± 0.000 | 0.200 ± 0.141 |
| D3-white-box | 0.0852 ± 0.0217 | 0 ± 0 | 13.7 ± 3.67 | 0.1 ± 0.226 | 0.000 ± 0.000 | **0.475 ± 0.057** |

Table 30: LLM-SRBench results for MatSci4. Best metric in **bold**, overlapping 95% CIs in gray.

| Method | ID Test Set | | OOD Test Set | | Symbolic Metrics | |
| | NMSE ↓ | $Acc_{0.1}$ ↑ | NMSE ↓ | $Acc_{0.1}$ ↑ | Symbolic Accuracy ↑ | Term Recall ↑ |
|---|---|---|---|---|---|---|
| SGED | 0.000384 ± 0.000173 | **1** ± 0 | 0.0346 ± 0.0308 | **1** ± 0 | 0.000 ± 0.000 | 0.000 ± 0.000 |
| SGED (top-K) | 0.000338 ± 0.000104 | 1 ± 0 | **0.0177** ± 0.00821 | 1 ± 0 | 0.000 ± 0.000 | 0.000 ± 0.000 |
| ICSR | 0.0065 ± 0.00606 | 0.8 ± 0.302 | 0.622 ± 0.617 | 0.7 ± 0.346 | 0.000 ± 0.000 | 0.000 ± 0.000 |
| LLM-SR | **0.000326** ± 0.000175 | 1 ± 0 | 0.0481 ± 0.0338 | 1 ± 0 | 0.000 ± 0.000 | 0.000 ± 0.000 |
| D3-white-box | 0.169 ± 0.0152 | 0 ± 0 | 12.5 ± 0.896 | 0 ± 0 | 0.000 ± 0.000 | 0.000 ± 0.000 |

Table 31: LLM-SRBench results for PO2. Best metric in **bold**, overlapping 95% CIs in gray .

| Method | ID Test Set | | OOD Test Set | | Symbolic Metrics | |
|---|---|---|---|---|---|---|
| | NMSE ↓ | Acc$_{0.1}$ ↑ | NMSE ↓ | Acc$_{0.1}$ ↑ | Symbolic Accuracy ↑ | Term Recall ↑ |
| SGED | $0.000125 \pm 9.26$e-05 | $0.1 \pm 0.226$ | $0.000357 \pm 0.000217$ | $0 \pm 0$ | $0.000 \pm 0.000$ | $0.633 \pm 0.075$ |
| SGED (top-K) | **4.32e-05** $\pm$ 3.06e-05 | **0.6** $\pm 0.369$ | **0.000137** $\pm$ 9.36e-05 | $0.3 \pm 0.346$ | $0.000 \pm 0.000$ | **0.667** $\pm 0.000$ |
| ICSR | $0.00127 \pm 0.000296$ | $0 \pm 0$ | $0.00232 \pm 0.000515$ | $0 \pm 0$ | $0.000 \pm 0.000$ | $0.267 \pm 0.101$ |
| LLM-SR | $0.00784 \pm 0.00508$ | $0 \pm 0$ | $0.0291 \pm 0.0231$ | $0 \pm 0$ | $0.000 \pm 0.000$ | $0.333 \pm 0.000$ |
| D3-white-box | $0.0442 \pm 0.00944$ | $0 \pm 0$ | $0.192 \pm 0.0458$ | $0 \pm 0$ | $0.000 \pm 0.000$ | $0.367 \pm 0.075$ |

Table 32: LLM-SRBench results for PO21. Best metric in **bold**, overlapping 95% CIs in gray .

| Method | ID Test Set | | OOD Test Set | | Symbolic Metrics | |
|---|---|---|---|---|---|---|
| | NMSE ↓ | Acc$_{0.1}$ ↑ | NMSE ↓ | Acc$_{0.1}$ ↑ | Symbolic Accuracy ↑ | Term Recall ↑ |
| SGED | $0.000595 \pm 0.000195$ | $0.1 \pm 0.226$ | **0.0667** $\pm 0.0323$ | $0.2 \pm 0.302$ | $0.000 \pm 0.000$ | $0.240 \pm 0.113$ |
| SGED (top-K) | **0.000266** $\pm 0.000153$ | $0.2 \pm 0.302$ | $4.77$e+04 $\pm 1.08$e+05 | **0.5** $\pm 0.377$ | $0.000 \pm 0.000$ | **0.300** $\pm 0.075$ |
| ICSR | $0.0113 \pm 0.0207$ | $0 \pm 0$ | $1.91$e+07 $\pm 2.81$e+07 | $0 \pm 0$ | $0.000 \pm 0.000$ | $0.100 \pm 0.075$ |
| LLM-SR | $0.00104 \pm 0.000261$ | $0 \pm 0$ | $4.06$e+07 $\pm 4.34$e+07 | $0 \pm 0$ | $0.000 \pm 0.000$ | $0.180 \pm 0.045$ |
| D3-white-box | $0.0375 \pm 0.000124$ | $0 \pm 0$ | $9.75$e+06 $\pm 9.25$e+06 | $0 \pm 0$ | $0.000 \pm 0.000$ | $0.200 \pm 0.000$ |

Table 33: LLM-SRBench results for PO39. Best metric in **bold**, overlapping 95% CIs in gray .

| Method | ID Test Set | | OOD Test Set | | Symbolic Metrics | |
|---|---|---|---|---|---|---|
| | NMSE ↓ | Acc$_{0.1}$ ↑ | NMSE ↓ | Acc$_{0.1}$ ↑ | Symbolic Accuracy ↑ | Term Recall ↑ |
| SGED | $0.00298 \pm 0.00157$ | $0 \pm 0$ | **0.0708** $\pm 0.086$ | $0 \pm 0$ | $0.000 \pm 0.000$ | $0.067 \pm 0.101$ |
| SGED (top-K) | **0.000758** $\pm 0.000116$ | $0 \pm 0$ | $0.0892 \pm 0.0392$ | $0 \pm 0$ | $0.000 \pm 0.000$ | $0.033 \pm 0.075$ |
| ICSR | $0.000982 \pm 7.48$e-05 | $0 \pm 0$ | $0.187 \pm 0.232$ | $0 \pm 0$ | $0.000 \pm 0.000$ | $0.000 \pm 0.000$ |
| LLM-SR | $0.00127 \pm 0.000767$ | $0 \pm 0$ | $1.15 \pm 2.2$ | $0 \pm 0$ | $0.000 \pm 0.000$ | $0.033 \pm 0.075$ |
| D3-white-box | $0.0643 \pm 0.000239$ | $0 \pm 0$ | $0.191 \pm 0.00284$ | $0 \pm 0$ | $0.000 \pm 0.000$ | $0.000 \pm 0.000$ |

**Complexity-Accuracy Trade-off Analysis.** To synthesize the relationship between symbolic reconstruction and predictive generalization, we visualize the trade-off between structural fidelity (Term Recall) and out-of-distribution accuracy (OOD NMSE) in Figure 11. In this analysis, the ideal model resides in the top-right quadrant, characterized by high term recall (recovering the correct physical terms) and low prediction error.

The Pareto frontiers, color-coded by scientific domain, trace the optimal methods that minimize error for a given level of symbolic recovery. We observe that *SGED* and the *SGED (top-K)* variant consistently populate the Pareto frontier across the Physics, Biology, and Chemistry domains. This indicates that SGED effectively balances the discovery of mechanistic terms with numerical precision. A notable exception is observed in the Material Science domain, where the *D3-white-box* baseline defines the frontier. Qualitative inspection suggests that while D3 successfully retains the intact functional forms of domain-specific terms (e.g., shifted dependencies like $\beta(T - T_0)$ or Arrhenius-type rates), the standard SGED variant tends to approximate these dynamics through linear expansions or alternative basis combinations. Consequently, this fragmentation of the ground-truth structure results in a lower Term Recall score, even when predictive accuracy is maintained. We anticipate that our **SGED-TLO** variant (detailed in Appendix E.10), which is explicitly designed to identify and optimize such internal constants, would bridge this structural gap; however, it was not deployed in this specific benchmark evaluation.

# G  COMPUTATIONAL RESOURCES

The computational experiments for this research were conducted using a combination of cloud-based services for Large Language Model (LLM) inference and local or server-based machines for model optimization and equation evaluation.

**LLM Inference:** The inference for the majority of the Large Language Models employed in this study was performed using the serverless API provided by Azure AI Foundry. This allowed for scalable access to various proprietary LLM endpoints.

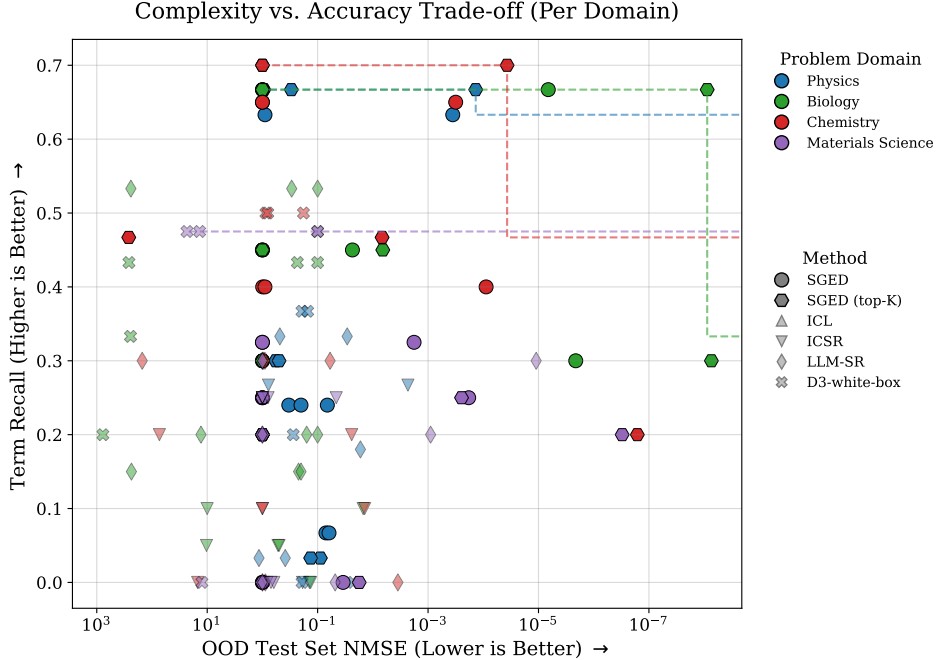

Figure 11: **Pareto Analysis of Structural Fidelity vs. Predictive Accuracy on LLM-SRBench.**
We plot the Out-of-Distribution (OOD) NMSE (log scale, lower is better, axis reversed such that
lower values are to the right) against Term Recall (higher is better) for all method-problem pairs.
Markers distinguish the discovery methods, while colors distinguish the scientific domains (Physics,
Biology, Material Science, Chemistry). The dashed lines of matching color represent the Pareto
frontier for each domain. SGED and SGED (top-K) lie on or near the frontier for Physics, Biology,
and Chemistry, indicating a robust balance between capturing ground-truth dynamics and minimizing
generalization error. In Material Science, D3-white-box demonstrates superior performance, defining
the frontier for that domain.

For the `Llama-3.3-70B` model, inference was conducted on a dedicated Azure Virtual Machine,
specifically a Standard_NC96ads_A100_v4 instance. Key specifications of this VM include:

- **Processor:** 96 non-multithreaded 3rd Gen AMD EPYC™ 7V13 (Milan) cores.
- **GPU Accelerators:** 4 NVIDIA A100 PCIe GPUs, each with 80GB of memory.
- **System Memory:** 880 GiB.

**Model Optimization and Equation Evaluation:** The optimization of model parameters (i.e., fitting
the linear model $\mathbf{w}$ for basis functions $\psi_j(\mathbf{x})$) and the evaluation of equations were carried out on the
following types of local workstations and servers:

- A workstation equipped with a 10-core Intel Core i9-10900K CPU and 64 GiB of RAM.
- A server equipped with an 80-Core AMD EPYC 9V84 CPU and approximately 630 GiB of
  RAM.

It is important to note that the optimization step of SGED, which involves fitting a linear model (we
use the `scikit-learn` library's implementation), does not require GPU acceleration and can be
efficiently performed on any reasonably powerful desktop computer.

**Execution Time:** The time of execution for SGED varied depending on several factors, including the
complexity of the dataset, the number of terms explored, the depth and breadth of the Monte Carlo
Tree Search (if utilized), and the response latency of the LLM APIs. Individual propose-and-prune
cycles involving LLM calls typically took seconds to minutes, while full MCTS runs could extend to
several hours for comprehensive exploration. No single run took longer than 3 hours total wall-clock
time. Fitting the linear models for equation evaluation was generally swift, on the order of seconds.

The total compute resources for the entire research project, including preliminary experiments and hyperparameter tuning, naturally exceeded that of the final reported experimental runs. However, the resources outlined above are representative of those required to reproduce the main findings.

## H  COMPUTATIONAL COST AND SCALABILITY ANALYSIS

To address the practical feasibility of SGED, we conducted a comprehensive analysis of its computational cost and scalability.

### H.1  COST AND WALL-CLOCK TIME COMPARISON

We benchmarked SGED's monetary cost (in USD, based on LLM API calls) and total wall-clock time against several baseline methods on the Lung Cancer (with Chemo. & Radio.) dataset. Cost is based on GPT-4o API pricing at the time of experimentation: $0.000005 per input token and $0.00002 per output token. The total cost and wall clock results, averaged over 5 seeds, are shown in Table 34, as well as the Test MSE taken from the Table 3 main experiment results where available. Hyperparameters used were the same as in the Table 3 experiment. Results for the non-MCTS (iterative) variant of SGED are also included as "SGED (Iterative)", with all hyperparameters matching the MCTS variant "SGED (Ours)", except the tree search being disabled and total number of iterative iterations being set to 30.

SGED's runtime is competitive with other modern symbolic regression methods and significantly faster than computationally intensive approaches like LaSR, while achieving a substantially lower Test MSE. For results under a comparable LLM token budget, see Appendix H.3, which makes the per-token performance efficiency of SGED especially clear.

Table 34: **Computational Cost and Performance Comparison.** Benchmarks on the Lung Cancer (with Chemo. & Radio.) dataset. Cost is based on GPT-4o API pricing at the time of experimentation.

| Method | Total Cost (USD) ↓ | Total Tokens ↓ | Wall Clock (s) ↓ | LLM Calls Wall Clock (s) ↓ | LLM Call Count ↓ | Test MSE ↓ |
|---|---|---|---|---|---|---|
| GPLearn | - | - | $92.2 \pm 5.8$ | - | - | $46.8 \pm 4.9$ |
| PySR (Cramner, 2023) | - | - | $282.7 \pm 6.0$ | - | - | $0.399 \pm 0.123$ |
| D3-white-box (Holt et al., 2024b) | $0.589 \pm 0.138$ | $262,636 \pm 2,406$ | $203.4 \pm 39.5$ | $178.6 \pm 23.8$ | $23.4 \pm 6.2$ | $253 \pm 273$ |
| LaSR (Grayeli et al., 2024) | $72.40 \pm 3.48$ | $5,171,391 \pm 238,935$ | $1444.5 \pm 137.5$ | - [†] | - [†] | $3.97 \pm 3.21$ |
| ICSR (Merler et al., 2024) | $1.08 \pm 0.16$ | $261,989 \pm 71,813$ | $774.2 \pm 140.6$ | $392.6 \pm 117.5$ | $80.0 \pm 0.0$ | $6.1 \pm 1.05$ |
| LLM-SR (Shojaee et al., 2025a) | $0.80 \pm 0.02$ | $61,905 \pm 2,220$ | $371.7 \pm 18.7$ | $355.1 \pm 19.2$ | $32.0 \pm 0.0$ | $32.1 \pm 48.4$ |
| SGED (Iterative) | $1.12 \pm 0.32$ | $191,026 \pm 60,666$ | $159.4 \pm 33.6$ | $155.7 \pm 32.7$ | $42.2 \pm 9.4$ | $3.29 \pm 4.16$ |
| **SGED (Ours)** | $1.87 \pm 0.71$ | $272,743 \pm 111,859$ | $382.8 \pm 137.6$ | $376.8 \pm 135.2$ | $93.4 \pm 17.7$ | $\mathbf{0.052 \pm 0.018}$ |

[†] Due to implementational difficulties, LaSR being implemented in Python *via Julia*, while we were able to obtain *"Total Tokens"* from the logs, we were unable to measure *"LLM Call Wall Clock (s)"* or *"LLM Call Count"* for this method.

### H.2  SCALABILITY WITH HIGH-DIMENSIONAL INPUTS

A key concern for symbolic regression methods is scalability with a large number of input features. We evaluated SGED's runtime on the RNA Polymerase dataset, which has 263 features. All hyperparameters were kept the same as in the Table 3 main experiment. As shown in Figure 12, SGED's runtime scales exceptionally well. The total wall-clock time increases only sub-linearly as the feature count grows from 10 to 263. This is a significant advantage of our framework, as the influence-based feedback allows the LLM to efficiently identify and prune irrelevant features, avoiding the combinatorial explosion that challenges many other methods. This demonstrates SGED's practical feasibility for large-scale, high-dimensional scientific problems.

### H.3  PERFORMANCE UNDER A FIXED COMPUTATIONAL BUDGET

To provide a direct and fair comparison of computational efficiency, we conducted an experiment where all LLM-based baselines were benchmarked under a fixed computational budget with respect to LLM usage – as the LLM calls are the most computationally intensive aspect of these methods. In order to ensure the methods consumed a nearly identical number of total LLM tokens (approximately 265k) we either adjusted the hyperparameters controlling itrative execution length of the method (number of iterations / generations or equivalent), or added early stopping based on the cumulative

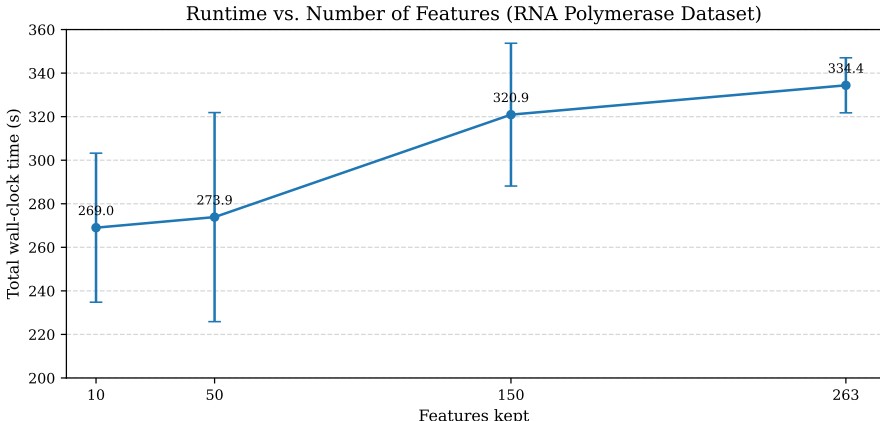

Figure 12: **Scalability with Increasing Number of Features.** Mean total wall-clock time on the RNA Polymerase dataset versus number of features used, averaged over 10 seeds. Points indicate means; error bars denote 95% confidence-interval half-widths. Runtime increases sub-linearly as the feature count grows from 10 to 263, indicating strong scalability of SGED.

number of tokens consumed in the experiment run. All other hyperparameters were unchanged from the Table 3 main experiment setting. The Lung Cancer (with Chemo. & Radio.) dataset was used and the results were averaged over 5 seeds.

The results, presented in Table 35, are conclusive. When the LLM computational resources are held equal, SGED's performance advantage is clear. It achieves an MSE that is nearly 50 times better than the next-best performing baseline method (ICSR). This demonstrates that SGED's architecture uses its computational budget more efficiently to find a superior solutions. While some simpler methods (like ICL with Basic Feedback) have a faster wall-clock time, they produce substantially less accurate results. SGED strikes an optimal balance, achieving a competitive runtime while delivering superior final equation(s).

Table 35: **Performance Comparison Under a Fixed Token Budget ($\approx$265k Tokens).** Results on the Lung Cancer (with Chemo. & Radio.) dataset, averaged over 5 seeds.

| Method | Test MSE ↓ | Total Tokens | Wall Clock (s) | LLM Call Count |
|---|---|---|---|---|
| ICL (Basic Feedback) | $16.04 \pm 22.1$ | $265{,}199 \pm 8{,}729$ | $82.4 \pm 3.8$ | $36.5 \pm 1.1$ |
| D3-white-box (Holt et al., 2024b) | $809.7 \pm 1754.4$ | $262{,}636 \pm 2{,}406$ | $302.1 \pm 33.3$ | $40.2 \pm 0.7$ |
| LaSR (Grayeli et al., 2024) | $101.3 \pm 112.1$ | $272{,}606 \pm 15{,}209$ | $409.3 \pm 28.1$ | - [†] |
| ICSR (Merler et al., 2024) | $2.66 \pm 2.11$ | $261{,}989 \pm 71{,}813$ | $1031.5 \pm 319.3$ | $146.0 \pm 0.0$ |
| LLM-SR (Shojaee et al., 2025a) | $9.88 \pm 20.1$ | $259{,}639 \pm 12{,}820$ | $1208.2 \pm 58.1$ | $116.0 \pm 0.0$ |
| SGED (Iterative) | $0.65 \pm 1.62$ | $263{,}308 \pm 1{,}180$ | $177.0 \pm 11.6$ | $58.1 \pm 1.7$ |
| **SGED (Ours)** | $\mathbf{0.054 \pm 0.074}$ | $272{,}743 \pm 111{,}859$ | $382.8 \pm 137.6$ | $105.5 \pm 3.7$ |

[†] Due to implementational difficulties, LaSR being implemented in Python *via Julia*, while we were able to obtain *"Total Tokens"* from the logs, we were unable to measure *"LLM Call Count"* for this method.

# I   REPRODUCIBILITY AND SETTINGS

To ensure reproducibility of the results, we detail the default hyperparameters and configuration settings used for the Structure Guided Equation Discovery (SGED) framework and the baseline methods. Unless otherwise specified in the specific experimental sections, these default values were maintained throughout the experiments reported in this paper.

SGED DEFAULT HYPERPARAMETERS

The SGED framework involves parameters governing the LLM interaction, the linear model fitting, and the Monte Carlo Tree Search (MCTS) strategy.

**LLM Agents:**

- **Model Version:** `gpt-4o` (version `2024-11-20`) was used as the backbone for both the "Propose" and "Prune" agents, unless otherwise noted in the LLM sensitivity analysis.
- **Inference parameters:** *Temperature* set to default 1.0 to allow for stochastic diversity in term proposal; all others kept at their OpenAI defaults.

**Term Generation and Pruning (Propose-and-Prune Cycle):**

- `terms_per_round`: 5. The number of new basis functions the LLM is prompted to suggest in standard iterations.
- `first_round_n_candidates`: 10. The number of terms requested in the initial iteration to seed the search.
- `keep_n_terms`: 6. The maximum number of terms retained after the pruning phase to ensure equation parsimony.
- `linear_model_type`: OLS (Ordinary Least Squares). Used for fitting weights $\mathbf{w}$ and calculating influence scores.
- `influence_calculation`: No-refit. Influence scores $\Delta_j$ are calculated on the validation set without refitting the model for every term removal, ensuring computational efficiency.

**Search Strategy (MCTS):**

- `total_budget`: 30. The total number of node expansions (iterations) allowed before termination.
- `n_successors`: 5. The number of distinct child nodes (variations) generated from a parent node during expansion.
- `exploration_constant` ($c$): $\sqrt{2}$. The constant used in the UCT formula to balance exploration and exploitation.
- `depth_limit`: 10. The maximum depth of the search tree.
- `rollout_is_just_node_reward`: True. We utilize Heuristic MCTS where the node's immediate validation MSE serves as the reward, without performing deep rollouts.

**SGED Variants:**

Throughout the paper and appendices, distinct variations of the SGED framework are evaluated to assess specific capabilities or ablations. The key variants are defined in the following sections:

- **SGED (Default / MCTS):** The primary configuration utilizing Monte Carlo Tree Search with influence-based feedback. See Section 2 and Appendix B.3.
- **SGED (Iterative):** A linear iterative refinement strategy where the search tree is replaced by a single refinement loop. Maximum number of iterations in this variant (`n_iters`) is set to 30, matching `total_budget` of the MCTS variant. All other hyperparameters match the MCTS variant. See Section 2.2 and Appendix B.
- **SGED-TLO (Term-Local Optimization):** An extension allowing for the optimization of internal scalar constants within proposed basis functions. See Appendix E.10.
- **SGED (Algorithmic Pruning / top-K):** A variant replacing the LLM "Prune" agent with a deterministic selection mechanism based on influence scores. See Appendix J.

BASELINE METHOD HYPERPARAMETERS

We compared SGED against several white-box and black-box baselines. Below are the specific settings used to ensure fair comparison, particularly regarding model complexity (parsimony). All LLM-based methods use `gpt-4o` (version `2024-11-20`), unless stated otherwise in the particular experiment section.

**GPLearn** (Stephens, 2015):

- `population_size`: 1000.
- `generations`: 30.
- `parsimony_coefficient`: 1.0. Selected to penalize complexity and encourage concise equations comparable to SGED.
- The remaining hyperparameters were set to the default values provided in the `gplearn` library.

**PySR** (Cranmer, 2023):

- `model_selection`: "best".
- `niterations`: 40 (default for version 0.19).
- Library version: 0.19, as bundled with the LaSR implementation[6].
- The remaining hyperparameters were set to the default values of the PySR 0.19 package.

**LLM-based Baselines (ZeroShot, ZeroOptim, ICL - Basic Feedback):**

- `max_terms`: 6. Constrained via prompting to match SGED's parsimony.
- `iterations`: 30 (for ICL - Basic Feedback). Matches the SGED search budget.

**D3 (white-box)** (Holt et al., 2024b):

- `max_features`: 6. Constrained to match SGED's parsimony.
- The remaining hyperparameters were set to the default values as described in (Holt et al., 2024b).

**LaSR** (Grayeli et al., 2024):

- `model_selection`: "best".
- `maxsize`: 30. Set to allow equations of sufficient complexity (at least as complex as the 6-term limit used elsewhere).
- `populations`: 15.
- `cycles`: 75 (`niterations` (5) × `populations` (15)).
- The remaining hyperparameters were set to the default values as in LaSR implementation[6] (Grayeli et al., 2024).

**LLM-SR** (Shojaee et al., 2025a):

- `max_nparams`: 8.
- `global_max_sample_num`: 30.
- The remaining hyperparameters were set to the default values as described in (Shojaee et al., 2025a).

**ICSR** (Merler et al., 2024):

- `max_nodes`: 30. Set to allow equations of sufficient complexity.
- `iterations`: 30.
- The remaining hyperparameters were set to the default values as described in (Merler et al., 2024).

**DyNODE, SINDy, RNN, Transformer**:

- Implementation and hyperparameters follow (Holt et al., 2024b).

**XGBoost**:

- All default hyperparameters of the `xgboost` library were used.

OTHER NOTES ON EXPERIMENTAL SETUP

Finally, please note below any additional details on the experimental setup:

---

[6] https://github.com/trishullab/LibraryAugmentedSymbolicRegression.jl

- **Multi-target Regression:** For baseline methods that do not natively support multi-target regression, a separate independent model was instantiated and fitted for each target variable.
- **Model Selection and Data Splits:** Where a method supports early stopping, the validation set was utilized to determine the optimal stopping point to prevent overfitting. For detailed information regarding the specific breakdown of training, validation, and test sets for each benchmark dataset, please refer to Appendix C.

## J   SGED WITH ALGORITHMIC PRUNING

While the primary formulation of Structure Guided Equation Discovery (SGED) employs a dual-agent architecture – using a "Propose" LLM to generate terms and a "Prune" LLM to refine them – the framework is inherently modular. The "Prune" agent is tasked with interpreting the per-term influence scores ($\Delta_j$) alongside domain descriptions to make selection decisions. However, in scenarios where pure regression performance is paramount, or where computational efficiency and immunity to potential LLM hallucinations are prioritized over semantic interpretation, the pruning step can be purely algorithmic.

We introduce *SGED with Algorithmic Pruning*, a variant where the second LLM agent is replaced by a deterministic selection mechanism based strictly on the quantitative feedback provided by the influence scores.

### MECHANISM

In this variant, the core iterative cycle remains largely unchanged (see Section 2.1), but the *Term Pruning Phase* is purely algorithmic and does not employ an LLM. Since $\Delta_j$ explicitly quantifies the marginal contribution of basis function $\psi_j$ to the validation accuracy, it serves as a robust ranking metric.

In multi-output scenarios where the model predicts multiple target variables simultaneously, a single term $\psi_j$ may have a distinct influence score for each target. To ensure that terms critical to *any* dimension of the system are preserved, we define the aggregate influence of a term as its maximum influence across all targets: $\Delta_j^{\text{agg}} = \max_m(\Delta_{j,m})$. Using this aggregate metric, we define two algorithmic pruning strategies:

1. **Top-K Selection:** The candidate terms are ranked in descending order of their aggregate influence scores $\Delta_j^{\text{agg}}$. The top $K$ terms are retained, where $K$ corresponds to the sparsity constraint (e.g., `keep_n_terms`).

2. **Threshold Selection:** Terms are retained if their aggregate influence score exceeds a specific threshold, $\Delta_j^{\text{agg}} > \epsilon$. This allows the model complexity to adapt dynamically to the signal-to-noise ratio of the discovered terms.

This approach effectively decouples the generative creativity of the LLM (used in the "Propose" phase) from the rigorous, data-driven selection process. It eliminates the token cost and latency associated with the second LLM call and removes the possibility of the LLM mistakenly discarding a high-influence term due to a failure in reasoning or instruction following.

### IMPLEMENTATION

The algorithmic modification simplifies the *Propose-and-Prune Cycle*. The modified procedure, incorporating both selection strategies and multi-target aggregation, is outlined in Algorithm 3.

---

**Algorithm 3** Algorithmic Pruning Phase (Variant of Algorithm 2)

---

1: **Input:** Candidate terms $\{\psi_j\}$, Influence scores $\{\Delta_{j,m}\}$ (per target $m$), Config ($K$ or $\epsilon$)
2: ... {Term Generation and Candidate Evaluation unchanged}
3:
4: *// Algorithmic Term Pruning Phase*
5: For each term $j$, compute aggregate influence: $\Delta_j^{\mathrm{agg}} \leftarrow \max_m \Delta_{j,m}$
6: **if** Strategy is **Top-K then**
7:     Sort $\{\psi_j\}$ by $\Delta_j^{\mathrm{agg}}$ in descending order.
8:     $final\_terms_{out} \leftarrow$ Select the top $K$ terms from the sorted list.
9: **else if** Strategy is **Threshold then**
10:     $final\_terms_{out} \leftarrow \{\psi_j \mid \Delta_j^{\mathrm{agg}} > \epsilon\}$
11: **end if**
12:
13: ... {Final Model Fitting and State Update unchanged}

---

PERFORMANCE AND USE CASES

We evaluated the *SGED (top-K)* variant on the LLM-SRBench suite, as detailed in Appendix F. The results demonstrate that this algorithmic variant yields superior numerical accuracy, achieving the best average rank for Normalized Mean Squared Error (NMSE) on both in-distribution and out-of-distribution test sets (see Table 21).

This finding suggests a complementary relationship between the two pruning modes:

- **LLM Pruning – *SGED*:** Better suited for scientific discovery tasks where *interpretability* and *domain alignment* are critical. The LLM can leverage the problem description to retain terms that are physically plausible even if their marginal influence is momentarily lower, or to reject numerically distinct but semantically nonsensical terms.

- **Algorithmic Pruning – *SGED (top-K)*:** Better suited for pure symbolic regression tasks where minimizing error is the sole objective. It provides a more rigorous, greedy optimization path and significantly reduces the computational overhead of the search.

This modularity allows the framework to be adapted to specific experimental and discovery priorities, prioritizing either the domain alignment offered by the LLM or the computational efficiency of deterministic selection.

