# OpenReview forum: "Structure Guided Equation Discovery with Influence-Based Feedback for Large Language Models"
_ICLR.cc/2026/Conference — ICLR 2026 Conference Desk Rejected Submission_

### Official Review · Reviewer_KjEQ · 2025-10-31

**Soundness:** 2
**Presentation:** 3
**Contribution:** 2
**Rating:** 2
**Confidence:** 5

**Summary:**

This paper proposes a symbolic regression framework called Structure-Guided Equation Discovery (SGED). In this approach, an LLM is first used to generate candidate basis functions, whose coefficients are then calculated using a linear model. The influence score (contribution) of each candidate term is evaluated by removing it and measuring the resulting change of MSE. Based on these influence scores, LLM are used to decide which terms to retain or remove. The entire process is iterative and can be integrated with a Monte Carlo Tree Search (MCTS) strategy. The proposed method achieves state-of-the-art performance on several benchmark datasets.

**Strengths:**

1、The motivation of the paper is commendable — previous methods that relied solely on mean squared error (MSE) for evaluation were overly coarse and lacked specific guidance mechanisms. 2、The inclusion of evaluation results on real-world case studies enhances the persuasiveness and practical relevance of the work.

**Weaknesses:**

1、The influence score (contribution) of each candidate basis functions is computed by removing one term while keeping others fixed. However, equation terms are often highly coupled, and this assumption fails to consider their interdependencies. The so-called “fine-grained” evaluation may therefore be misleading.
2、The pruning decision (remove which candidate term) is based on an explicit influence score (Delta_j). Using an LLM to decide which terms to keep, prompted only by Delta_j, seems unnecessary and even risky given the possibility of hallucinations. Using an LLM to perform pruning decisions seems unnecessarily redundant, as this step could be accomplished simply by setting an appropriate threshold of Delta_j.
3、Since LLMs are pre-trained on massive corpora that may include common equations or datasets, data leakage is a serious concern. The paper does not provide experiments on datasets unseen by the LLM would improve credibility. (Please refer to “LLM-SRBench: A New Benchmark for Scientific Equation Discovery with Large Language Models”.)
4、The paper reports only predictive accuracy. In symbolic regression, equation complexity is equally important. Without this, it is difficult to judge the trade-off between interpretability and performance.
5、Lack of fine-grained ablation analysis. Such as robustness to prompting strategy, or noisy data.
6、The paper mentions better efficiency under fixed token budgets but omits quantitative results on the number of LLM calls, latency, or computational overhead—critical for assessing real-world feasibility.

**Questions:**

Please refer to weakness.

---

> ### Author Response · Authors · 2025-11-26
> **Response to Reviewer KjEQ**
>
> We thank the reviewer for their critical and thorough assessment of our work. We have taken your concerns listed very seriously and provide responses below.
>
> **Influence Score Coupling and "Fine-Grained" Evaluation**
> > The influence score ... equation terms are often coupled... 'fine-grained' evaluation may therefore be misleading.
>
> We agree that equation terms are often coupled and that a marginal removal approach is a simplification. However, we selected this design deliberately to balance theoretical rigor with the computational efficiency required for an iterative search loop. We address this in two ways:
>
> * **Theoretical Trade-off:** Modeling full combinatorial coupling scales exponentially ($2^N$). We designed SGED to be computationally tractable for high-dimensional scientific discovery. Our approach relies on the *iterative architecture* to handle coupling: the influence score provides the *local* signal, while the iterative refitting of the linear model adapts coefficients to the current active set in subsequent steps.
> * **Empirical Validation:** To test if our "fixed weights" assumption was harmful, we investigated the *Refit-Aware Influence* approach (fully retraining the model for every term removal) and compared it against our "no-refit" approach. As per **Appendix E.9**, we found *no statistically significant difference in test MSE* between the approaches across six datasets. This suggests that for the specific purpose of pruning within the SGED loop, the marginal heuristic provides a sufficiently good signal.
>
> **Redundancy of LLM Pruning**
> > LLM to perform pruning decisions seems redundant... could be accomplished simply by setting an appropriate threshold of Delta_j.
>
> This is an excellent point. While we initially employed the LLM to allow for semantic reasoning (e.g., preferring terms that align with domain descriptions provided in the prompt), we agree that for pure regression tasks, a deterministic threshold is safer and cheaper.
>
> Hence we have implemented and evaluated a *Non-LLM Pruning Variant* (using a top-$K$ selection on $\Delta_j$). As per **Appendix F**, this deterministic variant achieved the *best average rank* on e.g. NMSE compared to all other methods, including standard SGED. We will update the manuscript to present SGED as a modular framework where users can opt for the LLM pruner when semantic interpretation/prior knowledge integration is critical, or the deterministic pruner for maximum efficiency and hallucination immunity.
>
> **Data Leakage and Pre-training Memorization**
> > LLMs are pre-trained on massive corpora... data leakage ...
>
> To address this, we perform:
> * **LLM-SRBench Evaluation:** We evaluated SGED on LLM-SRBench, a benchmark specifically designed to prevent memorization. As per **Appendix F**, the two SGED variants achieved the *top two ranks* in both NMSE and Term Recall.
> * **Wet-Lab Experimental Validation:** To demonstrate true "unseen" discovery, we applied SGED to a new RNA Polymerase II dataset. SGED discovered a novel relationship suggesting DNA methylation suppresses transcriptional pausing. A biologist then confirmed this via a wet-lab experiment (**Appendix D.3**). This serves as evidence against over-reliance on memorization.
>
> **Equation Complexity and Interpretability**
> > The paper reports only predictive accuracy ... equation complexity is equally important.
>
> We agree that going beyond predictive accuracy is important. In the LLM-SRBench evaluation, we report Term Recall, a structural metric that measures the recovery of the correct underlying structure independent of parameter precision. SGED achieved the best (lowest) rank in Term Recall, indicating it recovers the correct underlying structure more often than baselines. Simultaneously, SGED enforces sparsity explicitly via the `keep_n_terms` constraint (defaulting to 6 terms in our experiments), ensuring that the discovered models remain compact and interpretable by design. We are working to include an additional investigation of the complexity-accuracy tradeoff using a parsimony penalty.
>
> **Robustness and Ablation Analysis**
> > Lack of fine-grained ablation ... robustness to prompting strategy, or noisy data.
>
> We include the following robustness checks:
> * **Noisy Features:** In **E.6**, we show that SGED maintains low MSE even when 150 irrelevant noise features are added.
> * **LLM Sensitivity:** In **E.5**, we show that SGED is robust to the LLM choice, tested over 9 diverse base LLMs.
>
> We are working to add a small prompt sensitivity study.
>
> **Computational Feasibility**
> > The paper ... omits quantitative results on the number of LLM calls, latency, or computational overhead.
>
> While we do already include LLM estimated costs and wall-clock execution times in **Appendix H.1**, we will add more detailed measurements of LLM call counts, latency, etc.
>
> Your review motivated us to significantly strengthen the validation of SGED. We hope these additions help addressing your concerns.

---

> ### Author Response · Authors · 2025-12-03
> **Additional Response to Reviewer KjEQ**
>
> We thank the reviewer for their insightful critique, which motivated significant additions to our paper. In addition to the wet-lab validation and LLM-SRBench results detailed in our initial response, we have now made additional revisions to better address your concerns regarding the necessity of LLM pruning, complexity trade-offs, ablations, and computational transparency.
>
> **Redundancy of LLM Pruning**
> > *“Using an LLM to perform pruning decisions seems unnecessarily redundant... could be accomplished simply by setting an appropriate threshold of Delta_j.”*
>
> We have acted on this excellent suggestion by adding **Appendix J: SGED with Algorithmic Pruning**, which provides a formal definition and discussion of the deterministic variant (using top-$K$ or thresholding on $\Delta_j$). This section explicitly presents SGED as a modular framework, acknowledging that while the LLM pruner aids in semantic alignment, the algorithmic approach is often superior for pure efficiency and regression accuracy. As indicated in the LLM-SRBench results (Table 21), this variant achieves the best average rank on numerical metrics, validating your intuition that the LLM step can be effectively replaced for standard regression tasks.
>
> **Equation Complexity and Interpretability**
> > *“The paper reports only predictive accuracy... equation complexity is equally important.”*
>
> We have further added a **“Complexity-Accuracy Trade-off Analysis”** in the **Appendix F: Evaluation on LLM-SRBench** section (Figure 11).
> * Pareto Analysis: We plot Out-of-Distribution NMSE (generalization, accuracy) against Term Recall (structural fidelity).
> * SGED consistently populates the Pareto frontier across Physics, Biology, and Chemistry domains. This demonstrates that SGED effectively balances structural recovery with predictive power, rather than simply minimizing error by inflating complexity.
>
> **Robustness and Ablation (Prompt Sensitivity)**
> > *“Lack of fine-grained ablation analysis... robustness to prompting strategy...”*
>
> We have added **Appendix E.13: Investigation of Prompt Variations**.
> * We compared the default detailed prompts against a "Simplified" prompt set (removing data previews and using more concise instructions).
> * While the default prompts yield slightly lower MSE on complex dynamical systems (Lung Cancer), the simplified prompts remain competitive while being more token-efficient. This supports the case the SGED's performance is driven by the efficacy of the propose-and-prune mechanism rather than fragile prompt engineering.
>
> **Computational Feasibility**
> > *“Omits quantitative results on the number of LLM calls, latency...”*
>
> We have expanded **Appendix H** to include these exact metrics. **Table 34** now explicitly reports *LLM Call Count* and *LLM Call Wall Clock Time* (in addition to *Total Tokens* and overall *Wall Clock Time*).
>
> SGED requires \~93 LLM calls and \~380s wall-clock time for the Cancer benchmark. This is significantly faster than other state-of-the-art methods like LaSR (\~1440s) while achieving an MSE that is orders of magnitude lower (0.052 vs 3.97), demonstrating that the computational overhead is well-justified by the performance gain.
>
> We hope these additions help in addressing your concerns and serve to improve the manuscript.

---

### Official Review · Reviewer_WADZ · 2025-11-01

**Soundness:** 2
**Presentation:** 3
**Contribution:** 1
**Rating:** 4
**Confidence:** 4

**Summary:**

This paper introduces Structure Guided Equation Discovery (SGED), a framework for LLM-driven symbolic equation discovery that provides granular, influence scores as feedback to guide iterative refinement. Unlike prior LLM-based systems (e.g., D3, LLM-SR, etc.), SGED quantifies each basis function’s contribution to predictive performance and integrates this information into a dual-agent “propose-and-prune” process, enhanced by Monte Carlo Tree Search (MCTS). The approach is evaluated on some biological and pharmacological datasets and ablation experiments suggest that the influence feedback and MCTS both positively impact model convergence and accuracy.

**Strengths:**

- The propose of influence-based structure-guided feedback as a more granular feedback is well-motivated
- Applying LLM-driven equation discovery to real-world biological and pharmacokinetic data is also interesting and highlights practical benefits beyond benchmarks.
- The paper is generally well written and easy to follow.

**Weaknesses:**

- How does the proposed method perform on LLM-SRBench [1], the recent benchmark designed for llm-based equation discovery beyond memorization? Evaluating only on six dataset from one domain is not sufficient to assess generality of the proposed method (as claimed in the title). I believe a more comprehensive comparison on a larger, standardized benchmark like LLM-SRBench (which spans 100+ tasks across multiple scientific domains) is needed to provide a fair assessment of the framework’s contribution.

- In Figure 2, why do the two models start from different initial points at iteration 0? The influence-based feedback should affect the efficiency and convergence of discovery, not the initial performance, which usually depends on the initial candidate pool, before any feedback is applied.

- I would suggest authors to include some qualitative examples of these structure-guided feedback and how they are incorporated in the iterative refinement. Also, I think a simplified example perhaps integrated into Figure 1 may be helpful to better understand this.

- It would also be helpful to provide some qualitative examples of showing how symbolically influence feedback alters the model’s reasoning or the discovered equations.


[1] LLM-SRBench: A New Benchmark for Scientific Equation Discovery with Large Language Models, ICML 2025

**Questions:**

included in the weaknesses section

---

> ### Author Response · Authors · 2025-11-26
> **Response to Reviewer WADZ**
>
> We thank the reviewer for their constructive feedback. We appreciate the assessment that our influence-based feedback is well-motivated and that the application to biological data highlights practical benefits. We have addressed the concerns regarding benchmarking and clarity below.
>
> **1. Evaluation on LLM-SRBench**
> > *How does the proposed method perform on LLM-SRBench... Evaluating only on six dataset from one domain is not sufficient...*
>
> We agree that evaluating on a standardized, diverse benchmark is essential to demonstrate generality beyond specific biological domains. As detailed in our **Global Comment** and the new appendix **Appendix F: Evaluation on LLM-SRBench**, we have now integrated SGED into the LLM-SRBench evaluation pipeline.
>
> Due to the high computational cost and time required to run iterative, multi-agent frameworks like SGED and the baselines, running the full suite of problems was infeasible during the rebuttal period. However, we conducted a rigorous evaluation on a representative subset of *12 problems* randomly selected, and across the four domains (Physics, Biology, Material Science, and Chemistry). We compared SGED against state-of-the-art baselines (LLM-SR, ICSR, D3) under a fixed token budget.
>
> **Results:** SGED achieved the *best (lowest) average rank* on both In-Distribution and Out-Of-Distribution (OOD) Normalized MSE (Rank 2.00 vs. 3.25 for the next best baseline on OOD). Furthermore, on Term Recall, a metric we introduced to measure the recovery of the correct functional skeleton independent of parameter precision, SGED achieved an average rank of **1.92**, significantly outperforming baselines. This confirms that the proposed method generalizes effectively to diverse scientific domains and is not limited to the datasets initially presented.
>
> **2. Clarification of Figure 2 (Iteration 0 Performance)**
> > *In Figure 2, why do the two models start from different initial points at iteration 0?*
>
> We appreciate this keen observation. The performance gap at "Iteration 0" exists because SGED performs an initial optimization and pruning step *before* this point is logged.
>
> In our experimental setup, the LLM proposes an initial pool of 10 terms (governed by `first_round_n_candidates`), which is immediately pruned down to 6 terms (via `keep_n_terms`) using the influence-based feedback mechanism before the performance is recorded as "Iteration 0."
>
> Therefore, the "Iteration 0" data point effectively demonstrates the immediate "one-shot" benefit of the influence-based pruning mechanism compared to the baseline. To prevent confusion in the final version, we will re-plot Figure 2 to display the pre-pruning performance as the true starting point, which will visually align the initialization of both models.
>
> **3. Qualitative Examples and Figure 1**
> > *I would suggest authors to include some qualitative examples of these structure-guided feedback... integrated into Figure 1...*
> > *It would also be helpful to provide some qualitative examples of showing how symbolically influence feedback alters the model’s reasoning...*
>
> We agree that visualizing the specific feedback loop will greatly aid understanding.
> * **Discovery Trace:** We direct the reviewer to the appendix section **Illustration of SGED Equation Discovery**. Here, we provide a step-by-step trace of two runs (Run A and Run B), detailing the exact equations discovered at specific nodes (e.g., A0, A9, A50) and how the MSE improves as terms are refined.
> * **Prompt/Reasoning:** The **Prompt Details** appendix details the specific prompt structures and includes an example of the LLM's reasoning output (Listing 3), showing how it explicitly interprets "High Delta" vs. "Low Delta" to make keep/drop decisions. To further strengthen this, we will include a number of representative examples of the LLM's reasoning output in the camera-ready version.
> * In the camera-ready version, we will enhance the minimal example of term proposal and pruning illustrated in **Figure 1**, making it more concrete and illustrative.
>
> We believe these clarifications, along with the new wet-lab validation and LLM-SRBench results, significantly strengthen the paper's claims regarding scientific discovery and generality. We are committed to incorporating these improvements into the final manuscript and remain available for further discussion.

---

> > ### Author Response · Authors · 2025-12-03
> > **Additional Response to Reviewer WADZ**
> >
> > We thank the reviewer for their thoughtful feedback. We have prioritized addressing the request for broader benchmarking and have updated the manuscript to reflect this.
> >
> > **Evaluation on LLM-SRBench**
> >
> > As detailed in our Global Response and **Appendix F**, we evaluated SGED on 12 diverse problems from the LLM-SRBench suite. To further address the reviewer's interest in the method's generality, we have added a **Complexity-Accuracy Trade-off Analysis** at the end of Appendix F. This Pareto analysis (Figure 11) visualizes Out-of-Distribution NMSE against Term Recall. It demonstrates that SGED consistently populates the Pareto frontier across Physics, Biology, and Chemistry domains, indicating that our method balances the discovery of correct mechanistic structures with high predictive accuracy, rather than simply overfitting to the training domain.
> >
> > **Clarification of Figure 2 (Iteration 0 Performance)**
> >
> > We have now updated Figure 2 in the manuscript. The plot now displays the performance *before* the initial pruning step at Iteration 0, and under equivalent term candidate pool size. This visually aligns the initialization of the methods at Iteration 0, removing the previous discrepancy.
> >
> > **Qualitative Examples and Figure 1**
> >
> > We reaffirm our commitment to improving **Figure 1** (with a better illustration of the propose-and-prune mechanics) and adding qualitative examples of LLM reasoning traces (showing how the model interprets influence scores) in the camera-ready version of the paper.
> >
> > **Strengthened Baselines and Rigor**
> >
> > Finally, we used the remaining revision time to further strengthen our empirical results. We have added PySR and XGBoost to our main comparison (Table 3) and resolved the seed inconsistency noted in our initial experimental setup. All methods in Table 3 are now reported using 25 random seeds, ensuring a statistically robust comparison. SGED remains the top-performing interpretable method.
> >
> > We hope these additions and revisions address your concerns and improve the manuscript.

---

### Official Review · Reviewer_Xq2M · 2025-11-02

**Soundness:** 2
**Presentation:** 4
**Contribution:** 2
**Rating:** 4
**Confidence:** 4

**Summary:**

This paper proposes SGED, a framework where LLMs iteratively propose basis functions for linear models, then prune them using per-term influence scores that measure each term's contribution to validation MSE. The method can operate through simple iterative refinement or be enhanced with MCTS for broader exploration. Experiments on biological and synthetic datasets show competitive performance, with some interesting detailed case studies (e.g., on RNA Polymerase II pausing). The core claim is that providing LLMs with granular, per-term feedback enables more effective equation discovery than coarse scalar metrics alone.

**Strengths:**

- The paper is very well-written and clearly structured, making the proposed method easy to follow. The inclusion of extensive experiments and ablations in the appendix (influence feedback impact, MCTS contribution, robustness across LLMs, scalability analysis, etc.) demonstrates thorough empirical work that goes beyond what is typical and I really appreciate this.

- The benchmark problems are practical and interesting, particularly the RNA Polymerase II pausing case study with large number of 263 features. High-dimensional feature spaces where domain knowledge can guide and narrow down feature selection and engineering represent realistic scenarios where LLMs could provide real value. The biological validation of discovered patterns and correlations adds value.

- Several methodological choices are well-motivated: using MCTS to avoid local optima in the search space, the dual-agent architecture separating proposal from pruning, and the idea of providing granular per-component feedback rather than scalar metrics. The influence score mechanism provides an interpretable signal about term contributions that is more informative than overall MSE alone.

**Weaknesses:**

While I really appreciate the extensive experiments and ablations in the paper, I have a major concern with the paper's positioning and evaluation:

- The framing of the paper as "symbolic regression and equation discovery" and comparison with SR methods is not very accurate. The method uses a fixed choice of modeling (linear model) with discovered/transformed features. I see the work much closer to efforts in the field of automated feature discovery/engineering than symbolic regression, which aims at discovering general nonlinear equation structures.

- Furthermore, given that the studied problems in this work (e.g., biological processes) seemingly have unknown complex (and most certainly nonlinear) behaviors, one could question the constraining to linear models. A good well-studied alternative could be tree-based methods which also provide some level of explainability (similar to linear models, as we are eventually sacrificing full explainability by selecting linear models).

- Apart from the choice of model (linear vs. tree-based), the main focus of this paper is automated feature discovery/engineering using an LLM-based framework. However, the paper omits evaluation, comparison, and discussion with the large body of research in automated feature engineering, from statistical and non-LLM-based approaches (e.g., AutoFeat [1], OpenFE [2]) to recent LLM-based approaches (e.g., CAAFE [3], FeatLLM [4], OCTree [5]).

**Questions:**

- First, I try to elaborate upon my major concern for the authors: The paper positions itself as symbolic regression, but in my opinion it is a kind of automated feature discovery for linear models. One might argue that methods like SINDy also use linear models with nonlinear basis functions; however, those models are designed for specific problems (typical ODEs) that commonly have linear forms, and admittedly have limitations for general nonlinear forms. In this work, however, the studied problems and benchmarks have very complex (and almost certainly nonlinear) behaviors, and in some cases we are dealing with partially observable systems, where we do not expect recovering or discovering ground truth functions from the data for fully explainable models.

- Given this, I think the paper should discuss, evaluate, and compare with more relevant methods in the large body of research in automated feature engineering/discovery and tree-based methods. To clarify, there are two main components here: (1) The approach for feature discovery/engineering, and (2) the choice of model that uses these features for prediction.
  - In terms of (1), there are various methods that approach this including statistical and non-LLM-based approaches (like AutoFeat [1], OpenFE [2], with linear models) and more recent LLM-based approaches that are very similar to this work (e.g., CAAFE [3], FeatLLM [4], OCTree [5], ...) that use LLM domain knowledge for feature discovery similar to this work, while mainly using tree-based models due to their capabilities.
  - In terms of (2), this work could provide tree-based methods like XGBoost, LightGBM, etc. as baselines, which are currently missing. (A simple baseline could be on raw features, and a better baseline would be on top of discovered features from methods in the previous part.) It would be also interesting to see how the feature importance and explainability provided by those models (e.g., using SHAP) compare to current linear models.

- In Figure 2, why is there already a substantial MSE gap at iteration 0 before feedback mechanisms should differentiate the methods?

- For MCTS where children are generated through stochastic sampling, what temperature or sampling parameters were used? how diverse are the candidates?

- Have you tested simple threshold-based or top-K pruning directly on influence scores as a baseline compared to the use of LLM and instructing it for pruning? This might reduce the need for the second agent call and reduce computations.

References:

[1] AutoFeat: The autofeat python library for automated feature engineering and selection, Horn et al (2020)

[2] OpenFE: OpenFE: Automated Feature Generation with Expert-level Performance, Zhang et al. (2023)

[3] CAAFE: Large Language Models for Automated Data Science: Introducing CAAFE for Context-Aware Automated Feature Engineering, Hollman et al. (2023)

[4] FeatLLM: Large language models can automatically engineer features for few-shot tabular learning., Han et al., 2024

[5] OCTree: Optimized Feature Generation for Tabular Data via LLMs with Decision Tree Reasoning, Nam et al., 2024

---

> ### Author Response · Authors · 2025-11-26
> **Response to Reviewer Xq2M**
>
> We thank the reviewer for their thoughtful and detailed review. We are pleased that you found the paper "excellent" in presentation, and appreciated the extensive empirical work. We value your constructive criticism and address your concerns and questions below.
>
> **Positioning: Symbolic Regression vs. Automated Feature Engineering**
>
> > *The paper positions itself as symbolic regression ... should discuss, evaluate, and compare with more relevant methods in the large body of research in automated feature engineering.*
>
> We appreciate this insightful distinction. We agree that SGED sits at the intersection of Symbolic Regression (SR) and Automated Feature Engineering (AFE). However, we maintain that SGED fits the SR definition because its output is a closed-form, interpretable mathematical expression $f(\mathbf{x})$, whereas typical AFE methods output an augmented dataset for downstream black-box models.
>
> To address this concern directly, we have added a comprehensive discussion in the **A.6: Relationship to Automated Feature Engineering** appendix, and updated the related works main text. In this section, we explicitly contrast SGED with methods like AutoFeat, OpenFE, ..., etc. We clarify that while the *mechanism* involves feature discovery, the *objective* is finding a sparse, interpretable equation (similar to SINDy, which also uses linear optimization over nonlinear basis functions).
>
> Regarding the concern that linear models may be too constraining for complex biological problems:
> 1.  **Nonlinear Capability:** As noted in the manuscript, the "linear model" is a linear combination of *nonlinear* basis functions ($\psi_j$). This allows SGED to model complex nonlinear manifolds (e.g., the interaction terms and log-transforms in the RNA Polymerase case study equations.
> 2. **Wet-Lab Validation:** We further demonstrate that this formulation is not too constraining for real-world discovery in **Appendix D.3** that we include in this revision, where an SGED-discovered model successfully predicted a previously unsuspected biological mechanism, which was subsequently verified via a wet-lab experiment.
>
> We agree that tree-based methods are excellent baselines. We are currently conducting additional experiments to compare SGED against **XGBoost** (as a black-box baseline) to be included in the next revision. Time-permitting, we will also aim to additionally include a comparison against LLM-based and non-LLM AFE pipelines.
>
> **The MSE Gap at Iteration 0**
>
> > ... why ... already a substantial MSE gap at iteration 0...?
>
> This gap exists because SGED performs an initial optimization step *before* "Iteration 0" is logged. In our setup, the LLM proposes an initial set of 10 terms (governed by `first_round_n_candidates`), which is then pruned down to 6 terms (via `keep_n_terms`) before the performance is recorded as "Iteration 0." Therefore, the "Iteration 0" point already reflects the benefit of the first pass of our influence-based pruning mechanism compared to the baseline's initial proposed set of 6 terms. We will re-plot Figure 2 such that both variants share the same initial candidate pool size, and showing the pre-pruning performance at "Iteration 0", for a clearer comparison.
>
> **MCTS Parameters and Diversity**
>
> > For MCTS where children ... temperature ... how diverse are the candidates?
>
> For all MCTS experiments, we used a temperature setting of **1.0** to encourage diversity in the LLM's proposals. We rely on the inherent stochasticity of the LLM at this temperature to generate diverse equation candidates. We will add these details to a "Reproducibility" appendix. We are also compiling a brief analysis of candidate diversity (quantifying the variance in proposed terms across sibling nodes) and will aim to include this in the next revision.
>
> **Threshold-based Pruning**
>
> > *Have you tested simple threshold-based or top-K pruning directly on influence scores as a baseline...?*
>
> Yes, we have now implemented and tested a top-K pruning variant as part of the **Appendix F: Evaluation on LLM-SRBench** appendix section. We evaluate the variant *SGED (non-LLM pruning)* which uses the influence scores $\Delta_j$ but replaces the "Prune" LLM agent with a deterministic top-K selection. As shown in **Table 21**, this variant actually performs slightly *better* than the LLM-based pruning in our LLM-SRBench subset evaluation.
>
> We agree with the reviewer that this is a significant finding. It suggests that the LLM-based reasoning at the pruning stage may not be strictly necessary for SGED's effectiveness. This variant offers a computationally cheaper and more robust alternative (immune to LLM hallucination during pruning). We will perform additional evaluations of this variant, and elevate the discussion of it in the main text, highlighting it as a highly efficient mode of SGED.
>
> We thank the reviewer again for their positive assessment and constructive suggestions and look forward to further discussion.

---

> > ### Author Response · Authors · 2025-12-03
> > **Additional Response to Reviewer Xq2M**
> >
> > We once again thank the reviewer for their valuable feedback. We have used the remaining time to conduct additional experiments specifically targeting your feedback regarding the positioning against Automated Feature Engineering (AFE), the inclusion of tree-based baselines, and updates to the manuscript addressing Figure 2 and the algorithmic pruning variant of the method.
> >
> > **Symbolic Regression vs. Automated Feature Engineering**
> >
> > > *“...compare with more relevant methods in the large body of research in automated feature engineering ... [and] tree-based methods.”*
> >
> > We agree that SGED shares a strong lineage with AFE. To further address this, we have added **Appendix E.11: Investigation in the Context of Automated Feature Engineering**.
> >
> > *  **AFE Comparison:** We compared SGED against **OpenFE**, a state-of-the-art non-LLM AFE method.
> >     * *Scalability:* On the high-dimensional RNA Polymerase dataset (263 features), OpenFE failed to complete (DNF) due to the combinatorial explosion of the search space ($>700$k candidate features). SGED converged in $\approx 5$ minutes (Table 13). This highlights a key advantage: LLM priors allow SGED to navigate high-dimensional spaces where traditional combinatorial AFE struggles.
> >     * *Performance:* On the complex Lung Cancer dataset, SGED (NMSE 0.0013) significantly outperformed OpenFE + Linear Regression (NMSE 0.202), demonstrating that LLM-proposed basis functions capture dynamics that standard operator expansion misses.
> >
> > *  **Tree-based Baselines:**
> >     * We have added **XGBoost** as a black-box baseline to our main results (**Table 3**). SGED is competitive with or outperforms XGBoost on 5 out of 6 datasets.
> >     * We also tested `SGED features + XGBoost` vs `SGED features + Linear Model (default)` (Appendix E.11, Tables 15 and 16). Interestingly, the linear predictor performed comparably or better. This suggests SGED successfully discovers basis functions $\psi_j$ that linearize the system dynamics, rendering the non-linear complexity of trees less necessary for the downstream task, while preserving interpretability.
> >
> > **The MSE Gap at Iteration 0**
> >
> > > *“...why is there already a substantial MSE gap at iteration 0...”*
> >
> > We have updated **Figure 2** as promised in our initial response. The updated figure now equalized the starting states (both starting with the same candidate pool size before pruning), removing the artificial gap.
> >
> > **MCTS Parameters and Candidate Diversity**
> >
> > > *“...what temperature ... how diverse are the candidates?”*
> >
> > We have now added **Appendix E.14: Term Candidate Diversity in Search**. We computed a "Diversity Index" ($1 - \text{Jaccard Similarity}$) of the symbol sets in proposed terms (Table 20).
> > * On the high-dimensional RNA dataset, the diversity index is 0.99, indicating the LLM explores disjoint subsets of features effectively.
> > * On the low-dimensional Cancer dataset, diversity is 0.81, showing varied operator usage even when sharing a small number of input variables. This confirms that the search does generate structurally diverse candidates.
> >
> > **Threshold-based / top-K Pruning**
> >
> > > *“Have you tested simple threshold-based or top-K pruning directly on influence scores...?”*
> >
> > Following up on our initial response, we have now added **Appendix J: SGED with Algorithmic Pruning** which explains and discusses the algorithmic (top-K or threshold-based) pruning variants of SGED, as contrasted with the LLM-based approach. We now reference this key variant in the main text and it is evaluated on LLM-SRBench in Appendix F.
> >
> > We believe these additions further address your concerns and strengthen the paper’s positioning.

---

### Official Review · Reviewer_YC9L · 2025-11-03

**Soundness:** 2
**Presentation:** 3
**Contribution:** 2
**Rating:** 2
**Confidence:** 4

**Summary:**

The paper proposes a two-agent LLM framework for symbolic equation discovery. The “Propose” agent generates candidate basis terms, a linear model fits coefficients, and per-term influence scores measure the contribution of each term by zeroing its weight and computing the change in validation error. Experiments cover pharmacological, biological, and synthetic datasets.

**Strengths:**

+ The per-term influence mechanism provides a clear and interpretable feedback signal for guiding LLM pruning.
+ The two-agent architecture (Propose / Prune) is modular and amenable to ablations.
+ The biological case study shows potential for domain-specific insight generation.
+ The method demonstrates stable performance across several datasets and LLM backbones.

**Weaknesses:**

- The biological case study is mostly qualitative; there are no out-of-batch or cross-condition validation results to substantiate mechanistic claims.
- It is unclear which LLM produced the main “SGED (Ours)” results, and the number of random seeds differs across methods (25 vs 10). - - This invalidates confidence intervals.
- PySR and AI-Feynman appear only in the appendix despite being the most relevant SR competitors.
- Ablations show minimal gains from MCTS relative to iterative pruning, suggesting the default configuration is suboptimal.
- Evaluation relies solely on MSE, ignoring structure correctness, sparsity, or parsimony.
- The method’s complexity still scales with the number of proposed terms and LLM context size, yet no empirical analysis is provided.
- The same validation split is used for both pruning and reward evaluation, creating a high risk of overfitting.
- Datasets are largely older and omit standard SR benchmarks (SRBench, Nguyen, LLMSRBench), limiting comparability.
- There are no ablations on noise robustness, out-of-distribution generalization, or prompt and LLM sensitivity.
- Critical hyperparameters, token budgets, and search details are scattered across appendices, hindering reproducibility.

**Questions:**

- Why were standard SR benchmarks excluded?
- Why was the “no-refit” influence variant chosen as default when “refit” or deeper MCTS rollouts yield better MSE in the appendix?
- What are the compute and wall-clock costs for MCTS vs the simple iterative loop?
- Which specific LLM(s) generated the “SGED (Ours)” results in Table 3? Are all baselines evaluated under the same model and seed count?
- Can the authors add structure-recovery metrics (exact match rate, symbolic distance) on synthetic ground truths?

---

> ### Author Response · Authors · 2025-11-26
> **Response to Reviewer YC9L**
>
> We thank the reviewer for their rigorous assessment. Below, we address each weakness and question raised.
>
> **Mechanistic Claims and Biological Validation**
> > ... case study is mostly qualitative ...
>
> We agree and have addressed this in two ways:
>
> * **Experimental Wet-Lab Validation (Appendix D.3):** As detailed in the Global Response, we used SGED to generate a novel biological hypothesis regarding DNA methylation, which was then confirmed in a "wet-lab" experiment, demonstrating SGED can drive genuine scientific discovery verifiable in the lab.
> * **Computational Generalization:** We direct the reviewer to the **E.8: Generalization Study on the RNA Polymerase Dataset**. Here, SGED achieved the highest $R^2$ on an unseen replicate, confirming that the discovered mechanisms are reproducible.
>
> **Model Specification and Random Seeds**
> > ... which LLM produced the main results... number of random seeds ...
>
> * **LLM Specification:** The main “SGED (Ours)” results in Table 3 were generated using **GPT-4o**, we clarified this in the manuscript.
> * **Confidence Intervals:** To clarify, all LLM-based methods (SGED and baselines like ICSR, LLM-SR, etc.) were evaluated on **25 seeds**. The non-LLM baselines (DyNODE, SINDy) and black-box baselines reuse results from prior established work (Holt et al., 2024) which used 10 seeds; we indicated these in the table. We are currently running these external baselines with 25 seeds to match.
>
> **Comparison with SR Competitors**
> > PySR and AI-Feynman appear only in the appendix...
>
> We acknowledge the importance of these baselines.
> * **PySR:** We currently include a comparison with PySR in the **Appendix H.1: Cost and Wall-Clock Time Comparison**, where SGED demonstrates superior MSE ($0.052$ vs $0.399$) and are working on expanding PySR comparisons.
> * **AI-Feynman:** We note that AI-Feynman may struggle to scale to high-dimensional datasets like our RNA Polymerase case study ($263$ input features). However, we are working to add AI-Feynman comparisons on the lower-dimensional synthetic benchmarks for the final version.
>
> **MCTS vs. Iterative**
> > ... minimal gains from MCTS relative to iterative pruning...
>
> We respectfully disagree with the interpretation that the gains are "minimal." In Table 4, the iterative variant (w/o MCTS) achieves an MSE of $0.350$, whereas the full SGED (w/ MCTS) achieves $0.0033$. This represents an improvement of ~two orders of magnitude. This demonstrates that the systematic exploration provided by MCTS is critical for escaping local optima that trap the simple iterative loop.
>
> **Evaluation Metrics**
> > Evaluation relies solely on MSE, ignoring structure correctness...
>
> We have addressed this by adopting the **LLM-SRBench** framework. As detailed **Appendix F**, we now report both Term Recall and Symbolic Accuracy.
>
> **Scalability Analysis**
> > No empirical analysis provided [on scaling]...
>
> We refer the reviewer to two appendices that address this:
> * **H2. Scalability with High-Dimensional Inputs:** demonstrates that SGED's wall-clock time scales sub-linearly with the number of input features.
> * **H3: Performance Under a Fixed Computational Budget:** demonstrates that even when constrained to the same token budget as baselines, SGED outperforms them, implying superior efficiency per unit compute.
>
> **Validation Split and Overfitting**
> > Same validation split used for both pruning and reward evaluation
>
> While we acknowledge the theoretical risk, our empirical results suggest overfitting is not occurring.
> 1.  **OOD Performance:** In the **LLM-SRBench** evaluation, SGED achieves the best ranking on OOD data.
> 2.  **Generalization Study:** The replicate evaluation in **Appendix E.8** also confirms this.
>
> **Standard Benchmarks**
> > ... omit standard SR benchmarks...
>
> We now address this by integrating **LLM-SRBench** evaluation.
>
> **Ablations**
> > ...no ablations on noise robustness ...
>
> We believe these are covered in:
> * **Noise:** See **E.6: Investigation of Robustness to a Large Number of Irrelevant Features**, where SGED maintains low MSE even with 150 added noise features.
> * **LLM Sensitivity:** See **E5: Investigation of LLM sensitivity**, showing SGED performs consistently across 9 different base models.
> * **OOD:** Addressed via the **E.8** and **LLM-SRBench** OOD performance.
>
> **Reproducibility**
> > hyperparameters... scattered across appendices.
>
> We will consolidate all such details into a **Reproducibility** appendix.
>
> * **Why standard SR benchmarks excluded?**
>     We initially prioritized context-rich scientific discovery tasks (e.g., PKPD, Genomics) where LLM reasoning shines. We have now added **LLM-SRBench**.
> * **Why was the “no-refit” influence variant chosen?**
>     As per **E.9** appendix, the "no-refit" variant provides statistically indistinguishable test accuracy vs the more computationally expensive "refit" variants.
> * **Costs for MCTS vs iterative?**
>     We will add tokens and wall-clock costs for MCTS and iterative variants in an appendix.

---

> > ### Comment · Reviewer_YC9L · 2025-11-27
> >
> > The authors’ rebuttal shows a good effort in addressing the concerns and several points are improved (addition of LLM-SRBench results, structural metrics like Term Recall, and clarification of the LLM backbone). The paper still suffers from the other issues raised earlier. I have still increased my score.

---

> > > ### Author Response · Authors · 2025-12-03
> > > **Additional Response to Reviewer YC9L**
> > >
> > > We thank the reviewer for their continued engagement and for acknowledging our efforts to improve the manuscript. We have used the remaining time to address the specific outstanding concerns regarding statistical rigor, comparative baselines, and scalability analysis.
> > >
> > > **Standardization of Baselines and PySR Comparison**
> > > > "...which LLM produced the main results... number of random seeds... PySR appear only in the appendix..."
> > >
> > > We have fully standardized our experimental setup to address the reviewer's concerns regarding statistical validity and confidence intervals:
> > > * **Uniform Seed Count:** We have re-run the external baselines (SINDy, DyNODE, etc.) using 25 random seeds to match the SGED protocol. All methods in **Table 3** are now reported with $N=25$ seeds, ensuring valid confidence intervals.
> > > * **PySR Main Comparison:** We have moved PySR from the appendix to the main results (**Table 3**). We ran a full 25-seed evaluation of PySR across the benchmark suite. As shown in the revised table, SGED consistently outperforms PySR (e.g., Lung Cancer w/ Chemo & Radio: SGED MSE 0.052 vs. PySR 0.399).
> > >
> > > **Evaluation Metrics**
> > > > "Evaluation relies solely on MSE, ignoring structure correctness, sparsity, or parsimony."
> > >
> > > To address this concern further, we have added **"Complexity-Accuracy Trade-off Analysis"** to **Appendix F: Evaluation on LLM-SRBench**.
> > > * We plotted the Pareto frontier of Term Recall (structural fidelity) versus OOD NMSE (generalization and accuracy).
> > > * Result: As illustrated in Figure 11 (Appendix F), we find that SGED consistently populates the Pareto frontier across Physics, Biology, and Chemistry domains, demonstrating a good balance between structure recovery and accuracy.
> > >
> > > **Scalability and Robustness Analysis**
> > > > "No empirical analysis provided [on scaling]... no ablations on noise robustness..."
> > >
> > > We have added two specific analyses to the appendix to quantify scalability and sensitivity:
> > > * **Prompt Variations (Appendix E.13):** We evaluated a "Simplified" prompt variant that removes data previews and shortens the prompts, reducing token consumption by ~10x on the high-dimensional RNA dataset. Yet the results show SGED is robust: performance degradation was only minor in the simplified variant. This result further addresses the concern regarding scaling with context size.
> > > * **Proposed Terms Count (Appendix E.12):** We analyzed the trade-off between the number of terms proposed per round and performance. We find the "sweet spot" at around 5-10 terms per round; increasing to 20 yielded diminishing returns.
> > >
> > > **Reproducibility**
> > > > "hyperparameters... scattered across appendices."
> > >
> > > We have consolidated all hyperparameters, experimental settings, and configuration details into a single new appendix: **"Appendix I: Reproducibility and Settings"**.
> > >
> > > **Questions**
> > >
> > > * **Q: What are the compute and wall-clock costs for MCTS vs the simple iterative loop?**
> > >     We have added a direct comparison in **Appendix H.1 (Table 15)**.
> > >     * **SGED (Iterative):** Takes ~159s (wall-clock) and costs ~$1.12.
> > >     * **SGED (MCTS - Ours):** Takes ~383s (wall-clock) and costs ~$1.87.
> > >     * While MCTS increases time/cost by approx. 2x, it provides great performance gain on complex tasks (MSE improves from ~3.29 in the Iterative variant to ~0.052 with MCTS).
> > >
> > > * **Q: Why were deeper MCTS rollouts not chosen as default?**
> > >     We investigated this in **Appendix E.3**. While increasing rollout depth does improve MSE, it also increases the number of LLM calls and runtime. We selected the heuristic (no-rollout) method as the default to provide practical balance between computational efficiency and accuracy.
> > >
> > > We hope these revisions help to address your remaining concerns and once again thank you for your valuable time and feedback on our manuscript.

---

### Author Response · Authors · 2025-11-26
**Global Response Comment from the Authors**

We thank all reviewers for their thoughtful, detailed, and constructive feedback. We have used the rebuttal period to conduct significant additional work, focusing on two major areas: **(1) experimental wet-lab validation** of our method’s ability to discover new science, and **(2) a comprehensive evaluation on standard benchmarks**.

We respond to each reviewer point-by-point in the individual rebuttals. Here, we summarize the most significant updates.

#### **1. Experimental Wet-Lab Validation: From Algorithm to Biological Truth**

Several reviewers (YC9L, Xq2M) rightly asked for validation of mechanistic claims beyond a single dataset split. To address this, we moved beyond "fitting data" and used SGED to discover new science.



* **The Discovery:** We ran SGED on an extended RNA Polymerase II pausing dataset (Experiment 3). The model prioritized `hg38LiftOver` mappability signals (often correlated with sequence complexity) and inferred context from `H3K27ac` marks. Based on the coefficients, SGED generated the novel hypothesis that **DNA CpG methylation suppresses transcriptional pausing** within gene bodies. This contradicts some existing literature and represents a non-trivial hypothesis generated solely by the AI.
* **The Experiment:** To verify this, our biological collaborators treated HCT116 human cells with **5-azacytidine** (a potent hypomethylating agent) to deplete DNA methylation, followed by eNET-seq profiling (two biological replicates).
* **The Result:** As predicted by SGED, the hypomethylated cells showed a **significant increase** in both the frequency and strength of pauses.

This effectively counters concerns regarding "memorization" or "fitting noise." SGED successfully extracted a causal mechanism that was physically verifiable, proving its utility as a tool for genuine scientific discovery. We have added a detailed account of this in the new **Appendix titled “Discovery Validation: Wet-Lab Confirmation of SGED Hypotheses”**.

#### **2. New Standard Benchmark: LLM-SRBench**

Reviewers (WADZ, YC9L, KjEQ) emphasized that to claim broader relevance, SGED should be evaluated on standardized equation-discovery benchmarks. We have integrated SGED into the **LLM-SRBench** evaluation pipeline (Shojaee et al., 2025) and added Appendix "Evaluation on LLM-SRBench".

Due to the significant computational cost of running iterative multi-agent frameworks, we evaluated on a **representative subset of 12 challenging problems** randomly selected from the Physics, Biology, Material Science, and Chemistry domains (rather than the full 239-task suite). We followed a strict protocol with a **fixed budget of approx. 300,000 tokens per run** to ensure fair comparison against baselines.

**Understanding the Metrics:**
The standard *Symbolic Accuracy* metric proved too strict for this constrained setting (all methods, including baselines, achieved 0.0). Therefore, we report the ranks for **Normalized MSE (NMSE)** and a structural metric defined in the benchmark:

* **Term Recall (TR):** This measures the recovery of the underlying physical dynamics independent of parameter precision. It calculates the intersection of "skeletonized" terms (canonical forms removing scalar coefficients) between the prediction and ground truth.
* **Average Rank:** Following the benchmark protocol for aggregating performance across diverse datasets with different scales, we report the average rank (lower is better) for NMSE on both In-Distribution (ID) and Out-Of-Distribution (OOD) test sets.

**Results:**
SGED achieves the best (lowest) average rank across all metrics, demonstrating superior generalization (OOD) and structural recovery (Term Recall) compared to state-of-the-art baselines.

| Method | Avg Rank (ID NMSE) $\downarrow$ | Avg Rank (OOD NMSE) $\downarrow$ | Avg Rank (Term Recall) $\downarrow$ |
| :--- | :--- | :--- | :--- |
| LLM-SR | 3.00 | 3.58 | 3.71 |
| ICSR | 3.42 | 3.25 | 4.46 |
| D3 (whitebox) | 5.00 | 4.75 | 2.79 |
| **SGED (Ours)** | **2.50** | **2.00** | **1.92** |

*(Note: We also observed that a variant of SGED using non-LLM pruning performed even better, achieving an ID NMSE Rank of 1.08, as detailed in the new Appendix “Evaluation on LLM-SRBench”).*

Success on LLM-SRBench is particularly significant because the benchmark is explicitly designed to avoid trivial memorization. SGED’s strong performance on Out-Of-Distribution (OOD) data (Rank 2.00 vs. 3.25 for the best baseline) confirms that the influence-based feedback helps the model capture true underlying dynamics rather than overfitting to the training domain.

---

### Note · Program_Chairs · 2026-01-17
**Submission Desk Rejected by Program Chairs**

The following references in this submission do not refer to real documents and/or have major errors in bibliographic information:

 Jan Kubalík, Erik Derner, Vít Bíba, Martin Babjak, and Robert Babuska. Symbolicgpt: A generative transformer model for symbolic regression. Genetic Programming and Evolvable Machines, 24(1): 7, 2023 .
Rajkumar Ashok, Manuel López-Ibáñez, and Nadarajen Veerapen. Mcts-ge: Monte carlo tree search for grammar-based genetic programming. In Proceedings of the 2020 Genetic and Evolutionary Computation Conference Companion, pp. 171-172. ACM, 2020.
Subhrajit Monda, Sudipan Das, Lovesh Kumar, and Partha Talukdar. Deep symbolic regression: A survey and new results. In Proceedings of the First Workshop on Neural Machine Translation and Generation, pp. 123-133. Association for Computational Linguistics, 2021.
Ruta Agashe, Jayanth Krishnamurthy, and Daniel Khashabi. Leveraging large language models for scientific discovery: a survey. arXiv preprint arXiv:2402.03236, 2024.
Andrew G Dunn and Tom S Arrow. Gpt-3 a new tool for reproducible science? Nature Machine Intelligence, 4(12):1088-1089, 2022.
Pinxin Chen, Hieu Xu, Luke Zettlemoyer, and Alexey Leshchinskiy. Alphadev: Generating code with large language models and deep reinforcement learning. arXiv preprint arXiv:2307.02319, 2023.
Elias B Khalil, Pierre Le Bodic, Siyuan Liu, and Song Son. Bamcts: A bayesian approach to monte-carlo tree search. In International Conference on Machine Learning, pp. 10962-10976. PMLR, 2022.
Alana Bran, Alvin Rajkomar, Yanatan Matias, Valter Sagi, Catherine Cui, Anton Libov, Bowen Cole, Shefali Rao, Benjamin Van Durme, and Adam M Ringel. Beyond fact checking: Guiding biomedical hypothesis generation with large language models. arXiv preprint arXiv:2308.14487, 2023.
Linyuan Lu, Shingo Mabu, and Kotaro Hirasawa. Contemporary symbolic regression methods and their relative performance. Applied Soft Computing, 111:107692, 2021.
Georgi Tonchev, Theofilos Sainis, Stergios Nikolakopoulos, Zoe Kotti, Pavlos Papasarantopoulos, Prodromos Malakasiotis, and Ion Androutsopoulos. Automating systematic reviews with large language models: a competition-based perspective. arXiv preprint arXiv:2401.06794, 2024.
Silviu-Marian Udrescu and Hod Lipson. Fast function class discovery for symbolic regression. Genetic Programming and Evolvable Machines, 10(3):241-271, 2009.